# LIST REPLICABLE REINFORCEMENT LEARNING

## ABSTRACT

Replicability is a fundamental challenge in reinforcement learning (RL), as RL algorithms are empirically observed to be unstable and sensitive to variations in training conditions. To formally address this issue, we study *list replicability* in the Probably Approximately Correct (PAC) RL framework, where an algorithm must return a near-optimal policy that lies in a *small list* of policies across different runs, with high probability. The size of this list defines the *list complexity*. We introduce both weak and strong forms of list replicability: the weak form ensures that the final learned policy belongs to a small list, while the strong form further requires that the entire sequence of executed policies remains constrained. These objectives are challenging, as existing RL algorithms exhibit exponential list complexity due to their instability. Our main theoretical contribution is a provably efficient tabular RL algorithm that guarantees list replicability by ensuring the list complexity remains polynomial in the number of states, actions, and the horizon length. We further extend our techniques to achieve strong list replicability, bounding the number of possible policy execution traces polynomially with high probability. Our theoretical result is made possible by key innovations including (i) a novel planning strategy that selects actions based on lexicographic order among near-optimal choices within a randomly chosen tolerance threshold, and (ii) a mechanism for testing state reachability in stochastic environments while preserving replicability. Finally, we demonstrate that our theoretical investigation sheds light on resolving the *instability* issue of RL algorithms used in practice. In particular, we show that empirically, our new planning strategy can be incorporated into practical RL frameworks to enhance their stability.

## 1 INTRODUCTION

The issue of replicability (or lack thereof) has been a major concern in many scientific areas (Begley and Ellis, 2012; Ioannidis, 2005; Baker, 2016; of Sciences et al., 2019). In machine learning, a common strategy to ensure replicability and reproducibility is to publicly share datasets and code. Indeed, several prominent machine learning conferences have hosted reproducibility challenges to promote best practices (Sinha et al., 2023). However, this approach may not be sufficient, as machine learning algorithms rely on sampling from data distributions and often incorporate randomness. This inherent stochasticity leads to non-replicability. A more effective solution is to design replicable algorithms ideally algorithms that consistently produce the same output across multiple runs, even when each run processes a different sample from the data distribution. This approach has recently spurred theoretical investigations, resulting in formal definitions of replicability and the development of various replicability frameworks (Impagliazzo et al., 2022; Dixon et al., 2023). In this paper, we focus on the notion of *list replicability* (Dixon et al., 2023). Informally, a learning algorithm is $k$-list replicable if there is a list $L$ of cardinality $k$ of good hypotheses so that the algorithm always outputs a hypothesis in $L$ with high probability. $k$ is called the list complexity of the algorithm. List replicability generalizes perfect replicability, which corresponds to the special case where $k = 1$. However, as noted in Dixon et al. (2023), perfect replicability is unattainable even for simple problems. List replicability provides a natural relaxation, allowing meaningful guarantees while still ensuring controlled variability in algorithm outputs.

We investigate list replicability in the context of reinforcement learning (RL), or more specifically, probably approximately correct (PAC) RL in the tabular setting. In RL, an agent interacts with an unknown environment modeled as a Markov decision process (MDP) in which there is a set of states $S$ with bounded size that describes all possible status of the environment. At a state $s \in S$, the agent interacts with the environment by taking an action $a$ from an action space $A$, receives an immediate reward and transits to the next state. The agent interacts with the environment episodically, where each episode consists of $H$ steps. The goal of the agent is to interact with the environment by executing a series a policies, so that after a certain number of interactions, sufficient information

is collected so that the agent could find a policy that performs nearly optimally. Replicability is a well-known challenge in RL, as RL algorithms are empirically observed to be unstable and sensitive to variations in training conditions. Our work aims to address this issue by introducing and analyzing list replicability in the PAC-RL framework. Moreover, by studying the replicability of RL from a theoretical point of view, we could build a clearer understanding of the instability issue of RL algorithms, and finally make progress towards enhancing the stability of empirical RL algorithms.

Theoretically, there are multiple ways to define the notion of list replicability in the context of RL. We may say an RL algorithm is $k$-list replicable, if there is a list $L$ of policies with cardinality $k$, so that the near-optimal policy found by the agent always lies in $L$ with high probability, where the list $L$ depends only on the unknown MDP instance. Under this definition of list replicability, it is only guaranteed that the returned policy lies in a list with small size: there is no limit on the sequence of policies executed by the agent (the trace). We call such RL algorithms to be *weakly $k$-list replicable*.

In certain applications, the above weak notion of list replicability may not suffice, and a more desirable notion of list replicability is to require both the returned policy and the trace (i.e., sequence of policies executed by the agent) lies in a list of small-size. This stronger notion of list replicability has been studied in multi-armed bandit (MAB) (Chen et al., 2025), and similar definition of replicability has been studied by Esfandiari et al. (2023) in MAB under $\rho$-replicability (Impagliazzo et al., 2022). In these works, it has been argued that limiting the number of possible traces (in terms of actions) of an MAB algorithm is more desirable in scenarios including clinical trials and social experiments. Therefore, the stronger notion of list replicability for RL mentioned above is a natural generalization of existing replicability definitions in MAB, and in this work, we say an RL algorithm to be *strongly $k$-list replicable* if such stronger notion (in terms of traces of policies) of list replicability holds.

The central theoretical question studied in this work is whether we can design list replicable PAC RL algorithms in the tabular setting. We give an affirmative answer to this question. We note that existing algorithms can potentially generate an exponentially large number of policies (and their execution traces) for the same problem instance, and hence, new techniques are needed to achieve our goal.

Interestingly, our theoretical investigation offers insights into addressing the instability commonly observed in practical RL algorithms. In particular, the new technical tools developed through our analysis can be integrated into existing RL frameworks to enhance their stability.

Below we give a more detailed description of our theoretical and empirical contributions.

**Theoretical Contributions.** Our first theoretical result is a black-box reduction which converts any PAC RL algorithm in the tabular setting to one that is weakly $k$-list replicable with $k = O(|S|^2|A|H^2)$. Here, $|S|$ is the number of states, $|A|$ is the number of actions and $H$ is the horizon length. Due to space limitation, the description of the reduction and its analysis is deferred to Appendix F.

**Theorem 1.1** (Informal version of Theorem F.1). *Given a RL algorithm $\mathbb{A}(\epsilon_0, \delta_0)$ that interacts with an unknown MDP and returns an $\epsilon_0$-optimal policy with probability at least $1-\delta_0$. There is a weakly $k$-list replicable algorithm (Algorithm 3) with $k = O(|S|^2|A|H^2)$ that makes $|S|H$ calls to $\mathbb{A}$ with $\epsilon_0 = \frac{\epsilon\delta}{\text{poly}(|S|,|A|,H)}$ and $\delta_0 = \delta/(8|S||H|)$. For any unknown MDP instance $M$, with probability at least $1 - \delta$, the algorithm returns an $\epsilon$-optimal policy $\pi \in \Pi(M)$, where $\Pi(M)$ is a list of policies that depends only on the underlying MDP $M$ with size $|\Pi(M)| = k$.*

Using PAC RL algorithms in the tabular setting (e.g. the algorithm by Kearns and Singh (1998a)) with sample complexity polynomial in $|S|$, $|A|$, $H$, $1/\epsilon_0$ and $\log(1/\delta_0)$) as $\mathbb{A}$, the final sample complexity of our weakly $k$-list replicable algorithm in Theorem 1.1 would be polynomial in $|S|$, $|A|$, $H$, $1/\epsilon$ and $1/\delta$. Compared to existing algorithms in the tabular setting, the sample complexity of our algorithm has much worse dependence on $1/\delta$ (polynomial dependence instead of logarithm dependence), which is common for algorithms with list replicability guarantees (Dixon et al., 2023). On the other hand, the list complexity $k$ of our algorithm has no dependence on $\delta$.

Our second result is a new RL algorithm that is strongly $k$-list replicable with $k = O(|S|^3|A|H^3)$.

**Theorem 1.2** (Informal version of Theorem 6.1). *There is a strongly $k$-list replicable algorithm (Algorithm 2) with $k = O(|S|^3|A|H^3)$, such that for any unknown MDP instance $M$, with probability at least $1 - \delta$, the algorithm returns an $\epsilon$-optimal policy, and the sequence of policies executed by the algorithm and the returned policy lies in a list with size $k$ that depends only on $M$. Moreover, the sample complexity of the algorithm is polynomial in $|S|$, $|A|$, $H$, $1/\epsilon$, $1/\delta$.*

Our second result shows that, perhaps surprisingly, even under the more stringent definition of list replicability, designing RL algorithm in the tabular setting with polynomial sample complexity and polynomial list complexity is still possible. The description of Algorithm 2 is given in Section 6.

Finally, we prove a hardness result on the list complexity of weakly replicable RL algorithm in the tabular setting, completing our new algorithms.

**Theorem 1.3** (Informal version of Theorem H.3). *For any weakly $k$-list replicable RL algorithm that returns an* $\epsilon$*-optimal policy with probability at least* $1 - \delta$*, we have* $k \geq \frac{|S||A|(H - \lceil \log_{|A|} |S| \rceil - 3)}{3}$ *as long as* $\epsilon \leq \frac{1}{2|S||A|H}$ *and* $\delta \leq \frac{1}{|S||A|H + 1}$.

Theorem 1.3 shows that the list complexity of any weakly $k$-list replicable algorithm is $\Omega(SAH)$, provided that its suboptimality and failure probability are both at most $O(1/(SAH))$. Theorem 1.3 is proved by a reduction from RL to the MAB and known list complexity lower bound for MAB (Chen et al., 2025). Its formal proof can be found in Appendix H.

**Empirical Contributions.** We further show that our robust planner (presented in Section 5), one of our new technical tools for establishing Theorem 1.1 and Theorem 1.2, can be incorporated into practical RL frameworks to enhance their stability. The empirical findings are presented in Section 7.

## 2 RELATED WORK

There is a long line of research dedicated to understanding the complexity of reinforcement learning by studying learning in a Markov Decision Process (MDP). One well-established setting is the *generative model*, which abstracts away exploration challenges by assuming access to a simulator that allows sampling from any state-action pair. A number of works (Kearns and Singh, 1998a; Pananjady and Wainwright, 2020; Kakade, 2003; Azar et al., 2013; Agarwal et al., 2020; Wainwright, 2019b;a; Sidford et al., 2018a;b; Li et al., 2024b;a; 2022; Even-Dar and Mansour, 2003; Shi et al., 2023; Beck and Srikant, 2012; Cui and Yang, 2021; Sidford et al., 2018b; Wainwright, 2019b; Azar et al., 2013; Agarwal et al., 2020) have established near-optimal sample complexity bounds for learning a policy in this regime. Specifically, to learn an $\epsilon$-optimal policy with high probability, the statistically optimal sample complexity is of the order $\text{poly}(|S|, |A|, H, 1/\epsilon)$, where $H$ denotes the horizon or the effective horizon of the environment. These algorithms generally fall into two categories: those that estimate the probability transition model and those that directly estimate the optimal $Q$-function. However, due to the inherent randomness in sampling, these approaches do not guarantee *list-replicable* policies—each independent execution of the algorithm may return a different policy, potentially leading to an exponentially large set of output policies.

In contrast, the online RL setting—where there is no access to a generative model—has seen significant progress over the past decades in optimizing sample complexity. Notable contributions include (Kearns and Singh, 1998b; Brafman and Tennenholtz, 2002; Kakade, 2003; Strehl et al., 2009; Auer, 2002; Strehl et al., 2006; Strehl and Littman, 2008; Kolter and Ng, 2009; Bartlett and Tewari, 2009; Jaksch et al., 2010; Szita and Szepesvári, 2010; Lattimore and Hutter, 2012; Osband et al., 2013; Dann and Brunskill, 2015; Agrawal and Jia, 2017; Dann et al., 2017; Jin et al., 2018; Efroni et al., 2019; Fruit et al., 2018; Zanette and Brunskill, 2019; Cai et al., 2019; Dong et al., 2019; Russo, 2019; Neu and Pike-Burke, 2020; Zhang et al., 2020; 2021; Tarbouriech et al., 2021; Xiong et al., 2022; Ménard et al., 2021; Wang et al., 2020; Li et al., 2021b;a; Domingues et al., 2021; Zhang et al., 2022). These works typically evaluate algorithmic performance within the regret framework, comparing the accumulated reward of an algorithm against that of an optimal policy. When adapted to the Probably Approximately Correct (PAC) RL framework, these results imply a sample complexity of $\text{poly}(|S|, |A|, H, 1/\epsilon)$ to learn an $\epsilon$-optimal policy with high probability. To achieve a balance between exploration and exploitation, the aforementioned algorithms generally follow a common iterative framework—maintaining a policy and refining it as new data is collected. For example, UCB-type algorithms (e.g., Jin et al. (2018)) maintain an approximate $Q$-function and leverage an upper-confidence bound to guide data collection. However, due to the iterative updates of these algorithms, they inherently fail to achieve polynomial complexity in either the strong or the weak notion of list replicability, as policies are likely to change at each iteration, and small stochastic error could have significant impact on the policies executed by the algorithm.

Recent studies have begun exploring *replicable reinforcement learning*. (Karbasi et al., 2024; Eaton et al., 2023) examined $\rho$-replicability, as defined in (Impagliazzo et al., 2022). Intuitively, $\rho$-replicability ensures that two executions of the same algorithm, when initialized with the same random seed, yield the same policy with probability at least $1 - \rho$. Meanwhile, $(k, \delta)$-weak list replicability requires that an algorithm consistently outputs a policy

from a fixed list of at most $k$ policies with probability at least $1 - \delta$. However, a $\rho$-replicable algorithm may still generate an exponentially large number of distinct policies, as each seed may correspond to a different output policy. Thus, such algorithms may still suffer from exponential weak (or strong) list complexity. (Esfandiari et al., 2023) further studied the Multi-Armed Bandit (MAB) problem under $\rho$-replicability, where two independent executions of a $\rho$-replicable MAB algorithm, sharing the same random string, must follow the same sequence of actions with probability at least $1 - \rho$.

Beyond the above frameworks, there is a growing body of work studying replicability and closely related stability notions in classical learning theory. Chase et al. (2023) introduce global stability, a seed-independent variant of replicability, and clarify its relationship to classical notions of algorithmic stability. Bun et al. (2023) further show that several such stability notions are essentially equivalent and develop general "stability booster" constructions that yield replicable algorithms from non-replicable ones, revealing tight connections to differential privacy and adaptive data analysis. More recently, Kalavasis et al. (2024) investigate the computational landscape of replicable learning, identifying settings where efficient replicable algorithms provably do not exist, while Blondal et al. (2025) study stability and list replicability in the agnostic PAC setting and prove sharp trade-offs between excess risk, stability, and list size. Our results are complementary to this line of work: we focus on control problems rather than supervised learning, and we explicitly track the list complexity of both output policies and execution traces in tabular RL, showing that nontrivial list-replicability guarantees are achievable with polynomial sample complexity.

In the online learning setting, the only known work addressing list replicability is by Chen et al. (2025), who studied the concept in the context of Multi-Armed Bandits (MAB). The authors define an MAB algorithm as $(k, \delta)$-list replicable if, for any MAB instance, there exists a list of at most $k$ action traces such that the algorithm selects one of these traces with probability at least $1 - \delta$. Our definition of *strong list replicability* for RL naturally extends this notion to RL. However, due to the long-horizon nature of RL, achieving list replicability in RL presents significantly greater challenges.

Concurrent to our work, Hopkins et al. (2025) study sample-efficient replicable RL in the tabular setting. Their algorithms also stably identify a set of ignorable states and then perform backward induction using data collected from the remaining states, which is conceptually similar to our use of robust planning on non-ignorable states. However, they focus on fully replicable algorithms (a single policy that reappears with high probability), without explicitly analyzing the induced list size, whereas we design algorithms with explicit $(k, \delta)$-list-replicability guarantees while retaining near-optimal sample complexity.

## 3 PRELIMINARIES

**Notations.** For a positive integer $N$, we use $[N]$ to denote $\{0, 1, \ldots, N - 1\}$. For a condition $\mathcal{E}$, we use $\mathbb{1}[\mathcal{E}]$ to denote the indicator function, i.e., $\mathbb{1}[\mathcal{E}] = 1$ if $\mathcal{E}$ holds and $\mathbb{1}[\mathcal{E}] = 0$ otherwise. For a real number $x$ and $\epsilon \geq 0$, we use $\mathrm{Ball}(x, \epsilon)$ to denote $[x - \epsilon, x + \epsilon]$. For two real numbers $a < b$, we use $\mathrm{Unif}(a, b)$ to denote the uniform distribution over $(a, b)$.

**Markov Decision Process.** Let $M = (S, A, P, R, H, s_0)$ be a Markov Decision Process (MDP). Here, $S$ is the state space, and $A = \{1, 2, \ldots, |A|\}$ is the action space. $P = (P_h)_{h \in [H]}$, where for each $h \in [H]$, $P_h : S \times A \to \Delta(S)$ is the transition model at level $h$ which maps a state-action pair to a distribution over states. $R = (R_h)_{h \in [H]}$, where for each $h \in [H]$, $R_h : S \times A \to [0, 1]$ is the deterministic reward function at level $h$. $H \in \mathbb{Z}^+$ is the horizon length, and $s_0 \in S$ is the initial state. We further assume that the reward functions $R = (R_h)_{h \in [H]}$ are known. [1]

A (non-stationary) policy $\pi$ chooses an action $a \in A$ based on the current state $s \in S$ and the time step $h \in [H]$. Formally, $\pi = \{\pi_h\}_{h=0}^{H-1}$ where for each $h \in [H]$, $\pi_h : S \to A$ maps a given state to an action. The policy $\pi$ induces a (random) trajectory $s_0, a_0, r_0, s_1, a_1, r_1, \ldots, s_{H-1}, a_{H-1}, r_{H-1}$, where for each $h \in [H]$, $a_h = \pi_h(s_h)$, $r_h = R_h(s_h, a_h)$ and $s_{h+1} \sim P_h(s_h, a_h)$ when $h < H - 1$.

**Interacting with the MDP.** In RL, an agent interacts with an unknown MDP. In the online setting, in each episode, the agent decides a policy $\pi$, observes the induced trajectory, and proceeds to the next episode. In the generative

---

[1] For simplicity, we assume deterministic rewards and the initial state, and known reward function. Our algorithms can be easily extended to handle stochastic rewards and initial state, and unknown rewards distributions.

model setting, in each round, the agent is allowed to choose a state-action pair $(s, a) \in S \times A$ and a level $h \in [H]$, and receives a sample drawn from $P_h(s, a)$ as feedback.

**Value Functions and $Q$-Functions.** For an MDP $M$, given a policy $\pi$, a level $h \in [H]$ and $(s, a) \in S \times A$, the $Q$-function is defined as $Q_{h,M}^\pi(s, a) = \mathbb{E}\left[\sum_{h'=h}^{H-1} r_{h'} \mid s_h = s, a_h = a, M, \pi\right]$, and the value function is defined as $V_{h,M}^\pi(s) = \mathbb{E}\left[\sum_{h'=h}^{H-1} r_{h'} \mid s_h = s, M, \pi\right]$. We denote $Q_{h,M}^*(s, a) = Q_{h,M}^{\pi^*}(s, a)$ and $V_{h,M}^*(s) = V_{h,M}^{\pi^*}(s)$ where $\pi^*$ is the optimal policy. We also write $V_M^* = V_{0,M}^*(s_0)$ and $V_M^\pi = V_{0,M}^\pi(s_0)$ for a policy $\pi$. We may omit $M$ from the subscript of value functions and $Q$-functions when $M$ is clear from the context (e.g., when $M$ is the underlying MDP that the agent interacts with). We say a policy $\pi$ to be $\epsilon$-optimal if $V^\pi \geq V^* - \epsilon$.

The goal of the agent is to return a near-optimal policy $\pi$ after interacting with the unknown MDP $M$ by executing a sequence of policies (or by querying the transition model in the generative model).

**Further Notations.** For an MDP $M$, define the occupancy function $d_M^\pi(s, h) = \Pr[s_h = s \mid M, \pi]$ and $d_M^*(s, h) = \max_\pi \Pr[s_h = s \mid M, \pi]$. We may omit $M$ from the subscript of $d_M^\pi(s, h)$ and $d_M^*(s, h)$ when $M$ is clear from the context. For an MDP $M$, we write

$$\text{Gap}_M = \{V_{h,M}^*(s) - Q_{h,M}^*(s, a) \mid (s, a) \in S \times A, h \in [H]\}. \tag{1}$$

Two MDPs $M_1$ and $M_2$ are said to be $\epsilon$-*related* if $M_1$ and $M_2$ share the same state space $S$, action space $A$, reward function and initial state, and for all $(s, a) \in S \times A$ and $h \in [H - 1]$,

$$\sum_{s' \in S} \left| P_h^{M_1}(s' \mid s, a) - P_h^{M_2}(s' \mid s, a) \right| \leq \epsilon \tag{2}$$

where $P_h^{M_1}$ is the transition model of $M_1$ at level $h$ and $P_h^{M_2}$ is that of $M_2$ at the same level.

**List Replicability in RL.** We now formally define the notion of list replicability of RL algorithms in the online setting. For an RL algorithm $\mathbb{A}$, we say $\mathbb{A}$ to be *weakly $(k, \delta)$-list replicable*, if for any MDP instance $M$, there is a list of policies $\Pi(M)$ with cardinality at most $k$, so that $\Pr[\pi \in \Pi(M)] \geq 1 - \delta$, where $\pi$ is the (supposedly) near-optimal policy returned by $\mathbb{A}$ when interacting with $M$.

For an RL algorithm $\mathbb{A}$, we say $\mathbb{A}$ to be *strongly $(k, \delta)$-list replicable*, if for any MDP instance $M$, there is a list $\text{Trace}(M)$ with cardinality at most $k$, so that $\Pr[((\pi_0, \pi_1, \ldots), \pi) \in \text{Trace}(M)] \geq 1 - \delta$, where $(\pi_0, \pi_1, \ldots)$ is the (random) sequence of policies executed by $\mathbb{A}$ when interacting with $M$ and $\pi$ is the (supposedly) near-optimal policy returned by $\mathbb{A}$ when interacting with $M$.

## 4 Overview of New Techniques

In this section, we discuss the techniques for establishing Theorem 1.1 and Theorem 1.2.

**The Robust Planner.** To motivate our new approach, consider the following simple MDP instance for which most existing RL algorithms would fail to achieve polynomial list complexity. There is a state $s_h$ at each level $h \in [H]$, and the action space is $\{a_1, a_2\}$. At level $h$, if $a_i$ is chosen, $s_h$ transitions to $s_{h+1}$ with an unknown probability $p_{h,i}$, otherwise $s_h$ transitions to an absorbing state. The agent receives a reward of 1 at the last level. For this instance, if $|p_{h,1} - p_{h,2}| = \exp(-H)$, then for all $h \in [H]$, no RL algorithm could differentiate $p_{h,1}$ and $p_{h,2}$ unless we draw an exponential number of samples. Therefore, if the RL algorithm simply returns a policy by maximizing the estimated optimal $Q$-values for each $s_h$, then we would choose either $a_1$ or $a_2$, and hence, there could be $2^H$ different policies returned by the algorithm. As most existing RL algorithms choose actions by maximizing the estimated $Q$-values, they would all fail to achieve polynomial list complexity even for this simple instance. This also explains why existing RL algorithms tend to be unstable and sensitive to noise.

To better understand our new approach, let us first consider the simpler generative model setting. Standard analysis shows that by taking sufficient samples for all $(s, a) \in S \times A$ and $h \in [H]$ to build the empirical model $\hat{M}$, we would have $|\hat{Q}_h(s, a) - Q_{h,M}^*(s, a)| \leq \epsilon_0$ for all $(s, a) \in S \times A$ and $h \in [H]$. Here, $\hat{Q}_h(s, a) = Q_{h,\hat{M}}^*(s, a)$ is the estimated $Q$-value, and $\epsilon_0$ is a statistical error that can be made arbitrarily small by drawing more samples. Now, for a given state $s$ and level $h$, instead of choosing an action by maximizing $\hat{Q}_h(s, a)$, we go through all actions in a fixed order $1, 2, \ldots |A|$, and choose the lexicographically first action $a$ so that $\hat{Q}_h(s, a) \geq \max_a \hat{Q}_h(s, a) - r_{\text{action}}$, where $r_{\text{action}}$ is a tolerance parameter drawn from the uniform distribution.

Now we show that our new approach achieves small list complexity. The main observation is the that, for a fixed tolerance parameter $r_{\text{action}}$, if difference between $r_{\text{action}}$ and $\text{Gap}_h(s,a) = V_h^*(s) - Q_h^*(s,a)$ satisfies $r_{\text{action}} \notin \text{Ball}(\text{Gap}_h(s,a), 2\epsilon_0)$ for all $(s,a) \in S \times A$ and $h \in [H]$, then the returned policy will always be the same regardless of the estimation errors. To see this, for an action $a$, if $r_{\text{action}} \notin \text{Ball}(\text{Gap}_h(s,a), 2\epsilon_0)$, then whether $\hat{Q}_h(s,a) \geq \hat{V}_h(s) - r_{\text{action}}$ or not will always be the same regardless of the stochastic noise as long as $|\hat{Q}_h(s,a) - Q_h^*(s,a)| \leq \epsilon_0$. Since we always choose the lexicographically first action $a$ satisfying $\hat{Q}_h(s,a) \geq \hat{V}_h(s) - r_{\text{action}}$, the action chosen for $s$ will always be the same. Equivalently, by defining $\text{Bad}_{\text{action}} = \bigcup_{h,s,a} \text{Ball}(\text{Gap}_h(s,a), 2\epsilon_0)$, the returned policy will always be the same so long as $r_{\text{action}} \notin \text{Bad}_{\text{action}}$. By drawing $r_{\text{action}}$ from the uniform distribution over $(0, 2HSA\epsilon_0/\delta)$, we would have $\Pr[r_{\text{action}} \notin \text{Bad}_{\text{action}}] \geq 1 - \delta$. Moreover, for two tolerance parameters $r_{\text{action}}^1, r_{\text{action}}^2 \notin \text{Bad}_{\text{action}}$, if for all $(s,a) \in S \times A$ and $h \in [H]$ we have either $r_{\text{action}}^1 < r_{\text{action}}^2 < \text{Gap}_h(s,a)$ or $\text{Gap}_h(s,a) < r_{\text{action}}^1 < r_{\text{action}}^2$, then the returned policy will also be the same no matter $r_{\text{action}} = r_{\text{action}}^1$ or $r_{\text{action}} = r_{\text{action}}^2$. Since there are at most $|S||A|H + 1$ different values for $\text{Gap}_h(s,a)$ for the underlying MDP $M$, there could be at most $|S||A|H + 1$ different policies returned by our algorithm as long as $r_{\text{action}} \notin \text{Bad}_{\text{action}}$. Finally, the suboptimality of the returned policy can be easily shown to be $O(H \cdot r_{\text{action}})$.

**Weakly $k$-list Replicable Algorithm in the Online Setting.** Our algorithm in the online setting with weakly $k$-list replicable guarantee is based on building a policy cover (Jin et al., 2020). Given a black-box RL algorithm, for each $(s,h) \in S \times [H]$, we set the reward function to be $R_{h'}^{s,h}(s',a) = \mathbb{1}[s' = s, h = h']$, invoke the black-box RL algorithm with the modified reward function, and set the returned policy to be $\hat{\pi}^{s,h}$. Since $\hat{\pi}^{s,h}$ is an $\epsilon$-optimal policy, we have $d^{\hat{\pi}^{s,h}}(s,h) \geq d^*(s,h) - \epsilon$. At this point, one could use $\hat{\pi}^{s,h}$ to collect samples and estimate the transition model $P_h(s,a)$, and return a policy by invoking the robust planning algorithm mentioned above. The issue is that there could be some $(s,h) \in S \times [H]$ unreachable for any policy $\pi$, i.e., $d^*(s,h)$ is small. For those $(s,h)$, it is impossible to estimate the transition model $P_h(s,a)$ accurately. On the other hand, our robust planning algorithm requires $|\hat{Q}_h(s,a) - Q_h^*(s,a)| \leq \epsilon_0$ for all $(s,a) \in S \times A$ and $h \in [H]$.

To tackle the above issue, we use an additional truncation step to remove unreachable states. For each $(s,h) \in S \times [H]$, we first use the roll-in policy $\hat{\pi}^{s,h}$ to estimate the probability of reaching $s$ at level $h$. If the estimated probability is small, it would be clear that $d^*(s,h)$ is also small as $d^{\hat{\pi}^{s,h}}(s,h) \geq d^*(s,h) - \epsilon$, so that $(s,h)$ can be removed from the MDP. On the other hand, implementing the above truncation step naïvely would significantly increase the list complexity of our algorithm as the returned policy depends on the set of $(s,h) \in S \times [H]$ being removed. Here, we use an approach similar to the robust planning algorithm mentioned earlier. We use a randomly chosen reaching probability truncation threshold $r_{\text{trunc}}$ drawn from the uniform distribution, and for each $(s,h) \in S \times [H]$, we declare $(s,h)$ to be unreachable iff the estimated reaching probability (using $\hat{\pi}^{s,h}$) does not exceed $r_{\text{trunc}}$. Similar to the analysis in the robust planning algorithm, for a reaching probability truncation threshold $r_{\text{trunc}}$, the set of $(s,h)$ being removed would be the same as long as the difference $r_{\text{trunc}}$ and $d^*(s,h)$ is large enough for all $(s,h) \in S \times [H]$. Moreover, two reaching probability truncation thresholds $r_{\text{trunc}}^1$ and $r_{\text{trunc}}^2$ will result in the same set of $(s,h)$ being removed if for all $(s,h) \in S \times [H]$ we have either $r_{\text{trunc}}^1 < r_{\text{trunc}}^2 < d^*(s,h)$ or $d^*(s,h) < r_{\text{trunc}}^1 < r_{\text{trunc}}^2$. Therefore, the total number of different sets of $(s,h)$ being removed is at most $O(|S|H)$.

**Strongly $k$-list Replicable Algorithm in the Online Setting.** Unlike the case of weak list replicability where we can use a black-box RL algorithm to determine the set of unreachable states independently at each level, for strongly list replicable RL, such a method would not suffice due to the potentially large list complexity of the black-box algorithm. Our algorithm with strongly $k$-list replicable guarantees employs a level-by-level approach: for each level $h$, we find a policy $\hat{\pi}^{s,h}$ to reach $s$ at level $h$ for each $s \in S$, build an empirical transition model for level $h$, and proceed to the next level $h + 1$. To ensure list replicability guarantees, for each $(s,h) \in S \times [H]$, we use the same robust planning algorithm to find $\hat{\pi}^{s,h}$. As mentioned ealier, for any level $h$, there could be unreachable states, and the estimated transition model for those states could be inaccurate. To handle this, for each level $h$, based on the estimated transition models of previous levels, we test the reachability of all states in level $h$ by using the same mechanism as in our previous algorithm, and remove those unreachable states by transitioning them to an absorbing state $s_{\text{absorb}}$ in the estimated model.

Although the algorithm is conceptually straightforward given existing components, the analysis is not. For the new algorithm, states removed at level $h$ have significant impact on the reaching probabilities of later levels, which also affect the planned roll-in policies of later levels. Such dependency issue must be handled carefully to have a polynomial list complexity. To handle this, we prove several structural properties of reaching probabilities

in truncated MDPs in Section D. For the time being we assume that in our algorithm, for each level $h$, instead of using estimated reaching probabilities, the algorithm has access to the true reaching probabilities, and those reaching probabilities have taken unreachable states removed in previous levels into consideration. I.e., for a reaching probability truncation threshold $r_{\text{trunc}}$, we first remove all states in the first level that cannot be reached with probability higher than $r_{\text{trunc}}$, recalculate the reaching probability in the second level after truncating the first level, remove unreachable states in the second level (again using the same threshold $r_{\text{trunc}}$), an so on. We use $U_h(r_{\text{trunc}})$ to denote the set of states removed in level $h$ during the above process, and see Definition D.1 for a formal definition. We show that for different $r_{\text{trunc}}$, $U_h(r_{\text{trunc}})$ could not be an arbitrary subset of the state space, and the main observation is that $U_h(r_{\text{trunc}})$ satisfies certain monotonicity property, i.e., given $r_1, r_2 \in [0, 1]$, if $r_1 < r_2$ then we have $U_h(r_1) \subseteq U_h(r_2)$. This observation can be proved by induction on $h$, and see Lemma D.2 and its proof for more details.

As an implication, if we write $U(r) = (U_0(r), U_1(r), \ldots, U_{H-1}(r))$, then there could be at most $|S|H + 1$ different choices of $U(r)$ for all $r \in [0, 1]$ by the pigeonhole principle. Therefore, after fixing the reaching probability truncation threshold, the set of states that will be removed at each level will be fixed, and for all different reaching probability truncation thresholds, there could be at most $|S|H + 1$ different ways to remove states even if we consider all levels simultaneously.

The above discussion heavily relies on the true reaching probabilities. As another implication of the monotonicity property, there is a critical reaching probability threshold $\text{Crit}(s, h)$ for each $(s, h)$, and $s \in U_h(r)$ iff $r \leq \text{Crit}(s, h)$ (cf. Corollary D.5). Therefore, for a fixed reaching probability truncation threshold $r_{\text{trunc}}$, as long as the distance between $r_{\text{trunc}}$ and $\text{Crit}(s, h)$ is much larger than the statistical errors, the set of states being removed will still be the same as $U(r_{\text{trunc}})$ even with statistical errors. In particular, if we draw $r_{\text{trunc}}$ from a uniform distribution as in previous algorithms, with high probability $r_{\text{trunc}}$ and $\text{Crit}(s, h)$ would have a large distance for all $(s, h) \in S \times [H]$, in which case the set of removed states will be one of those $|S|H + 1$ different choices of $U(r)$.

## 5   ROBUST PLANNING

In this section, we formally describe our robust planning algorithm (Algorithm 1). Here, it is assumed that there is an unknown underlying MDP $M$. Algorithm 1 receives an MDP $\hat{M}$ and a tolerance parameter $r_{\text{action}}$ as input, and it is assumed that $M$ and $\hat{M}$ are $\epsilon_0$-related (see (2) for the definition). In Algorithm 1, for each $(s, h) \in S \times [H]$, we go through all actions in the action space $A$ in a fixed order $1, 2, \ldots, |A|$, and choose the first action $a$ so that $Q^*_{h,\hat{M}}(s, a) \geq V^*_{h,\hat{M}}(s) - r_{\text{action}}$.

---

**Algorithm 1** Robust Planning

1: **Input:** MDP $\hat{M}$, tolerance parameter $r_{\text{action}}$.
2: **Output:** near-optimal policy $\hat{\pi}$
3: Define $\hat{\pi}_h(s) = \min\{a \in A \mid Q^*_{h,\hat{M}}(s, a) \geq V^*_{h,\hat{M}} - r_{\text{action}}\}$ for each $(s, h) \in S \times [H]$
4: **return** $\hat{\pi}$

---

Our first lemma characterizes the suboptimality of the returned policy. Its formal proof is based on the performance difference lemma (Kakade and Langford, 2002) and can be found in Section C.

**Lemma 5.1.** *Suppose $M$ and $\hat{M}$ are $\epsilon_0$-related. The policy $\hat{\pi}$ returned by Algorithm 1 satisfies $V^{\hat{\pi}}_M \geq V^*_M - 2H^2\epsilon_0 - r_{\text{action}}H$.*

Our second lemma shows that if $r_{\text{action}}$ is chosen to be far from $\text{Gap}_{h,M}(s, a) = V^*_{h,M}(s) - Q^*_{h,M}(s, a)$ for all $(s, a) \in S \times A$ and $h \in [H]$, then the returned policy $\hat{\pi}$ depends only on $M$ and $r_{\text{action}}$. Moreover, for two choices $r^1_{\text{action}}$ and $r^2_{\text{action}}$ of the tolerance parameter $r_{\text{action}}$, the returned policy will be the same if $r^1_{\text{action}}$ and $r^2_{\text{action}}$ always lie on the same side of $\text{Gap}_{h,M}(s, a)$ for all $(s, a) \in S \times A$ and $h \in [H]$. Full proof of the lemma and corollary can be found in Section C.

**Lemma 5.2.** *Suppose $M$ and $\hat{M}$ are $\epsilon_0$-related. For two tolerance parameters $r^1_{\text{action}}$ and $r^2_{\text{action}}$, if*

- $r^1_{\text{action}}, r^2_{\text{action}} \notin \bigcup_{g \in \text{Gap}_M} \text{Ball}(g, 2H^2\epsilon_0)$ *where $\text{Gap}_M$ is as defined in (1);*

- *for any $g \in \text{Gap}_M$, either $g < r^1_{\text{action}} < r^2_{\text{action}}$ or $r^1_{\text{action}} < r^2_{\text{action}} < g$,*

*then the returned policy $\hat{\pi}$ depends only on $M$ and $r_{\mathrm{action}}$, and for both tolerance parameters $r_{\mathrm{action}}^1$ and $r_{\mathrm{action}}^2$, the returned policy $\hat{\pi}$ would be identical for the same underlying MDP $M$.*

As a corollary of Lemma 5.1 and Lemma 5.2, we show how to design a list-replicable RL algorithm in the generative model setting by invoking Algorithm 1 with a randomly chosen parameter $r_{\mathrm{action}}$.

**Corollary 5.3.** *In the generative model setting, there is an algorithm with sample complexity polynomial in $|S|$, $|A|$, $1/\epsilon$ and $1/\delta$, such that with probability at least $1 - \delta$, the returned policy is $\epsilon$-optimal and always lies in a list $\Pi(M)$ where $\Pi(M)$ is a list of policies that depend only on the unknown underlying MDP $M$ with $|\Pi(M)| = O(|S||A|H)$.*

# 6 STRONGLY $k$-LIST REPLICABLE RL ALGORITHM

In this section, we present our strongly $k$-list replicable algorithm (Algorithm 2). As mentioned in Section 4, Algorithm 2 employs a layer-by-layer approach. In Algorithm 2, for each $h \in [H]$, $\hat{U}_h$ is the set of states estimated to be unreachable at level $h$, and we initialize $\hat{U}_0 = S \setminus \{s_0\}$ where $s_0$ is the fixed initial state. For each iteration $h$, we assume that $\hat{U}_h$ has been calculated, and for all $s \notin \hat{U}_h$, we assume that a roll-in policy $\hat{\pi}^{s,h}$ has been determined (except for $h = 0$, since any policy would suffice for reaching the initial state). Now we describe how to proceed to the next iteration $h + 1$.

For each $s \notin \hat{U}_h$ and $a \in A$, we build a policy $\hat{\pi}^{s,h,a}$ based on $\hat{\pi}^{s,h}$, and execute $\hat{\pi}^{s,h,a}$ to collect samples and calculate $\hat{P}_h(s, a)$ as our estimate of $P_h(s, a)$. Based on $\{\hat{P}_{h'}(s, a)\}_{h' \leq h}$ and $\{\hat{U}_{h'}\}_{h' \leq h}$, we build an MDP $\tilde{M}^{h+1}$ (cf. (3)). For each $h' \leq h$ and $s \in S$, if $s \notin \hat{U}_{h'}$ the transition model of $s$ in $\tilde{M}^{h+1}$ at level $h'$ would be the same as $\hat{P}_{h'}(s, \cdot)$. If $s \in \hat{U}_{h'}$, we always transit $s$ to an absorbing state $s_{\mathrm{absorb}}$ in $\tilde{M}^{h+1}$ at level $h'$. Given $\tilde{M}^{h+1}$, for each $s \in S$, we calculate $d^*_{\tilde{M}^{h+1}}(s, h + 1)$ as our estimate of $d^*(s, h + 1)$, and we include $s$ in $\hat{U}_{h+1}$ if $d^*_{\tilde{M}^{h+1}}(s, h + 1) \leq r_{\mathrm{trunc}}$. Here, $r_{\mathrm{trunc}}$ is a reaching probability truncation threshold drawn from the uniform distribution. For each $s \notin \hat{U}_{h+1}$, we further find a roll-in policy $\hat{\pi}^{s,h+1}$ by invoking Algorithm 1 on $\tilde{M}^{h+1}$ with a modified reward function $R_{h'}^{s,h+1}(s', a) = \mathbb{1}[h' = h + 1, s' = s]$ and tolerance parameter $r_{\mathrm{action}}$, where $r_{\mathrm{action}}$ is also drawn from the uniform distribution. After finishing all these steps, we proceed to the next iteration.

Finally, after finishing all iterations, we invoke Algorithm 1 again with MDP $\tilde{M}^{H-1}$ and the same tolerance parameter $r_{\mathrm{action}}$, and return the output of Algorithm 1 as the final output. The formal guarantee of Algorithm 2 is stated in the following theorem. Its proof can be found in Section E.

**Theorem 6.1.** *For any unknown MDP instance $M$, there is a list $\mathrm{Trace}(M)$ with size at most $k = O(|S|^3|A|H^3)$ that depends only on $M$, and with probability at least $1 - \delta$, the policy $\pi$ returned by Algorithm 2 is $\epsilon$-optimal, and $((\pi_0, \pi_1, \ldots), \pi) \in \mathrm{Trace}(M)$, where $(\pi_0, \pi_1, \ldots)$ is the sequence of policies executed by Algorithm 2 when interacting with $M$.*

# 7 EXPERIMENTS

In this section, we show that our new planning strategy can be incorporated into empirical RL frameworks to enhance their stability. In our experiments, we use three different environments in Gymnasium (Towers et al., 2024): Cartpole-v1, Acrobot-v1 and MountainCar-v0. For each environment, we use a different empirical RL algorithms: DQN (Mnih et al., 2015), Double DQN (Van Hasselt et al., 2016) and tabular Q-learning based on discretization. We combine our robust planner in Section 5 with the above empirical RL algorithm by replacing the planning algorithm with Algorithm 1. Unlike our theoretical analysis, we treat the tolerance parameter $r_{\mathrm{action}}$ as a hyperparameter and experiment with different choices of $r_{\mathrm{action}}$. Note that when $r_{\mathrm{action}} = 0$, Algorithm 1 is equivalent to picking actions that maximize the estimated $Q$-value as in the original empirical RL algorithms (DQN, Double DQN and tabular Q-learning). The results are presented in Figure 1. Here we repeat each experiment by 25 times. The $x$-axis is the number of training episodes, the $y$-axis is the average award of the trained policy, $\pm$ standard deviation across 25 runs. More details can be found in Appendix I.

Our experiments show that by choosing a larger tolerance parameter $r_{\mathrm{action}}$, the performance of the algorithm becomes more stable at the cost of worse accuracy. Therefore, by choosing a suitable hyperparameter $r_{\mathrm{action}}$, we could achieve a balance between stability and accuracy.

We further use our new planning strategy in more challenging Atari environments, such as NameThisGame. Using the BTR algorithm ( (Clark et al., 2024)) as the baseline, we find that simply augmenting it with the robust planner

---

**Algorithm 2** Strongly $k$-list Replicable RL Algorithm

---

1: **Input:** error tolerance $\epsilon$, failure probability $\delta$

2: **Output:** near-optimal policy $\pi$

3: Initialize $C_1 = \frac{8AS^2H^2}{\delta}$, $\epsilon_0 = \frac{\epsilon\delta}{1440S^3H^7A}$, $\epsilon_1 = 5C_1H^2\epsilon_0$, $\eta_0 = 3\epsilon_1H$, $W = \frac{S^2\log(8HS^2A/\delta)}{\epsilon_0^2\eta_0}$

4: Generate random numbers $r_{\text{action}} \sim \text{Unif}(\epsilon_1, 2\epsilon_1)$, $r_{\text{trunc}} \sim \text{Unif}(3\eta_0, 6\eta_0)$

5: Initialize $\hat{U}_0 = S \setminus \{s_0\}$

6: **for** $h \in [H-1]$ **do**

7:     **for** $(s,a) \in (S \setminus \hat{U}_h) \times A$ **do**

8:         Define policy $\hat{\pi}^{s,h,a}$, where for each $h' \in [H]$, $\hat{\pi}_{h'}^{s,h,a}(s') = \begin{cases} a & h' \geq h \\ \hat{\pi}_{h'}^{s,h}(s') & h' < h \end{cases}$

9:         Collect $W$ trajectories $\{(s_0^{(w)}, a_0^{(w)}, \ldots, s_{H-1}^{(w)}, a_{H-1}^{(w)})\}_{w=1}^W$ by executing $\hat{\pi}^{s,h,a}$ for $W$ times

10:         For each $s' \in S$, set $\hat{P}_h(s' \mid s,a) = \frac{\sum_{w=1}^W \mathbb{1}[(s_h^{(w)}, a_h^{(w)}, s_{h+1}^{(w)}) = (s,a,s')]}{\sum_{w=1}^W \mathbb{1}[(s_h^{(w)}, a_h^{(w)}) = (s,a)]}$

11:     **end for**

12:     Define MDP $\tilde{M}^{h+1} = (S \cup \{s_{\text{absorb}}\}, A, \tilde{P}^{h+1}, R, H, s_0)$, where for each $h' \in [H]$,

$$\tilde{P}_{h'}^{h+1}(s' \mid s,a) = \begin{cases} \hat{P}_{h'}(s' \mid s,a) & h' \leq h, s \notin \hat{U}_{h'} \cup \{s_{\text{absorb}}\} \text{ and } s' \neq s_{\text{absorb}} \\ 0 & h' \leq h, s \notin \hat{U}_{h'} \cup \{s_{\text{absorb}}\} \text{ and } s' = s_{\text{absorb}} \\ \mathbb{1}[s' = s_{\text{absorb}}] & h' > h \text{ or } s \in \hat{U}_{h'} \cup \{s_{\text{absorb}}\} \end{cases} \tag{3}$$

13:     Set $\hat{U}_{h+1} = \{s \in S \mid d_{\tilde{M}^{h+1}}^*(s, h+1) \leq r_{\text{trunc}}\}$

14:     **for** $s \in S \setminus \hat{U}_{h+1}$ **do**

15:         Define MDP $\tilde{M}^{s,h+1} = (S \cup \{s_{\text{absorb}}\}, A, \tilde{P}^{h+1}, R^{s,h+1}, H, s_0)$, where $\tilde{P}^{h+1}$ is as defined in (3) and $R_{h'}^{s,h+1}(s', a) = \mathbb{1}[h' = h+1, s' = s]$

16:         Invoke Algorithm 1 with input $\tilde{M}^{s,h+1}$ and $r_{\text{action}}$, and set $\hat{\pi}^{s,h+1}$ to be the returned policy

17:     **end for**

18: **end for**

19: Invoke Algorithm 1 with input $\tilde{M}^{H-1}$ and $r_{\text{action}}$, and set $\pi$ to be the returned policy

20: **return** $\pi$

---

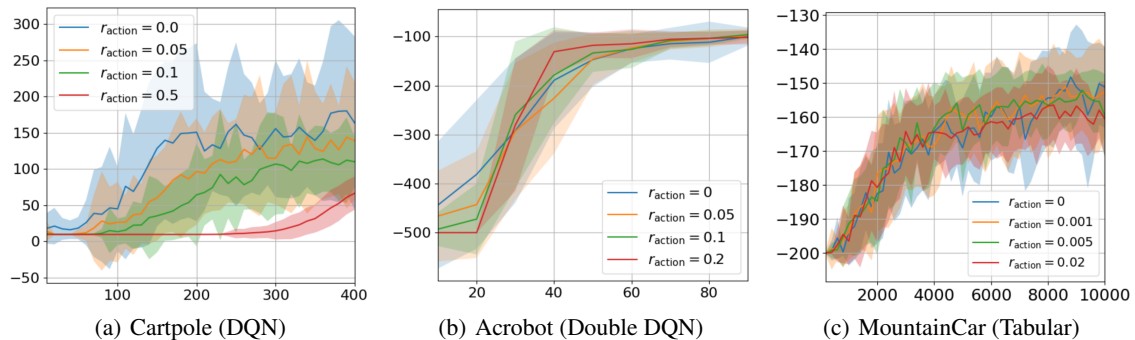

(a) Cartpole (DQN)          (b) Acrobot (Double DQN)          (c) MountainCar (Tabular)

Figure 1: Different threhold

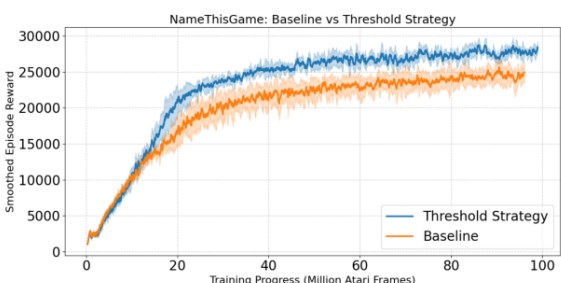

Figure 2: Namethisgame ( BTR )

leads to a substantial improvement. In particular, the performance on NameThisGame increases by more than 10%, demonstrating that even this lightweight modification can yield significant gains in practice. The results are presented in Figure 9.

## 8 CONCLUSION

We conclude the paper by several interesting directions for future work. Theoretically, our results show that even under a seemingly stringent definition of replicability (strong list replicability), efficient RL is still possible in the tabular setting. An interesting future direction is to develop replicable RL algorithms under more practical definitions of replicability and/or with function approximation schemes using our new techniques. Empirically, it would be interesting to incorporate our robust planner with other practical RL algorithms to see whether their stability could be improved. Currently, our robust planner can only work with discrete action spaces, and it remains to develop new techniques to overcome this limitation.

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

## A    OVERLINE OF THE PROOFS

### A.1    DEFINATIONS

**PAC RL and sample complexity.**    We work in the standard Probably Approximately Correct (PAC) framework for episodic reinforcement learning. Let $M$ be a finite-horizon Markov decision process with state space $\mathcal{S}$, action space $\mathcal{A}$, horizon $H$, and a fixed initial-state distribution. Consider a (possibly randomized) learning algorithm Alg that interacts with $M$ (either via a generative model or by running episodes). Denote by $\pi_{\mathsf{Alg},M}$ the final policy output by Alg, and by $V_M^\pi$ the value of a policy $\pi$ in $M$.

**PAC RL.** Given accuracy $\epsilon > 0$ and confidence $\delta \in (0,1)$, we say that Alg is an $(\epsilon, \delta)$-PAC RL algorithm for a class of MDPs $\mathcal{M}$ if, for every $M \in \mathcal{M}$,

$$\mathbb{P}\big(V_M^{\pi_{\mathsf{Alg},M}} \geq V_M^\star - \epsilon\big) \geq 1 - \delta,$$

where the probability is over all randomness of Alg and the environment, and $V_M^\star$ is the value of an optimal policy in $M$.

**Sample complexity.** The *sample complexity* of Alg in this PAC RL setting is the worst-case (over $M \in \mathcal{M}$) expected number of environment samples used by Alg before it outputs its final policy and stops. In the episodic setting this is the total number of state–action–next-state transitions (equivalently, time steps across all episodes); in the generative-model setting this is the total number of generative queries. We are interested in algorithms whose sample complexity is polynomial in $|\mathcal{S}|$, $|\mathcal{A}|$, $H$, $1/\epsilon$, and $1/\delta$.

### A.2    APPENDIX ROADMAP

We begin with a concise guide to the appendix materials.

Appendix A provides an outline of the appendix, high-level proof blueprints for strong and weak list replicability, and several schematic figures for intuition.

Appendices B and I contain experiments:

- Appendix B presents a direct toy experiment in the generative model with $|A| = 2$ that compares the robust planner with the greedy planner by measuring the size of returned policies;
- Appendix I documents the implementation details for the experiments reported in the main text.

Appendix G gathers perturbation tools used across proofs, split into two parts: (i) when two MDPs have close transition kernels, their value functions are close; and (ii) after truncation, the resulting value functions remain close to those of the original MDP.

Appendices C– E develop the theory for strong list replicability.

- Appendix C analyzes the robust planner: it proves a small sub-optimality gap, establishes the mapping between the tolerance parameter $r_{\text{action}}$ and the selected actions, and derives the generative-model list-size result.

- Appendix D proves structural properties used by the strong resultmost notably, that the number of distinct truncated MDPs (as a function of the reachability threshold) is finite and instance-dependent.

- Appendix E then combines the above ingredients into the complete proof of strong list replicability.

Appendix F presents the algorithm and proof for weak list replicability, which is technically simpler than the strong case.

Appendix H establishes the hardness (lower-bound) result on list complexity.

### A.3 PROOF OUTLINE OF ROBUST PLANNER

This part introduces the following scenario: when we have obtained estimates of all transition probabilities with small errors ($M$ and $\hat{M}$ are $\epsilon_0$-related as defined in Equation 2) , the returned policy satisfies both list replicability (Lemma 5.2) and approximate optimality (Lemma 5.1) .

Lemma 5.1: We obtain approximate optimality through the following decomposition:

$$\underbrace{V_M^* - V_M^{\hat{\pi}}}_{\text{Lemma } 5.1} = \underbrace{V_M^* - V_{\hat{M}}^*}_{\text{Lemma } G.1} + \underbrace{V_{\hat{M}}^* - V_{\hat{M}}^{\hat{\pi}}}_{\text{Lemma } C.1} + \underbrace{V_{\hat{M}}^{\hat{\pi}} - V_M^{\hat{\pi}}}_{\text{Lemma } G.2}$$
$$\leq 2H^2\epsilon_0 + r_{\text{action}}H.$$

Lemma 5.2:

We use $\hat{Q}_h(s,a) - \hat{V}_h(s)$ as an estimate of $\text{Gap}_h(s,a) = V_h^*(s) - Q_h^*(s,a)$ .

$$|\hat{Q}_h(s,a) - \hat{V}_h(s) - \text{Gap}_h(s,a)| \leq |\hat{Q}_h(s,a) - Q_{h,M}^*(s,a)| + |\hat{V}_h(s) - V_h^*(s)|$$
$$= \underbrace{|Q_{h,\hat{M}}^*(s,a) - Q_{h,M}^*(s,a)|}_{\text{Lemma } G.1} + \underbrace{|V_{h,M}^*(s) - V_{h,\hat{M}}^*(s)|}_{\text{Lemma } G.1}$$
$$\leq 2H^2\epsilon_0$$

Note that there are $|S||A|H$ elements in the set $\text{Gap}_M = \{V_{h,M}^*(s) - Q_{h,M}^*(s,a) \mid (s,a) \in S \times A, h \in [H]\}$ which is defined in Equation 1 .

From the figure above, we observe that for the $r_{\text{action}}^1$ and $r_{\text{action}}^2$ not in the shaded regions $\bigcup_{g \in \text{Gap}_M} \text{Ball}(g, 2H^2\epsilon_0)$ , if they lie in the same blank region between the two shaded regions, the policies $\hat{\pi}$ they return are identical.

When $\epsilon_0$ is sufficiently small, the proportion of the shaded area, as well as the failure probability, becomes sufficiently small.

Corollary 5.3: Naturally, for the generative model, the length of the list is $|S||A|H + 1$ .

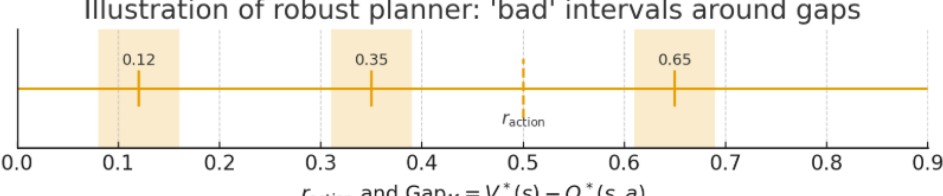

Figure 3: Robust planner

Figure 4: rtrunc illustration

## A.4  PROOF OUTLINE OF WEAKLY REPLICABLE RL

For weak replicability, introduced in Algorithm 3, we first estimate the reachability probabilities using a black-box algorithm, and then remove the states with low reachability probabilities.

We use $\hat{d}(s, h)$ defined in Algorithm 3 to estimate $d_M^*(s, h)|$. When the sample size is sufficiently large, their values are very close:

$$\underbrace{|\hat{d}(s, h) - d_M^*(s, h)|}_{\text{Lemma } F.7} = \underbrace{|d_M^{\hat{\pi}^{s,h}}(s, h) - \hat{d}(s, h)|}_{\text{chernoff bound}} + \underbrace{|d_M^*(s, h) - d_M^{\hat{\pi}^{s,h}}(s, h)|}_{\text{properties of the Algorithm } \mathbb{A}}$$

$$\leq 2\epsilon_0.$$

Lemma F.13: Following the above approach, we define the shaded regions similarly for $r_{\text{trunc}}$:

$$\text{Bad}'_{\text{trunc}} = \bigcup_{(s,h) \in S \times [H]} \text{Ball}(d_M^*(s, h), 2\epsilon_0),$$

There are $|S|H$ elements in the set $\{d_M^*(s, h)\}$, also note that the $r_{\text{trunc}}$ values lying in the same blank region correspond to the same truncated MDP; thus, there are a total of $|S|H + 1$ truncated MDPs $\overline{M}^r$.

Based on the proof of the robust planner above (Lemma 5.2), each truncated MDP $\overline{M}^r$ corresponds to at most $|S||A|H + 1$ policies; thus, the total list length for weak replicability is $(|S|H + 1)(|S||A|H + 1)$

Lemma F.12: The returned policy $\pi$ is $\epsilon$-optimal.

$$V_M^* - V_M^\pi = \underbrace{V_M^* - V_{M^{r_{\text{trunc}}}}^*}_{\text{Lemma } G.3} + \underbrace{V_{M^{r_{\text{trunc}}}}^* - V_{M^{r_{\text{trunc}}}}^\pi}_{\text{Lemma } 5.1} + \underbrace{V_{M^{r_{\text{trunc}}}}^\pi - V_M^\pi}_{\text{Lemma } G.3}$$

$$= H^2 |S| r_{\text{trunc}} + 2H^2 \epsilon_0 + r_{\text{action}} H + 0$$

$$\leq \epsilon$$

## A.5 Proof outline of strong replicable RL

The key difference of strong list replicability lies in that we do not eliminate all the states to be removed at once; instead, we estimate the reachability probabilities using replicable policies **layer by layer** to remove the states. (Algorithm 2)

Due to the dependency between the states removed across layers, the shaded regions we defined earlier are also interdependent; therefore, we must rely on structured information to control the number of truncated MDPs. (This is shown in Appendix D)

Specifically, this property manifests as a form of **monotonicity**: the more states are removed in a given layer, the smaller the estimated reachability probabilities for the next layer, thereby leading to the removal of more states in the subsequent layer. Thus, each state corresponds to a critical $r_{\text{trunc}}$ that determines whether the state is removed, this is defined in Defination D.4:

For each $(s, h) \in S \times [H]$, define $\text{Crit}(s, h) = \inf\{r \in [0, 1] \mid s \in U_h(r)\}$.

Therefore, it is easy to know that there are at most $|S|H + 1$ truncated MDPs.

We note that for each truncated MDP, when selecting policies for arbitrary states via layer-wise estimation, the policies lie within the list of length $|S||A|H + 1$ (Lemma 5.2). Since we perform this operation for all $|S|H$ states, the length of the returned trajectory list for each truncated MDP is $|S|H(|S||A|H + 1)$.

Combining with there are at most $|S|H + 1$ truncated MDPs, the strong list size is $O(|S|^3|A|H^3)$.

Note that we use $d_{\tilde{M}^h}^*(s, h+1)$ to estimate $d_{M^{r_{\text{trunc}}}}^*(s, h+1)$ then for any $s \in S$, $|d_{M^{r_{\text{trunc}}}}^*(s, h+1) - d_{\tilde{M}^h}^*(s, h+1)| \leq H^2 \epsilon_0$ (Lemma E.2).

So we just need $\eta_0$ to be big enough and the failure probability will be small.

The same as weak replicability, we have the returned policy $\pi$ is $\epsilon$-optimal.

$$V_M^* - V_M^\pi = \underbrace{V_M^* - V_{M^{r_{\text{trunc}}}}^*}_{\text{Lemma } G.3} + \underbrace{V_{M^{r_{\text{trunc}}}}^* - V_{M^{r_{\text{trunc}}}}^\pi}_{\text{Lemma } 5.1} + \underbrace{V_{M^{r_{\text{trunc}}}}^\pi - V_M^\pi}_{\text{Lemma } G.3}$$

$$= H^2 |S| r_{\text{trunc}} + 2H^2 \epsilon_0 + r_{\text{action}} H + 0$$

$$\leq \epsilon$$

## B EXPERIMENTS DEMONSTRATING LISTREPLICABILITY

### B.1 MINIMAL CHAINMDP

We conduct preliminary numerical experiments to validate our theoretical predictions.

It directly validates our key claim for the robust planner (Algorithm 1): replacing strict argmax planning with the tolerance and lexicographic rule collapses the set of policies observed across independent runs from many (often exponential in the horizon on neartie instances) to a small list, consistent with our theory for the generative model .

#### B.1.1 SETUP

We consider the following the Chain MDP with horizon $H = 8$; at each level $h \in \{0, \ldots, H - 1\}$ there is a single state and two actions $a \in \{0, 1\}$. Choosing $a$ either advances to the next level (success) or transitions to an absorbing failure state (no reward). Only success at the last level yields reward 1. We make the two actions nearly tied:

$$p_{h,0} = 0.5 + \Delta, \quad p_{h,1} = 0.5 - \Delta, \quad \Delta = 0.02.$$

This is the standard near-tie chain where small estimation noise can flip action choices at many levels, yielding up to $2^H$ distinct greedy policies, exactly the pathology highlighted in Section 4.

For each levelaction pair $(h, a)$, we draw $n = 40$ i.i.d. next-state samples from the simulator, form an empirical MDP $\widehat{M}$, and compute $\widehat{Q}, \widehat{V}$ by backward DP. We notice this is exactly the generative model case.

We compared the following two planners.

- Greedy: $\pi_h = \arg\max_a \widehat{Q}_h(\cdot, a)$.

- Robust planner (Alg. 1): with a fixed tolerance $r_{\text{action}}$, select the first action in a fixed lexicographic order (action 0 before 1) among those satisfying

$$\widehat{Q}_h(\cdot, a) \geq \max_{a'} \widehat{Q}_h(\cdot, a') - r_{\text{action}}.$$

When $r_{\text{action}} = 0$, this reduces exactly to greedy.

Over $R = 500$ independent runs with fresh samples, we count the number of distinct final deterministic policies produced by each planner, denoted distinct policies. This is the empirical analogue of the weak list size.

#### B.1.2 RESULT

Figure 5 shows that when using the greedy algorithm, policies are more dispersed, whereas when using the robust planner, policies are more concentrated, demonstrating stronger replicability and stability.

Figure 6 shows that the list size monotonically decreases with threshold.

#### B.1.3 ANALYZE

**(1)** We observed from Figure 6 that the list size monotonically decreases with threshold. The line plot shows that when $r_{\text{action}}$ increases from 0 to 0.03, the number of distinct policies drops from 168 to 12, almost monotonically. This is completely consistent with the core criterion of Lemma 5.2.

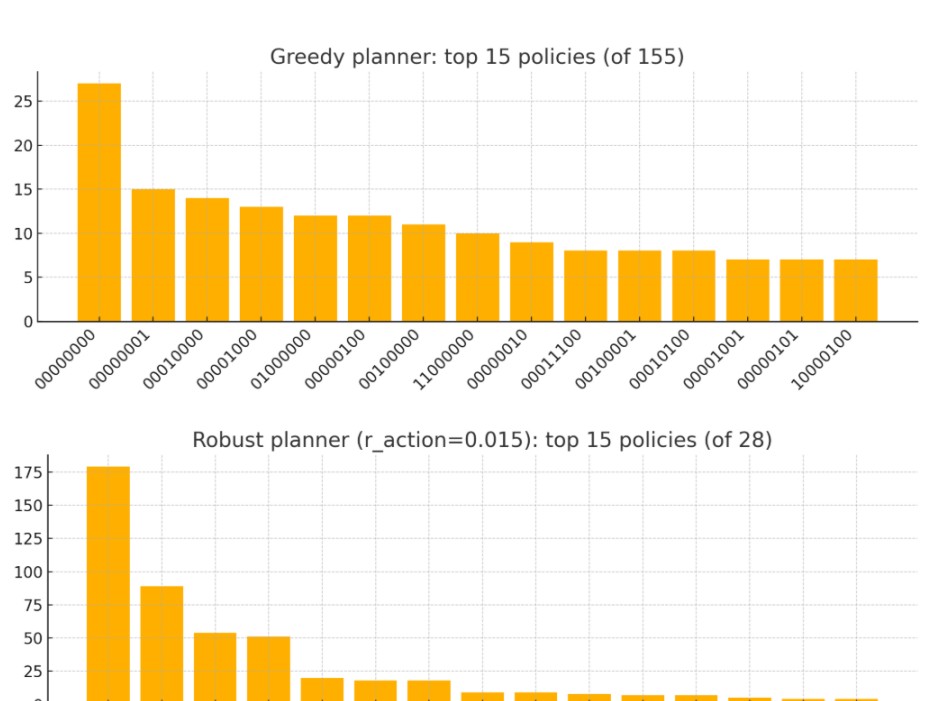

Figure 5: Policy

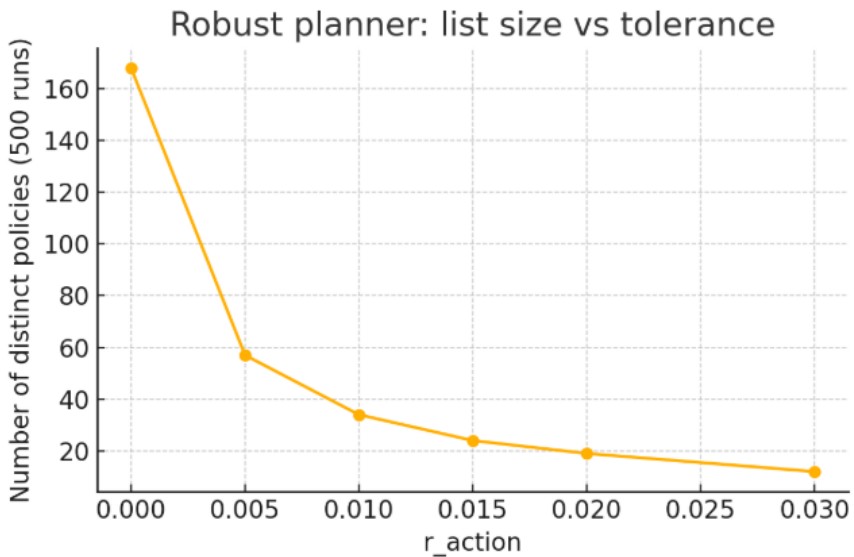

Figure 6: Numbers of Policies over $r_{action}$

**(2)** We observed that Greedy ($r_{\text{action}} = 0$) is extremely unstable, which matches the exponential policy count of the chain counter example . The line plot shows 168 policies at $r_{\text{action}} = 0$ (over 500 runs), while theoretically, the greedy policy in the chain MDP can have up to $\approx 2^H$ outputs under multi-level tiny gaps. The observation is entirely isomorphic to the chain example in Section 4 of the paper: strict $\arg\max Q$ amplifies tiny statistical fluctuations at each level layer by layer, leading to discontinuous jumps across exponentially many policies across runs.

**(3)** Robust Planner Turns Exponential into Polynomial: Under the generative model setting, Corollary 5.3 proves that if $r_{\text{action}}$ is chosen randomly and avoids bad gaps, the number of possible output policies is at most $|S||A|H+1$. Our chain environment satisfies $|S| = H$, $|A| = 2$, so the upper bound is $2H^2 + 1$. For $H = 8$, the upper bound is 129; our list size (1257) for $r_{\text{action}} \in [0.005, 0.03]$ is significantly below the worst-case upper bound. This is consistent with the theoretical expectation that the upper bound is for the worst case, and specific instances are often smaller.

## B.2   AN GRIDWORLD EXPERIMENT

Given that the experimental setup described earlier is overly simplistic, we have conducted analogous experiments in the more complex discrete GridWorld environment. Since the analytical process is analogous to that presented previously, we only elaborate on the experimental setup and report the corresponding results herein.

### B.2.1   SETUP

- **Environment**: An $N \times N$ grid (default $5 \times 5$), with the start state $(0, 0)$ and the terminal state $(N - 1, N - 1)$. The action set is $\{\text{R}, \text{U}\}$.

- **Transition**: Executing R/U succeeds in moving forward with probability $p_{\text{true}}(s, a)$; otherwise, the agent enters a failure absorbing state. Reaching the terminal state yields a reward of 1 and terminates the episode. To create nearly tied action values, a checkerboard-style minor advantage is introduced:

$$p_{\text{true}}(s, \text{R}) = 0.5 \pm \delta, \quad p_{\text{true}}(s, \text{U}) = 0.5 \mp \delta \ (\text{opposite signs for adjacent grids})$$

- **Learning/Planning**: Generative sampling is used to estimate $\hat{p}(s, a)$ (with $n_{\text{per pair}}$ samples per state-action pair), followed by dynamic programming to obtain $\hat{Q}$.

  - **Ordinary**: Greedily select actions via $\text{argmax}\hat{Q}$ for each grid.
  - **Robust**: Select actions lexicographically (R < U) within $\max_a \hat{Q}(s, a) - r_{\text{action}}$ (a simplified implementation of Algorithm 1).

- **Metrics**:

  1. **Policy**: Count the number of distinct output policies across the entire table.
  2. **Trajectory-level (Strong List)**: Follow the learned policy from the start state to the terminal state, count the number of distinct action sequences, and report the minimum $k$ required to cover 90% of runs.

### B.2.2   RESULT

**Result 1:  List Size Shrinks Significantly with Increasing $r_{\text{action}}$ (Policy-level)** We extend $r_{\text{action}}$ to $[0, 0.001, 0.002, 0.0035, 0.005, 0.01, 0.02]$.

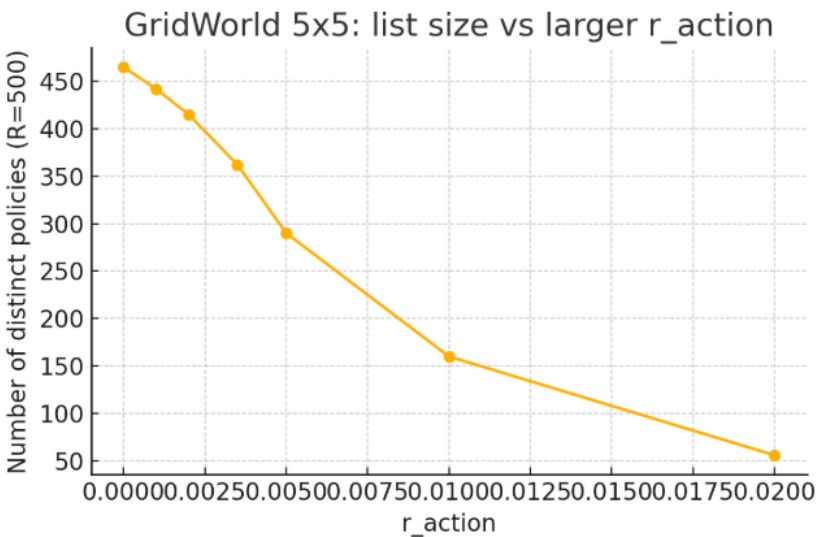

Figure 7:

Number of distinct output policies (over 500 runs):

| $r_{\text{action}}$ | 0.0000 | 0.0010 | 0.0020 | 0.0035 | 0.0050 | 0.01 | 0.02 |
|---|---|---|---|---|---|---|---|
| List Size | 465 | 442 | 415 | 362 | 290 | 160 | 56 |

A monotonic and rapid decrease is also observable in the figure: when $r$ increases from 0 to 0.01, the list size drops from ~465 to ~160; further increasing to 0.02, only 56 policies remain. (Upper line chart: GridWorld $5 \times 5$: list size vs larger $r_{\text{action}}$)

**Result 2: Trace Collapses to Very Few Trajectories under Large** $r$ For $r_{\text{action}} = 0.02$, the number of distinct action sequences from start to terminal state and the minimum $k$ required to cover 90% of runs are as follows:

- Greedy: 64 distinct trajectories, $k_{90} = 40$, and Top-1 coverage is only 9.2%.

- Robust: 5 distinct trajectories, $k_{90} = 2$, and Top-1 coverage is 89.0%.

## C    MISSING PROOFS IN SECTION 5

**Lemma C.1.** *Suppose that two MDPs $M$ and $\hat{M}$ are $\epsilon_0$-related. For the policy $\hat{\pi}$ returned by Algorithm 1, it holds that*

$$0 \leq V_{\hat{M}}^* - V_{\hat{M}}^{\hat{\pi}} \leq r_{\text{action}} H.$$

*Proof.* The lower bound, i.e., $0 \leq V_{\hat{M}}^* - V_{\hat{M}}^{\hat{\pi}}$, is immediate from the definition of $V_{\hat{M}}^*$.

We now prove the upper bound by induction on the time step $h$.

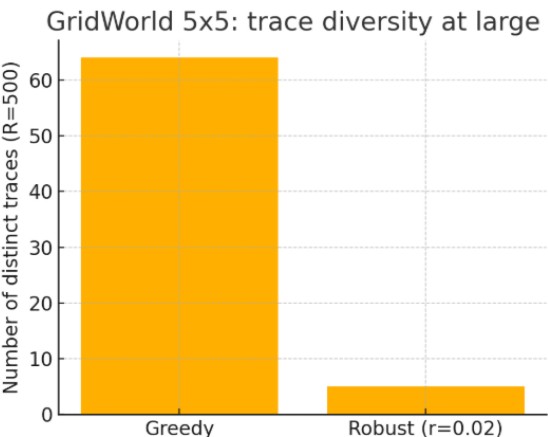

Figure 8: Trace

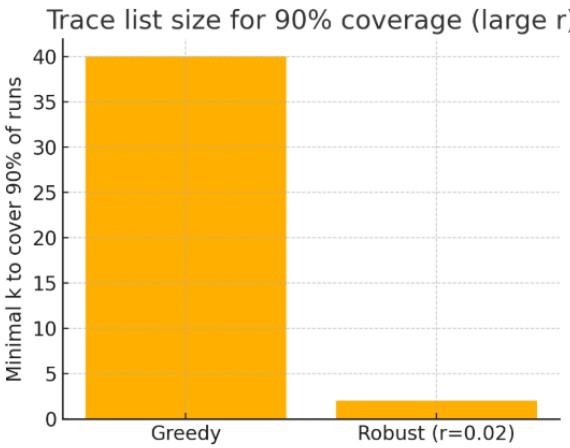

Figure 9: Enter Caption

For $0 \leq h \leq H - 1$, we have

$$V_{h,\hat{M}}^*(s) - V_{h,\hat{M}}^{\hat{\pi}}(s) = V_{h,\hat{M}}^*(s) - Q_{h,\hat{M}}^*(s, \hat{\pi}_h(s)) + Q_{h,\hat{M}}^*(s, \hat{\pi}_h(s)) - Q_{h,\hat{M}}^{\hat{\pi}}(s, \hat{\pi}_h(s))$$

$$\overset{(1)}{\leq} r_{\text{action}} + \sum_{s'} \hat{P}_h(s'|s, \hat{\pi}_h(s)) \cdot V_{h+1,\hat{M}}^*(s') - \sum_{s'} \hat{P}_h(s'|s, \hat{\pi}_h(s)) \cdot V_{h+1,\hat{M}}^{\hat{\pi}}(s')$$

$$= r_{\text{action}} + \sum_{s'} \hat{P}_h(s'|s, \hat{\pi}_h(s)) \cdot \left( V_{h+1,\hat{M}}^*(s') - V_{h+1,\hat{M}}^{\hat{\pi}}(s') \right)$$

$$\leq r_{\text{action}} + \max_s \left( V_{h+1,\hat{M}}^*(s) - V_{h+1,\hat{M}}^{\hat{\pi}}(s) \right).$$

Inequality (1) follows from the definition of $\hat{\pi}$, which guarantees that

$$V_{h,\hat{M}}^*(s) - Q_{h,\hat{M}}^*(s, \hat{\pi}_h(s)) \leq r_{\text{action}}.$$

When $h = H$, we have $V_{H,\hat{M}}^*(s) = V_{H,\hat{M}}^{\hat{\pi}}(s) = 0$. By induction, we have

$$V_{\hat{M}}^* - V_{\hat{M}}^{\hat{\pi}} \leq r_{\text{action}} H.$$

This completes the proof. $\qquad\square$

*Proof of Lemma 5.1.* From Lemma C.1, we have:

$$V_{\hat{M}}^* - V_{\hat{M}}^{\hat{\pi}} \leq r_{\text{action}} H.$$

By Lemma G.1, it follows that:

$$\left| V_M^* - V_{\hat{M}}^* \right| \leq H^2 \epsilon_0.$$

Similarly, from Lemma G.2, we obtain:

$$\left| V_M^{\hat{\pi}} - V_{\hat{M}}^{\hat{\pi}} \right| \leq H^2 \epsilon_0.$$

By combining these inequalities, we have

$$V_M^* - V_M^{\hat{\pi}} = V_M^* - V_{\hat{M}}^* + V_{\hat{M}}^* - V_{\hat{M}}^{\hat{\pi}} + V_{\hat{M}}^{\hat{\pi}} - V_M^{\hat{\pi}}$$

$$\leq 2H^2 \epsilon_0 + r_{\text{action}} H.$$

$\qquad\square$

*Proof of Lemma 5.2.* By Lemma G.1,

For any $(h, s, a) \in [H - 1] \times S \times A$

$$\left| V_{h,M}^*(s) - V_{h,\hat{M}}^*(s) \right| \leq H^2 \epsilon_0,$$

$$\left| Q_{h,M}^*(s, a) - Q_{h,\hat{M}}^*(s, a) \right| \leq H^2 \epsilon_0.$$

Hence,

$$\left| \left( V^*_{h,\hat{M}}(s) - Q^*_{h,\hat{M}}(s,a) \right) - \left( V^*_{h,M}(s) - Q^*_{h,M}(s,a) \right) \right|$$

$$\leq \left| V^*_{h,M}(s) - V^*_{h,\hat{M}}(s) \right| + \left| Q^*_{h,M}(s,a) - Q^*_{h,\hat{M}}(s,a) \right|$$

$$\leq 2H^2 \epsilon_0.$$

For any $g \in \mathrm{Gap}_M$, where $g = V^*_h(s) - Q^*_h(s,a)$, if $g < r^1_{\mathrm{action}} < r^2_{\mathrm{action}}$, then, because $r^1_{\mathrm{action}} \notin \bigcup_{g \in \mathrm{Gap}_M} \mathrm{Ball}(g, 2H^2\epsilon_0)$ and $r^2_{\mathrm{action}} \notin \bigcup_{g \in \mathrm{Gap}_M} \mathrm{Ball}(g, 2H^2\epsilon_0)$, we have

$$\left( V^*_{h,M}(s) - Q^*_{h,M}(s,a) \right) + 2H^2\epsilon_0 < r^1_{\mathrm{action}} < r^2_{\mathrm{action}}.$$

Using the previous bound, we conclude that

$$V^*_{h,\hat{M}}(s) - Q^*_{h,\hat{M}}(s,a) < r^1_{\mathrm{action}} < r^2_{\mathrm{action}}.$$

Similarly, if $r^1_{\mathrm{action}} < r^2_{\mathrm{action}} < g$, we also have:

$$r^1_{\mathrm{action}} < r^2_{\mathrm{action}} < V^*_{h,\hat{M}}(s) - Q^*_{h,\hat{M}}(s,a).$$

Therefore, for both tolerance parameters $r^1_{\mathrm{action}}$ and $r^2_{\mathrm{action}}$, the chosen action $\hat{\pi}_h(s)$ remains the same for all $(s,h) \in S \times [H]$. As a result, the policy $\hat{\pi}$ depends only on $M$ and $r_{\mathrm{action}}$. Moreover, for both tolerance parameters $r^1_{\mathrm{action}}$ and $r^2_{\mathrm{action}}$, the policy $\hat{\pi}$ returned would be identical. $\qquad \square$

**Corollary C.2.** *In the generative model setting, there is an algorithm with sample complexity polynomial in $|S|$, $|A|$, $1/\epsilon$ and $1/\delta$, such that with probability at least $1 - \delta$, the returned policy is $\epsilon$-optimal and always lies in a list $\Pi(M)$ where $\Pi(M)$ is a list of policies that depend only on the unknown underlying MDP $M$ with $|\Pi(M)| = O(|S||A|H)$.*

*Proof.* We collect $N$ samples for each $(s,a) \in S \times A$ and $h \in [H]$ where $N$ is polynomial in $|S|$, $|A|$, $H$, $1/\epsilon$ and $1/\delta$, and use the samples to build an empirical transition model $\hat{P}$ to form an MDP $\hat{M}$. We then invoke Algorithm 1 with MDP $\hat{M}$ and $r_{\mathrm{action}} \sim \mathrm{Unif}(0, \epsilon/(5H))$ and return its output. Standard analysis shows tha $M$ and $\hat{M}$ are $\epsilon_0$-related with $\epsilon_0 = \delta\epsilon/(20H^3)$ with probability at least $1 - \delta/2$. Moreover, $r_{\mathrm{action}} \notin \bigcup_{g \in \mathrm{Gap}_M} \mathrm{Ball}(g, 2H^2\epsilon_0)$ with probability at least $1 - \delta/2$. We condition on the intersection of the above two events which holds with probability at least $1 - \delta$ by union bound. By Lemma 5.1, the returned policy is $\epsilon$-optimal. By Lemma 5.2, the returned policy lies in a list $\Pi(M)$ with size at most $|S||A|H + 1$ since $|\mathrm{Gap}_M| \leq |S||A|H$. $\qquad \square$

# D STRUCTURAL CHARACTERIZATIONS OF REACHING PROBABILITIES IN TRUNCATED MDPS

In this section, we prove several properties of reaching probabilities in MDPs with truncation which will be used later in the analysis Given a reaching probability threshold $r \in [0,1]$, we first define the set of unreachable states $U_h(r)$ for each $h \in [H]$.

**Definition D.1.** *For the underlying MDP $M = (S, A, P, R, H, s_0)$, given a real number $r \in [0,1]$, we define $U_h(r) \subseteq S$ inductively for each $h \in [H]$ as follows:*

- $U_0(r) = \{s \in S \mid \Pr[s_0 = s] \le r\}$;

- *Suppose $U_{h'}(r) \subseteq S$ is defined for all $0 \le h' < h$, define*

$$U_h(r) = \{s \in S \mid \max_\pi \Pr[s_h = s, s_0 \notin U_0(r), s_1 \notin U_1(r), \dots, s_{h-1} \notin U_{h-1}(r) \mid M, \pi] \le r\}.$$

*We also write $U(r) = (U_0(r), U_1(r), \dots, U_{H-1}(r))$.*

Intuitively, the set of unreachable states $U_h(r)$ at level $h \in [H]$ includes all those states that can not be reached with probability larger than a threshold $r$ for any policy $\pi$, where we ignore those unreachable states included in $U_{h'}(r)$ for all levels $h' < h$ when calculating the reaching probabilities. Also note that $U_h(1) = S$.

The main observation is that $U_h(r)$ satisfies the following monotonicity property.

**Lemma D.2.** *Given $0 \le r_1 \le r_2 \le 1$, for any $h \in [H]$, we have $U_h(r_1) \subseteq U_h(r_2)$.*

*Proof.* We prove the above claim by induction on $h$. The claim is clearly true when $h = 0$. Suppose the above claim is true for all $0 \le h' < h$, now we prove that $U_h(r_2) \subseteq U_h(r_1)$. Considering a fixed state $s \in S$, for any fixed policy $\pi$, we have

$$\Pr[s_h = s, s_0 \notin U_0(r_1), s_1 \notin U_1(r_1), \dots, s_{h-1} \notin U_{h-1}(r_1) \mid M, \pi]$$
$$\ge \Pr[s_h = s, s_0 \notin U_0(r_2), s_1 \notin U_1(r_2), \dots, s_{h-1} \notin U_{h-1}(r_2) \mid M, \pi],$$

since $U_{h'}(r_1) \subseteq U_{h'}(r_2)$ for all $h' < h$ under the induction hypothesis. Therefore,

$$\max_\pi \Pr[s_h = s, s_0 \notin U_0(r_1), s_1 \notin U_1(r_1), \dots, s_{h-1} \notin U_{h-1}(r_1) \mid M, \pi]$$
$$\ge \max_\pi \Pr[s_h = s, s_0 \notin U_0(r_2), s_1 \notin U_1(r_2), \dots, s_{h-1} \notin U_{h-1}(r_2) \mid M, \pi]$$

which implies $U_h(r_1) \subseteq U_h(r_2)$. $\qquad\qquad\square$

An important corollary of Lemma D.2, is that the total number of distinct $U(r)$ for all $r \in [0, 1]$ is upper bounded by $|S|H + 1$.

**Corollary D.3.** *For all $r \in [0, 1]$, there are at most of $|S|H + 1$ unique sequences of sets $U(r)$.*

*Proof.* Assume for the sake of contradiction that there are more than $|S|H + 1$ unique sequences of sets $U(r)$. Note that $0 \le \sum_{h \in [H]} |U(r)| \le |S|H$ for all $r \in [0, 1]$. By the pigeonhole principle, there exists $0 \le r_1 < r_2 \le 1$ such that $U(r_1) \ne U(r_2)$ while $\sum_{h \in [H]} |U(r_1)| = \sum_{h \in [H]} |U(r_2)|$. By Lemma D.2, for all $h \in [H]$, we have $U_h(r_1) \subseteq U_h(r_2)$ and thus $|U_h(r_1)| \le |U_h(r_2)|$. This implies that $|U_h(r_1)| = |U_h(r_2)|$ for all $h \in [H]$. For any $h \in [H]$, we have $U_h(r_1) \subseteq U_h(r_2)$ and $|U_h(r_1)| = |U_h(r_2)|$ which implies $U_h(r_1) = U_h(r_2)$, contradicting the assumption that $U(r_1) \ne U(r_2)$. $\qquad\qquad\square$

For each $(s, h) \in S \times [H]$, we define $\mathrm{Crit}(s, h)$ to be the infimum of those reaching probability threshold $r \in [0, 1]$ so that $s$ would be unreachable under $r$.

**Definition D.4.** *For each $(s, h) \in S \times [H]$, define $\mathrm{Crit}(s, h) = \inf\{r \in [0, 1] \mid s \in U_h(r)\}$.*

Note that $\{r \in [0, 1] \mid s \in U_h(r)\}$ is never an empty set since $U_h(1) = S$.

Lemma D.2 implies that $\mathrm{Crit}(s, h)$ is the critical reaching probability threshold for $(s, h)$, formalized as follows.

**Corollary D.5.** *For any $(s, h) \in S \times [H]$, we have*

- *for any $1 \geq r > \mathrm{Crit}(s, h)$, $s \in U_h(r)$;*

- *for any $0 \leq r < \mathrm{Crit}(s, h)$, $s \notin U_h(r)$.*

Given the definition of unreachable states $U_h(r)$, for each $r \in [0, 1]$, we now formally define the truncated MDP $M^r$ where we direct the transition probabilities of all unreachable states to an absorbing state $s_{\mathrm{absorb}}$.

**Definition D.6.** *For the underlying MDP $M = (S, A, P, R, H, s_0)$, given a real number $r \in [0, 1]$, define $M^r = (S \cup \{s_{\mathrm{absorb}}\}, A, P^r, R, H, s_0)$, where*

$$P_h^r(s' \mid s, a) = \begin{cases} P_h(s' \mid s, a) & s \notin U_h(r) \cup \{s_{\mathrm{absorb}}\}, s' \neq s_{\mathrm{absorb}} \\ 0 & s \notin U_h(r) \cup \{s_{\mathrm{absorb}}\}, s' = s_{\mathrm{absorb}} \\ \mathbb{1}[s' = s_{\mathrm{absorb}}] & s \in U_h(r) \cup \{s_{\mathrm{absorb}}\} \end{cases} \tag{4}$$

The following lemma builds a connection between the occupancy function in $M^r$ and the set of unreachable states $U_h(r)$.

**Lemma D.7.** *For any $r \in [0, 1]$, for any $(s, h) \in S \times [H]$*

$$d_{M^r}^*(s, h) = \max_\pi \Pr[s_h = s, s_0 \notin U_0(r), s_1 \notin U_1(r), \ldots, s_{h-1} \notin U_{h-1}(r) \mid M, \pi],$$

*and therefore $s \in U_h(r)$ if and only if $d_{M^r}^*(s, h) \leq r$.*

*Proof.* By the construction of $M^r$,

$$d_{M^r}^\pi(s, h) = \Pr[s_h = s, s_0 \notin U_0(r), s_1 \notin U_1(r), \ldots, s_{h-1} \notin U_{h-1}(r) \mid M, \pi],$$

and therefore,

$$d_{M^r}^*(s, h) = \max_\pi \Pr[s_h = s, s_0 \notin U_0(r), s_1 \notin U_1(r), \ldots, s_{h-1} \notin U_{h-1}(r) \mid M, \pi],$$

which also implies that $s \in U_h(r)$ if and only if $d_{M^r}^*(s, h) \leq r$ by Definition D.1. $\qquad\square$

Combining Lemma D.7 and Lemma D.2, we have the following corollary which shows that $d_{M^r}^*(s, h)$ is monotonically non-increasing as we increase $r$.

**Corollary D.8.** *For the underlying MDP $M = (S, A, P, R, H, s_0)$, for any $0 \leq r_1 \leq r_2 \leq 1$ and any $(s, h) \in S \times [H]$, we have $d_{M^{r_1}}^*(s, h) \geq d_{M^{r_2}}^*(s, h)$. Moreover, $d_M^*(s, h) \geq d_{M^r}^*(s, h)$ for any $(s, h) \in S \times [H]$ and $r \in [0, 1]$.*

As illustrated in the following lemma, $d_{M^r}^*(s, h) \leq \mathrm{Crit}(s, h)$ whenever $r > \mathrm{Crit}(s, h)$, and $d_{M^r}^*(s, h) \geq \mathrm{Crit}(s, h)$ if $r < \mathrm{Crit}(s, h)$.

**Lemma D.9.** *For any $r \in [0, 1]$ and $(s, h) \in S \times [H]$,*

- *if $r > \mathrm{Crit}(s, h)$, $d_{M^r}^*(s, h) \leq \mathrm{Crit}(s, h)$;*

- *if $r < \mathrm{Crit}(s, h)$, $d_{M^r}^*(s, h) \geq \mathrm{Crit}(s, h)$.*

*Proof.* We only consider the case $r > \text{Crit}(s, h)$ in the proof, and the case $r < \text{Crit}(s, h)$ can be handled using exactly the same argument.

Since $r > \text{Crit}(s, h)$, by Corollary D.5, we have $s \in U_h(r)$, which implies $d^*_{M^r}(s, h) \le r$ by Lemma D.7. Assume for the sake of contradiction that $d^*_{M^r}(s, h) > \text{Crit}(s, h)$. Let $r'$ be an arbitrary real number satisfying $\text{Crit}(s, h) < r' < d^*_{M^r}(s, h) \le r$. By Corollary D.8, we have $d^*_{M^{r'}}(s, h) \ge d^*_{M^r}(s, h) > r'$, which implies $s \notin U_h(r')$ by Lemma D.7. On the other hand, since $r' > \text{Crit}(s, h)$, we must have $s \in U_h(r')$ by Corollary D.5 which leads to a contradiction. $\square$

For each $(s, h) \in S \times [H]$ and $r \in [0, 1]$, we also define an auxiliary MDP $M^{r,s,h}$ based on $M^r$, which will be later used in the analysis of our algorithm.

**Definition D.10.** *For each $(s, h) \in S \times [H]$ and $r \in [0, 1]$, define $M^{r,s,h}$ to be the MDP that has the same state space, action space, horizon length and initial state as $M^r$. The reward function of $M^{r,s,h}$ is $R^{s,h}_{h'}(s', a) = \mathbb{1}[h' = h, s' = s]$ for all $h' \in [H]$ and $(s', a) \in (S \cup \{s_{\text{absorb}}\}) \times A$, and the transition model of $M^{r,s,h}$ is*

$$P^{r,h}_{h'}(s'' \mid s', a) = \begin{cases} P^r_{h'}(s'' \mid s', a) & h' < h \\ \mathbb{1}[s'' = s_{\text{absorb}}] & h' \ge h \end{cases}, \tag{5}$$

*where $P^r$ is the transition model of $M^r$ define in (6).*

A direct observation is that for any $(s, h) \in S \times [H]$ and $r \in [0, 1]$, for any policy $\pi$, $d^\pi_{M^r}(s, h) = V^\pi_{M^{r,s,h}}$, which also implies $d^*_{M^r}(s, h) = V^*_{M^{r,s,h}}$.

# E    MISSING PROOFS IN SECTION 6

In this section, we give the formal proof of Theorem 6.1 based on the tools developed in Section D.

**Lemma E.1.** *Consider a pair of fixed choices of $r_{\text{trunc}}$ and $r_{\text{action}}$ in Algorithm 2. For a fixed $h \in [H - 1]$, if for all $s \in S \setminus \hat{U}_h$ we have $d^{\hat{\pi}^{s,h}}_M \ge \eta_0$ whenever $h > 0$, then with probability $1 - \frac{\delta}{2H}$, for all $(s, a) \in (S \setminus \hat{U}_h) \times A$,*

$$\sum_{s' \in S} |P_h(s' \mid s, a) - \hat{P}_h(s' \mid s, a)| \le \epsilon_0.$$

*Proof.* We divide the proof into two parts. First, we demonstrate that we have a sufficient number of effective samples. Second, we show that the estimation error is small.

For a given $(s, a) \in (S \setminus \hat{U}_h) \times A$, we first prove that with probability at least $1 - \frac{\delta}{4H|S||A|}$, the number of effective samples is greater than $\frac{W\eta_0}{2}$, where the number of effective samples is defined as

$$W_{\text{effective}} = \sum_{w=1}^{W} \mathbb{1}[(s^{(w)}_h, a^{(w)}_h) = (s, a)].$$

Given that $d^{\hat{\pi}^{s,h}}_M \ge \eta_0$, we have

$$\frac{\mathbb{E}[W_{\text{effective}}]}{W} = \frac{W \cdot d^{\hat{\pi}^{s,h}}_M}{W} = d^{\hat{\pi}^{s,h}}_M \ge \eta_0,$$

and therefore by Chernoff bound,

$$\mathbb{P}\left(W_{\text{effective}} < \frac{\eta_0}{2}W\right) \leq \mathbb{P}\left(d_M^{\hat{\pi}^{s,h}} - \frac{W_{\text{effective}}}{W} > \frac{\eta_0}{2}\right) < 2e^{-2\left(\frac{\eta_0}{2}\right)^2 W} < \frac{\delta}{4H|S||A|}.$$

Thus, with probability at least $1 - \frac{\delta}{4H|S||A|}$, the number of effective samples is at least $\frac{W\eta_0}{2}$.

Next, we show that if the number of effective samples is greater than $\frac{W\eta_0}{2}$, then with probability at least $1 - \frac{\delta}{4H|S||A|}$,

$$\sum_{s' \in S} |P_h(s' \mid s, a) - \hat{P}_h(s' \mid s, a)| \leq \epsilon_0.$$

To establish this, we first prove that for any specific $s'$, with probability at least $1 - \frac{\delta}{4H|S|^2|A|}$, we have

$$|P_h(s' \mid s, a) - \hat{P}_h(s' \mid s, a)| \leq \frac{\epsilon_0}{|S|}.$$

Using the Chernoff bound,

$$\mathbb{P}\left(|P_h(s' \mid s, a) - \hat{P}_h(s' \mid s, a)| \geq \frac{\epsilon_0}{|S|}\right) < 2e^{-2\left(\frac{\epsilon_0}{S}\right)^2 W_{\text{effective}}} < \frac{\delta}{4H|S|^2|A|}.$$

Therefore, by the union bound, with probability at least $1 - \frac{\delta}{4H|S||A|}$, we have for all $s' \in S$,

$$|P_h(s' \mid s, a) - \hat{P}_h(s' \mid s, a)| \leq \frac{\epsilon_0}{|S|}.$$

Summing over all $s'$ gives

$$\sum_{s' \in S} |P_h(s' \mid s, a) - \hat{P}_h(s' \mid s, a)| \leq \epsilon_0.$$

Combining these results, we conclude that for a specific $(s, a)$, with probability at least $1 - \frac{\delta}{2H|S||A|}$,

$$\sum_{s' \in S} |P_h(s' \mid s, a) - \hat{P}_h(s' \mid s, a)| \leq \epsilon_0.$$

Thus, for a fixed $h \in [H-1]$, if for all $s \in S \setminus \hat{U}_h$ we have $d_M^{\hat{\pi}^{s,h}} \geq \eta_0$ whenever $h > 0$, then with probability $1 - \frac{\delta}{2H}$, for all $(s, a) \in (S \setminus \hat{U}_h) \times A$,

$$\sum_{s' \in S} |P_h(s' \mid s, a) - \hat{P}_h(s' \mid s, a)| \leq \epsilon_0.$$

$\square$

**Lemma E.2.** *Consider a pair of fixed choices of $r_{\text{trunc}} < 1$ and $r_{\text{action}}$ in Algorithm 2. For any $h \in [H-1]$, if for all $h' \leq h$, we have*

- $\hat{U}_{h'} = U_{h'}(r_{\text{trunc}})$;

- $\sum_{s'} |\hat{P}_{h'}(s' \mid s, a) - P_{h'}(s' \mid s, a)| \leq \epsilon_0$ *for all* $(s, a) \in (S \setminus \hat{U}_{h'}) \times A$,

*then for any $s \in S$, $|d^*_{M^{r_{\text{trunc}}}}(s, h+1) - d^*_{\bar{M}^h}(s, h+1)| \leq H^2 \epsilon_0$.*

*Proof.* Consider a fixed level $h \in [H-1]$ and state $s \in S$. Note that $d^*_{M^{r_{\text{trunc}}}}(s, h+1) = V^*_{M^{r_{\text{trunc}},s,h+1}}$ and $d^*_{\tilde{M}^h}(s, h+1) = V^*_{\tilde{M}^{s,h+1}}$.

Note that $M^{r_{\text{trunc}},s,h+1}$ and $\tilde{M}^{s,h+1}$ share the same state space, action space, reward function and initial state. Moreover, we have $\hat{U}_{h'} = U_{h'}(r_{\text{trunc}})$ for all $h' \leq h$ and $\sum_{s'} |\hat{P}_{h'}(s' \mid s, a) - P_{h'}(s' \mid s, a)| \leq \epsilon_0$ for all $h' \leq h$ and $(s, a) \in (S \setminus \hat{U}_{h'}) \times A$. Let $P^{r_{\text{trunc}},h+1}$ be the transition model of $M^{r_{\text{trunc}},s,h+1}$ defined in (5), and $\tilde{P}^{h+1}$ be the transition model of $\tilde{M}^{s,h+1}$ defined in (3). For all $h' \in [H]$, for any $(s, a) \in (S \cup \{s_{\text{absorb}}\}) \times A$, we have

$$\sum_{s' \in S \cup \{s_{\text{absorb}}\}} |P^{r_{\text{trunc}},h+1}_{h'}(s' \mid s, a) - \tilde{P}^{h+1}_{h'}(s' \mid s, a)| \leq \epsilon_0.$$

By Lemma G.1, we have $|V^*_{M^{r_{\text{trunc}},s,h+1}} - V^*_{\tilde{M}^{s,h+1}}| \leq H^2 \epsilon_0$, which implies the desired result. $\qquad\square$

**Lemma E.3.** *Consider a pair of fixed choices of $r_{\text{trunc}} \in (\eta_1, 2\eta_1)$ and $r_{\text{action}}$ in Algorithm 2. For any $h \in [H-1]$, if for all $h' \leq h$, we have*

- $\hat{U}_{h'} = U_{h'}(r_{\text{trunc}})$;

- $\sum_{s'} |\hat{P}_{h'}(s' \mid s, a) - P_{h'}(s' \mid s, a)| \leq \epsilon_0$ *for all $(s, a) \in (S \setminus \hat{U}_{h'}) \times A$,*

*then for any $s \in (S \setminus \hat{U}_{h+1})$, $d^{\hat{\pi}^{s,h+1}}_M(s, h+1) \geq \eta_0$.*

*Proof.* Consider a fixed level $h \in [H-1]$ and $s \in (S \setminus \hat{U}_{h+1})$. Since $s \in (S \setminus \hat{U}_{h+1})$, we have

$$d^*_{\tilde{M}^h}(s, h+1) > r_{\text{trunc}}.$$

By Lemma E.2,

$$d^*_{M^{r_{\text{trunc}}}}(s, h+1) \geq r_{\text{trunc}} - H^2 \epsilon_0 \geq \eta_1 - \eta_0.$$

Notice that $2H^2 \epsilon_0 + r_{\text{action}} H \leq 2H^2 \epsilon_0 + 2\epsilon_1 H \leq 3\epsilon_1 H \leq \eta_0$. By the same analysis as in Lemma E.2, for the returned policy $\hat{\pi}^{s,h+1}$, by Lemma 5.1,

$$V^{\hat{\pi}^{s,h+1}}_{M^{r_{\text{trunc}},s,h+1}} \geq V^*_{M^{r_{\text{trunc}},s,h+1}} - \eta_0 = d^*_{M^{r_{\text{trunc}}}}(s, h+1) - \eta_0 \geq \eta_1 - 2\eta_0 \geq \eta_0,$$

and therefore $d^{\hat{\pi}^{s,h+1}}_{M^{r_{\text{trunc}}}}(s, h+1) \geq \eta_0$. By Lemma D.8, this implies $d^{\hat{\pi}^{s,h+1}}_M(s, h+1) \geq \eta_0$. $\qquad\square$

**Definition E.4.** *Define*

$$\text{Bad}_{\text{trunc}} = \bigcup_{(s,h) \in S \times [H]} \text{Ball}(\text{Crit}(s, h), H^2 \epsilon_0),$$

*where $\text{Crit}(s, h)$ is as defined in Definition D.4.*

**Lemma E.5.** *Consider a pair of fixed choices of $r_{\text{trunc}} \in (\eta_1, 2\eta_1)$ and $r_{\text{action}}$ in Algorithm 2 such that $r_{\text{trunc}} \notin \text{Bad}_{\text{trunc}}$. For any $h \in [H-1]$, if for all $h' \leq h$, we have*

- $\hat{U}_{h'} = U_{h'}(r_{\text{trunc}})$;

- $\sum_{s'} |\hat{P}_{h'}(s' \mid s, a) - P_{h'}(s' \mid s, a)| \leq \epsilon_0$ *for all $(s, a) \in (S \setminus \hat{U}_{h'}) \times A$,*

*then $\hat{U}_{h+1} = U_{h+1}(r_{\text{trunc}})$.*

*Proof.* By Lemma E.2, for any $s \in S$ we have

$$|d^*_{M^{r_{\text{trunc}}}}(s, h+1) - d^*_{\tilde{M}^h}(s, h+1)| \leq H^2 \epsilon_0.$$

Therefore, for any $s \in U_{h+1}(r_{\text{trunc}})$, we have

$$d^*_{\tilde{M}^h}(s, h+1) \leq d^*_{M^{r_{\text{trunc}}}}(s, h+1) + H^2 \epsilon_0.$$

By Corollary D.5, we have $r_{\text{trunc}} \geq \text{Crit}(s, h+1)$. Moreover, since $r_{\text{trunc}} \notin \text{Bad}_{\text{trunc}}$, it holds that

$$r_{\text{trunc}} \notin [\text{Crit}(s, h+1) - H^2 \epsilon_0, \text{Crit}(s, h+1) + H^2 \epsilon_0],$$

which further implies that

$$r_{\text{trunc}} > \text{Crit}(s, h+1) + H^2 \epsilon_0.$$

Combining the above inequality with Lemma D.9, we have

$$r_{\text{trunc}} > \text{Crit}(s, h+1) + H^2 \epsilon_0 \geq d^*_{M^{r_{\text{trunc}}}}(s, h+1) + H^2 \epsilon_0 \geq d^*_{\tilde{M}^h}(s, h+1),$$

which implies $s \in \hat{U}_{h+1}$.

For those $s \notin U_{h+1}(r_{\text{trunc}})$, it can be shown that $s \notin \hat{U}_{h+1}$ using the same argument. Therefore, $\hat{U}_{h+1} = U_{h+1}(r_{\text{trunc}})$.

$\square$

**Lemma E.6.** *Consider a pair of fixed choices of $r_{\text{trunc}} \in (\eta_1, 2\eta_1)$ and $r_{\text{action}}$ in Algorithm 2 such that $r_{\text{trunc}} \notin$* $\text{Bad}_{\text{trunc}}$*. With probability at least $1 - \delta/2$, we have*

- $\hat{U}_h = U_h(r_{\text{trunc}})$ *for all $h \in [H]$;*

- $\sum_{s'} |\hat{P}_h(s' \mid s, a) - P_h(s' \mid s, a)| \leq \epsilon_0$ *for all $h \in [H-1]$ and $(s, a) \in (S \setminus \hat{U}_{h'}) \times A$.*

*Proof.* For each $h \in [H]$, let $\mathcal{E}_h$ be the event that

- $\hat{U}_h = U_h(r_{\text{trunc}})$;

- if $h > 0$, $d^{\hat{\pi}^{s,h}}_M(s, h) \geq \eta_0$ for all $s \in S \setminus \hat{U}_h$;

- if $h > 0$, $\sum_{s' \in S} |\hat{P}_{h-1}(s' \mid s, a) - P_{h-1}(s' \mid s, a)| \leq \epsilon_0$ for all $(s, a) \in (S \setminus \hat{U}_{h-1}) \times A$.

Note that $\mathcal{E}_0$ holds deterministically, since we always have $r_{\text{trunc}} < 1$ which implies $U_0(r_{\text{trunc}}) = S \setminus \{s_0\}$. For each $h < H$, conditioned on $\bigcap_{h' \leq h} \mathcal{E}_{h'}$, by Lemma E.5 and Lemma E.3, we have $\hat{U}_{h+1} = U_{h+1}(r_{\text{trunc}})$, and for all $s \in S \setminus \hat{U}_{h+1}$, $d^{\hat{\pi}^{s,h+1}}_M(s, h+1) \geq \eta_0$. Moreover, by Lemma E.1, with probability at least $1 - \delta/(2H)$,

$$\sum_{s' \in S} |\hat{P}_h(s' \mid s, a) - P_h(s' \mid s, a)| \leq \epsilon_0$$

for all $(s, a) \in (S \setminus \hat{U}_h) \times A$. Therefore, conditioned on $\bigcap_{h' \leq h} \mathcal{E}_{h'}$, $\mathcal{E}_{h+1}$ holds with probability at least $1 - \delta/(2H)$. By the chain rule, $P\left(\bigcap_{h \in [H]} \mathcal{E}_h\right) \geq (1 - \delta/(2H))^{H-1} \geq 1 - \delta/2$.

$\square$

**Definition E.7.** *For a real number $r \in [0, 1]$, define*

$$\text{Gap}(r) = \left(\bigcup_{h \in [H], s \in S \setminus U_h(r)} \text{Gap}_{M^{r,s,h}}\right) \cup \text{Gap}_{M^r}.$$

*Moreover, define*

$$\mathrm{Bad}_{\mathrm{action}}(r) = \bigcup_{g \in \mathrm{Gap}(r)} \mathrm{Ball}(g, 2H^2\epsilon_0).$$

Clearly, for any $r \in [0,1]$, $|\mathrm{Gap}(r)| \leq 2|S|^2 H^2|A|$. Moreover, since $M^r$ and $M^{r,s,h}$ depends only on $U(r)$ (cf. Definition D.6 and Definition D.10), for $r_1, r_2 \in [0,1]$ with $U(r_1) = U(r_2)$, we would have $\mathrm{Gap}(r_1) = \mathrm{Gap}(r_2)$ and $\mathrm{Bad}_{\mathrm{action}}(r_1) = \mathrm{Bad}_{\mathrm{action}}(r_2)$.

**Lemma E.8.** *Given $r^1_{\mathrm{trunc}}, r^2_{\mathrm{trunc}} \in (\eta_1, 2\eta_1) \setminus \mathrm{Bad}_{\mathrm{trunc}}$ and $r^1_{\mathrm{action}}, r^2_{\mathrm{action}} \in (\epsilon_1, 2\epsilon_1)$, suppose*

- $U(r^1_{\mathrm{trunc}}) = U(r^2_{\mathrm{trunc}})$;

- $r^1_{\mathrm{action}} \notin \mathrm{Bad}_{\mathrm{action}}(r^1_{\mathrm{trunc}})$, and $r^2_{\mathrm{action}} \notin \mathrm{Bad}_{\mathrm{action}}(r^1_{\mathrm{trunc}})$;

- *for any $g \in \mathrm{Gap}(r^1_{\mathrm{trunc}})$, either $g < r^1_{\mathrm{action}} < r^2_{\mathrm{action}}$ or $r^1_{\mathrm{action}} < r^2_{\mathrm{action}} < g$,*

*conditioned on the event in Lemma E.6, in Algorithm 2 , the returned policy $\pi$ and $\hat{\pi}^{s,h+1,a}$ will be identical for all $h \in [H-1]$, $(s,a) \in \left(S \setminus \hat{U}_{h+1}\right) \times A$, for all $(r_{\mathrm{action}}, r_{\mathrm{trunc}}) \in \{r^1_{\mathrm{action}}, r^2_{\mathrm{action}}\} \times \{r^1_{\mathrm{trunc}}, r^2_{\mathrm{trunc}}\}$.*

*Proof.* Consider a fixed $h \in [H-1]$ and $(s,a) \in \left(S \setminus \hat{U}_{h+1}\right) \times A$. Since $U(r^1_{\mathrm{trunc}}) = U(r^2_{\mathrm{trunc}})$, we write

- $U(r_{\mathrm{trunc}}) = U(r^1_{\mathrm{trunc}}) = U(r^2_{\mathrm{trunc}})$;

- $\mathrm{Bad}_{\mathrm{action}}(r_{\mathrm{trunc}}) = \mathrm{Bad}_{\mathrm{action}}(r^1_{\mathrm{trunc}}) = \mathrm{Bad}_{\mathrm{action}}(r^2_{\mathrm{trunc}})$;

- $\mathrm{Gap}(r_{\mathrm{trunc}}) = \mathrm{Gap}(r^1_{\mathrm{trunc}}) = \mathrm{Gap}(r^2_{\mathrm{trunc}})$; and

- $M^{r_{\mathrm{trunc}},s,h+1} = M^{r^1_{\mathrm{trunc}},s,h+1} = M^{r^2_{\mathrm{trunc}},s,h+1}$

in the remaining part of the proof.

Let $P^{r_{\mathrm{trunc}}}$ be the transition model of $M^{r_{\mathrm{trunc}},s,h+1}$ defined in (6), and $\tilde{P}^{h+1}$ be the transition model of $\tilde{M}^{s,h+1}$ defined in (3). Note that conditioned on the event in Lemma E.6, $\hat{U}_{h+1} = U_{h+1}(r_{\mathrm{trunc}})$, and therefore, for all $h' \in [H]$, for any $(s,a) \in (S \cup \{s_{\mathrm{absorb}}\}) \times A$, we have

$$\sum_{s' \in S \cup \{s_{\mathrm{absorb}}\}} |P^{r_{\mathrm{trunc}},h+1}_{h'}(s' \mid s,a) - \tilde{P}^{h+1}_{h'}(s' \mid s,a)| \leq \epsilon_0.$$

By Definition E.7, for any $g \in \mathrm{Gap}_{M^{r_{\mathrm{trunc}},s,h+1}}$ , we have

- $r^1_{\mathrm{action}}, r^2_{\mathrm{action}} \notin \mathrm{Ball}(g, 2H^2\epsilon_0)$;

- either $g < r^1_{\mathrm{action}} < r^2_{\mathrm{action}}$ or $r^1_{\mathrm{action}} < r^2_{\mathrm{action}} < g$,

which implies $\hat{\pi}^{s,h+1}$ in Algorithm 2 will be identical for all $(r_{\mathrm{action}}, r_{\mathrm{trunc}}) \in \{r^1_{\mathrm{action}}, r^2_{\mathrm{action}}\} \times \{r^1_{\mathrm{trunc}}, r^2_{\mathrm{trunc}}\}$ by Lemma 5.2. This also implies that $\hat{\pi}^{s,h+1,a}$ will be identical for all $(r_{\mathrm{action}}, r_{\mathrm{trunc}}) \in \{r^1_{\mathrm{action}}, r^2_{\mathrm{action}}\} \times \{r^1_{\mathrm{trunc}}, r^2_{\mathrm{trunc}}\}$. Similarly, the desired property holds also for the returned policy $\pi$. $\square$

*Proof of Theorem 6.1.* Note that

$$\Pr[r_{\mathrm{trunc}} \notin \mathrm{Bad}_{\mathrm{trunc}}] \geq 1 - \delta/4.$$

For any fixed choice of $r_{\text{trunc}}$,

$$\Pr[r_{\text{action}} \notin \text{Bad}_{\text{action}}(r_{\text{trunc}})] \geq 1 - \delta/4.$$

Combining these with Lemma E.6, with probability at least $1 - \delta$, we have

- $r_{\text{trunc}} \notin \text{Bad}_{\text{trunc}}$;

- $r_{\text{action}} \notin \text{Bad}_{\text{action}}(r_{\text{trunc}})$;

- $\hat{U}_h = U_h(r_{\text{trunc}})$ for all $h \in [H]$;

- $\sum_{s'} |\hat{P}_h(s' \mid s, a) - P_h(s' \mid s, a)| \leq \epsilon_0$ for all $h \in [H-1]$ and $(s, a) \in (S \setminus \hat{U}_{h'}) \times A$.

We condition on the above event in the remaining part of the proof.

Conditioned on the above event, for the returned policy $\pi$, we have

$$V_M^\pi \geq V_{M^{r_{\text{trunc}}}}^\pi \geq V_{M^{r_{\text{trunc}}}}^* - 2H^2\epsilon_0 - r_{\text{action}}H \geq V_M^* - 2H^2\epsilon_0 - r_{\text{action}}H - H^2|S|r_{\text{trunc}} \geq V_M^* - \epsilon,$$

where the first inequality is due to Lemma G.3, the second inequality is due to Lemma 5.1, the third inequality is due to Lemma G.3, and the last inequality is due to $r_{\text{trunc}} \leq 2\eta_1$ and $r_{\text{action}} \leq 2\epsilon_1$. Therefore, the returned policy $\pi$ is $\epsilon$-optimal.

By Lemma D.3, there are at most of $SH + 1$ unique sequences of sets $U(r)$. Moreover, for each $r$, $|\text{Gap}(r)| \leq 2|S|^2H^2|A|$. By Lemma E.6, the sequence of policies executed by Algorithm 2 and the policy returned by Algorithm 2 lie in a list $\text{Trace}(M)$ with size $|\text{Trace}(M)| \leq (SH+1)(2|S|^2H^2|A|+1)$. $\qquad\square$

## F  WEAKLY $k$-LIST REPLICABLE RL ALGORITHM

In this section, we present our RL algorithm with weakly $k$-list replicability guarantees. See Algorithm 3 for the formal description of the algorithm. In Algorithm 3, it is assumed that we have access to a black-box algorithm $\mathbb{A}(\epsilon_0, \delta_0)$, so that after interacting with the underlying MDP, with probability at least $1 - \delta_0$, $\mathbb{A}$ returns an $\epsilon_0$-optimal policy.

In Algorithm 3, for each $(s, h) \in S \times H$, we first invoke $\mathbb{A}$ on the underlying MDP with modified reward function $R_{h'}^{s,h}(s', a) = \mathbb{1}[h' = h, s' = s]$ for all $h' \in [H]$ and $(s', a) \in S \times A$. The returned policy $\hat{\pi}^{s,h}$ is supposed to reach state $s$ at level $h$ with probability close to $d^*(s, h)$, and therefore we use $\hat{\pi}^{s,h}$ to collect samples and calculate $\hat{d}(s, h)$ which is our estimate of $d^*(s, h)$. For each action $a \in A$, we also construct a policy $\hat{\pi}^{s,h,a}$ based on $\hat{\pi}^{s,h}$ to collect samples for $(s, a) \in S \times A$ at level $h \in [H]$, and we calculate $\hat{P}_h(s, a)$ which is our estimate of $P_h(s, a)$ based the obtained samples.

For those $(s, h) \in S \times [H]$ with $\hat{d}(s, h) \leq r_{\text{trunc}}$, we remove state $s$ from level $h$ by including $s$ in $\hat{T}_h$. Here $r_{\text{trunc}}$ is a randomly chosen reaching probability threshold drawn from the uniform distribution.

Finally, based on $\hat{P}$ and $\hat{T}$, we build an MDP $\hat{M}$ which is our estimate of the underlying MDP $M$. For each $(s, h)$, if $s \in \hat{T}_h$, then we always transit $s$ to an absorbing state $s_{\text{absorb}}$. Otherwise, we directly use our estimated transition model $\hat{P}_h(s, a)$. We then invoke Algorithm 1 with MDP $\hat{M}$ and tolerance parameter $r_{\text{action}}$, where $r_{\text{action}}$ is also drawn from the uniform distribution .

The formal guarantee of Algorithm 3 is summarized in the following theorem.

**Theorem F.1.** *Suppose $\mathbb{A}$ is an algorithm such that with probability at least $1 - \delta_0$, $\mathbb{A}$ returns an $\epsilon_0$-optimal policy. Then with probability at least $1 - \delta$, Algorithm 3 return a policy $\pi$, such that*

- *$\pi$ is $\epsilon$-optimal;*

- *$\pi \in \Pi(M)$, where $\Pi(M)$ is a list of policies that depend only on the unknown underlying MDP $M$ with size $|\Pi(M)| \leq (H|S||A| + 1)(H|S| + 1)$.*

In the remaining part of this section, we give the full proof of Theorem F.1.

Following the definition of $U_h(r)$ in Definition D.1, we define $T_h(r)$.

**Definition F.2.** *For the underlying MDP $M = (S, A, P, R, H, s_0)$, given a real number $r \in [0, 1]$, we define $T_h(r) \subseteq S$ for each $h \in [H]$ as follows:*

- *$T_0(r) = \{s \in S \mid \Pr[s_0 = s] \leq r\}$;*

- *$T_h(r) = \{s \in S \mid \max_\pi \Pr[s_h = s \mid M, \pi] \leq r\}$.*

*We also write $T(r) = (T_0(r), T_1(r), \ldots, T_{H-1}(r))$.*

**Lemma F.3.** *For all $r \in [0, 1]$, there are at most of $|S|H + 1$ unique sequences of sets $T(r)$.*

*Proof.* By the same analysis as in Lemma D.2 , we know that given $0 \leq r_1 \leq r_2 \leq 1$, for any $h \in [H]$, we have $T_h(r_1) \subseteq T_h(r_2)$. Moreover, by the same analysis as in Corollary D.3 , for all $r \in [0, 1]$, there are at most of $|S|H + 1$ unique sequences of sets $T(r)$.

$\square$

**Definition F.4.** *For the underlying MDP $M = (S, A, P, R, H, s_0)$, given a real number $r \in [0, 1]$, define $\overline{M}^r = (S \cup \{s_{\mathrm{absorb}}\}, A, \overline{P}^r, R, H, s_0)$, where*

$$\overline{P}^r_h(s' \mid s, a) = \begin{cases} P_h(s' \mid s, a) & s \notin T_h(r), s' \neq s_{\mathrm{absorb}} \\ 0 & s \notin T_h(r), s' = s_{\mathrm{absorb}} \\ \mathbb{1}[s' = s_{\mathrm{absorb}}] & s \in T_h(r) \cup \{s_{\mathrm{absorb}}\} \end{cases} \tag{6}$$

**Definition F.5.** *For each $(s, h) \in S \times [H]$, define $\mathrm{Crit}'(s, h) = \inf\{r \in [0, 1] \mid s \in T_h(r)\}$.*

Note that $\{r \in [0, 1] \mid s \in T_h(r)\}$ is never an empty set since $T_h(1) = S$.

**Lemma F.6.** *Consider a pair of fixed choices of $r_{\mathrm{trunc}}$ and $r_{\mathrm{action}}$ in Algorithm 3. For all $h \in [H - 1]$, if for all $s \in S \backslash \hat{T}_h$ we have $d_M^{\hat{\pi}^{s,h}} \geq \epsilon_1$ whenever $h > 0$, then with probability $1 - \frac{\delta}{4}$, for all $(s, a, h) \in (S \backslash \hat{T}_h) \times A \times [H-1]$,*

$$\sum_{s' \in S} |P_h(s' \mid s, a) - \hat{P}_h(s' \mid s, a)| \leq \epsilon_0.$$

*Proof.* By the same analysis as Lemma E.1, for a fixed $h \in [H - 1]$, if for all $s \in S \setminus \hat{T}_h$ we have $d_M^{\hat{\pi}^{s,h}} \geq \epsilon_1$ whenever $h > 0$, then with probability $1 - \frac{\delta}{4H}$, for all $(s, a) \in (S \setminus \hat{T}_h) \times A$,

$$\sum_{s' \in S} |P_h(s' \mid s, a) - \hat{P}_h(s' \mid s, a)| \leq \epsilon_0.$$

By union bound, we know that with probability $1 - \frac{\delta}{4}$, for all $h \in [H - 1]$, the inequality holds.

---

**Algorithm 3** Weakly $k$-list Replicable RL Algorithm

---

1: **Input:** RL algorithm $\mathbb{A}(\epsilon_0, \delta_0)$, error tolerance $\epsilon$, failure probability $\delta$

2: **Output:** near-optimal policy $\pi$

3: **Initialization:**

4: Initialize constants $C_1 = \frac{4|A||S|H}{\delta}$, $\epsilon_0 = \frac{\epsilon\delta}{100|S|H^5|A|}$, $\epsilon_1 = 5C_1H^2\epsilon_0$

5: Generate random numbers $r_{\text{action}} \sim \text{Unif}(\epsilon_1, 2\epsilon_1)$, $r_{\text{trunc}} \sim \text{Unif}(2\epsilon_1, 3\epsilon_1)$

6: **for** $h \in [H-1]$ **do**

7:    **for** each $s \in S$ **do**

8:       Invoke $\mathbb{A}$ with $\epsilon_0 = \epsilon_0$ and $\delta_0 = \delta/(8|S|H)$ on the underlying MDP with modified reward function $R_{h'}^{s,h}(s', a) = \mathbb{1}[h' = h, s' = s]$ for all $h' \in [H]$ and $(s', a) \in S \times A$

9:       Set $\hat{\pi}^{s,h}$ to be the policy returned in the previous step

10:       Collect $W = \frac{|S|^2}{\epsilon_0^2\epsilon_1} \log \frac{16|S|^2AH}{\delta}$ trajectories $\{(s_0^{(w)}, a_0^{(w)}, \ldots, s_{H-1}^{(w)}, a_{H-1}^{(w)})\}_{w=1}^W$ by executing $\hat{\pi}^{s,h}$ for $W$ times

11:       Set

$$\hat{d}(s, h) = \frac{\sum_{w=1}^W \mathbb{1}[s_h^{(w)} = s]}{W}$$

12:       **for** each $a \in A$ **do**

13:          Define policy $\hat{\pi}^{s,h,a}$, where for each $h' \in [H]$ and $s' \in S$,

$$\hat{\pi}_{h'}^{s,h,a}(s') = \begin{cases} a & h' = h, s' = s \\ \hat{\pi}_{h'}^{s,h}(s') & h' \neq h \text{ or } s' \neq s \end{cases}$$

14:          Collect $W = \frac{|S|^2}{\epsilon_0^2\epsilon_1} \log \frac{16|S|^2AH}{\delta}$ trajectories $\{(s_0^{(w)}, a_0^{(w)}, \ldots, s_{H-1}^{(w)}, a_{H-1}^{(w)})\}_{w=1}^W$ by executing $\hat{\pi}^{s,h,a}$ for $W$ times

15:          For each $s' \in S$, set

$$\hat{P}_h(s' \mid s, a) \leftarrow \frac{\sum_{w=1}^W \mathbb{1}[(s_h^{(w)}, a_h^{(w)}, s_{h+1}^{(w)}) = (s, a, s')]}{\sum_{w=1}^W \mathbb{1}[(s_h^{(w)}, a_h^{(w)}) = (s, a)]}$$

16:       **end for**

17:    **end for**

18: **end for**

19: For each $h \in [H-1]$, set $\hat{T}_h = \{s \in S \mid \hat{d}(s, h) \leq r_{\text{trunc}}\}$.

20: Define MDP $\hat{M} = (S \cup \{s_{\text{absorb}}\}, A, \tilde{P}, R, H, s_0)$, where for each $h \in [H-1]$,

$$\tilde{P}_h(s' \mid s, a) = \begin{cases} \hat{P}_h(s' \mid s, a) & s \notin \hat{T}_h \\ \mathbb{1}\{s' = s_{\text{absorb}}\} & s \in \hat{T}_h \end{cases}$$

21: Invoke Algorithm 1 with MDP $\hat{M}$ and tolerance parameter $r_{\text{action}}$, and set $\pi$ to be the returned policy

22: **return** $\pi$

---

$\square$

**Lemma F.7.** *With probability at least* $1 - \frac{\delta}{4}$*, for all* $s, h \in S \times [H-1]$*,*

$$|\hat{d}(s,h) - d_M^*(s,h)| \le 2\epsilon_0,$$
$$|d_M^{\hat{\pi}^{s,h}} - \hat{d}(s,h)| \le \epsilon_0.$$

*Proof.* For a specific pair $(s,h)$, for the policy returned by $\mathbb{A}$, with probability at least $1 - \frac{\delta}{8|S|H}$, we have

$$\left| d_M^*(s,h) - d_M^{\hat{\pi}^{s,h}}(s,h) \right| \le \epsilon_0.$$

Thus, by Chernoff bound, with probability at least $1 - \frac{\delta}{8|S|H}$, we have

$$\left| d_M^{\hat{\pi}^{s,h}}(s,h) - \hat{d}(s,h) \right| \le \epsilon_0.$$

Combining the above two inequalities, with probability at least $1 - \frac{\delta}{4|S|H}$,

$$|\hat{d}(s,h) - d_M^*(s,h)| \le 2\epsilon_0.$$

Using the union bound, we know that with probability at least $1 - \frac{\delta}{4}$, for all $s, h \in S \times [H-1]$

$$|\hat{d}(s,h) - d_M^*(s,h)| \le 2\epsilon_0,$$
$$|d_M^{\hat{\pi}^{s,h}} - \hat{d}(s,h)| \le \epsilon_0.$$

$\square$

**Definition F.8.** *Define*
$$\mathrm{Bad}'_{\mathrm{trunc}} = \bigcup_{(s,h) \in S \times [H]} \mathrm{Ball}(\mathrm{Crit}'(s,h), 2\epsilon_0),$$

*where* $\mathrm{Crit}'(s,h)$ *is as defined in Definition F.5.*

**Lemma F.9.** *Consider a pair of fixed choices of* $r_{\mathrm{trunc}} \in (\eta_1, 2\eta_1)$ *and* $r_{\mathrm{action}}$ *in Algorithm 2 such that* $r_{\mathrm{trunc}} \notin \mathrm{Bad}'_{\mathrm{trunc}}$*. With probability at least* $1 - \delta/2$*, we have*

- $\hat{T}_h = T_h(r_{\mathrm{trunc}})$ *for all* $h \in [H-1]$*;*

- $\sum_{s'} |\hat{P}_h(s' \mid s, a) - P_h(s' \mid s, a)| \le \epsilon_0$ *for all* $h \in [H-1]$ *and* $(s,a) \in (S \setminus \hat{T}_{h'}) \times A$*.*

*Proof.* Let $\mathcal{E}_1$ denote the event that for all $(s,h)$, the following two conditions hold:

- $|\hat{d}(s,h) - d_M^*(s,h)| \le 2\epsilon_0$

- $|d_M^{\hat{\pi}^{s,h}} - \hat{d}(s,h)| \le \epsilon_0$

By Lemma F.7, we know that with probability at least $1 - \frac{\delta}{4}$, event $\mathcal{E}_1$ occurs.

Let $\mathcal{E}_2$ denote the event that for all $(s,a,s',h) \in S \times A \times S \times [H-1]$, the following conditions are satisfied:

- $\hat{T}_h = T_h(r_{\text{trunc}})$;

- $d_M^{\hat{\pi}^{s,h}}(s,h) \geq \epsilon_1$ for all $s \in S \setminus \hat{T}_h$;

- $d_M^*(s,h) \leq 4\epsilon_1$ for all $s \in \hat{T}_h$;

- $\sum_{s' \in S} \left| \hat{P}_h(s' \mid s,a) - P_h(s' \mid s,a) \right| \leq \epsilon_0$ for all $(s,a) \in (S \setminus \hat{T}_h) \times A$.

When $\mathcal{E}_1$ occurs, we know that $|\hat{d}(s,h) - d_M^*(s,h)| \leq 2\epsilon_0$. Therefore, when $r_{\text{trunc}} \notin \text{Bad}_{\text{trunc}}$, if $r_{\text{trunc}} > d_M^*(s,h)$, it follows that $r_{\text{trunc}} > \hat{d}(s,h)$, if $r_{\text{trunc}} < d_M^*(s,h)$, it follows that $r_{\text{trunc}} < \hat{d}(s,h)$. Hence, we conclude that $\hat{T}_h = T_h(r_{\text{trunc}})$.

For the second condition, when $\mathcal{E}_1$ occurs, we know that $|d_M^{\hat{\pi}^{s,h}} - \hat{d}(s,h)| \leq \epsilon_0$, and by definition, $\hat{d}(s,h) > 2\epsilon_1$. Thus, we obtain that

$$d_M^{\hat{\pi}^{s,h}} > 2\epsilon_1 - \epsilon_0 > \epsilon_1.$$

For the third condition, when $\mathcal{E}_1$ occurs, we know that $|\hat{d}(s,h) - d_M^*(s,h)| \leq 2\epsilon_0$, and by definition, $\hat{d}(s,h) < 3\epsilon_1$. Thus, we have

$$d_M^*(s,h) < 3\epsilon_1 + 2\epsilon_0 < 4\epsilon_1.$$

For the forth condition, combining the second condition with Lemma F.6, we conclude that with probability at least $\left(1 - \frac{\delta}{4}\right)^2 \leq 1 - \frac{\delta}{2}$, the fourth condition holds.

Therefore, with probability at least $1 - \frac{\delta}{2}$, event $\mathcal{E}_2$ occurs, which implies the desired result.

$\square$

**Definition F.10.** *For a real number $r \in [0,1]$, define*

$$\text{Bad}'_{\text{action}}(r) = \bigcup_{g \in \text{Gap}_{\overline{M}^r}} \text{Ball}(g, 2H^2\epsilon_0).$$

Clearly, for any $r \in [0,1]$, $|\text{Gap}(r)| \leq |S|HA$. Moreover, since $\overline{M}^r$ depends only on $T(r)$ (cf. Definition F.4), for $r_1, r_2 \in [0,1]$ with $T(r_1) = T(r_2)$, we would have $\text{Gap}(r_1) = \text{Gap}(r_2)$ and $\text{Bad}'_{\text{action}}(r_1) = \text{Bad}'_{\text{action}}(r_2)$.

**Lemma F.11.** *Given $r_{\text{trunc}}^1, r_{\text{trunc}}^2 \in (2\epsilon_1, 3\epsilon_1) \setminus \text{Bad}_{\text{trunc}}$ and $r_{\text{action}}^1, r_{\text{action}}^2 \in (\epsilon_1, 2\epsilon_1)$, suppose*

- $T(r_{\text{trunc}}^1) = T(r_{\text{trunc}}^2)$;

- $r_{\text{action}}^1 \notin \text{Bad}'_{\text{action}}(r_{\text{trunc}}^1)$, and $r_{\text{action}}^2 \notin \text{Bad}'_{\text{action}}(r_{\text{trunc}}^1)$;

- *for any $g \in \text{Gap}(r_{\text{trunc}}^1)$, either $g < r_{\text{action}}^1 < r_{\text{action}}^2$ or $r_{\text{action}}^1 < r_{\text{action}}^2 < g$,*

*conditioned on the event in Lemma F.9, the returned policy $\pi$ in Algorithm 3 will always be the same for all $(r_{\text{action}}, r_{\text{trunc}}) \in \{r_{\text{action}}^1, r_{\text{action}}^2\} \times \{r_{\text{trunc}}^1, r_{\text{trunc}}^2\}$.*

*Proof.* The proof of the lemma follows the same reasoning as in the proof of Lemma E.8. $\square$

**Lemma F.12.** *Conditioned on the event in Lemma F.9, the returned policy $\pi$ is $\epsilon$-optimal.*

*Proof.*

$$V_M^\pi \geq V_M^{\pi r_{\text{trunc}}} \geq V_M^{* r_{\text{trunc}}} - 2H^2\epsilon_0 - r_{\text{action}}H \geq V_M^* - 2H^2\epsilon_0 - r_{\text{action}}H - H^2|S|r_{\text{trunc}} \geq V_M^* - \epsilon.$$

where the first inequality is due to Lemma G.3, the second inequality is due to Lemma 5.1, the third inequality is due to Lemma G.3, and the last inequality is due to $r_{\text{trunc}} \leq 3\epsilon_1$ and $r_{\text{action}} \leq 2\epsilon_1$. Therefore, the returned policy $\pi$ is $\epsilon$-optimal. $\square$

**Lemma F.13.** *Conditioned on the event in Lemma F.9, with probability at least $1 - \frac{\delta}{2}$, the returned policy $\pi$ belongs to the set $\Pi(M)$, where $\Pi(M)$ is a list of policies that depend only on the unknown underlying MDP $M$, and the size of $\Pi(M)$ satisfies $|\Pi(M)| \leq (H|S||A| + 1)(H|S| + 1)$.*

*Proof.* First, we have $\Pr[r_{\text{trunc}} \in \text{Bad}'_{\text{trunc}}] \leq \frac{5|S|H\epsilon_0}{\epsilon_1} < \frac{\delta}{4}$. Moreover, for a fixed $r_{\text{trunc}} \notin \text{Bad}'_{\text{trunc}}$, we have $\Pr[r_{\text{action}} \in \text{Bad}'_{\text{action}}(r_{\text{trunc}})] \leq \frac{5H^2\epsilon_0 * |S||A|H}{\epsilon_1} < \frac{\delta}{4}$. Thus, with probability at least $1 - \frac{\delta}{2}$, it is satisfied that $r_{\text{action}} \notin \text{Bad}'_{\text{action}}(r_{\text{trunc}})$ and $r_{\text{trunc}} \notin \text{Bad}'_{\text{trunc}}$.

By Lemma F.11, and applying similar reasoning as in the proof of Theorem 6.1, we conclude that conditioned on the event in Lemma F.9, with probability at least $1 - \frac{\delta}{2}$, the policy $\pi$ belongs to the set $\Pi(M)$, where $\Pi(M)$ is a list of policies that depend only on the unknown underlying MDP $M$. Moreover, the size of $\Pi(M)$ is bounded by $|\Pi(M)| \leq (H|S||A| + 1)(H|S| + 1)$. $\square$

*Proof of Theorem F.1.* The proof follows by combining Lemma F.9, Lemma F.12 and Lemma F.13 $\square$

# G  PERTURBATION ANALYSIS IN MDPS

**Lemma G.1.** *Consider two MDP $M_1$ and $M_2$ that are $\epsilon_0$-related. Let $P'$ and $P''$ denote the transition models of $M_1$ and $M_2$, respectively. It holds that*

$$\left|V_{h,M_1}^*(s) - V_{h,M_2}^*(s)\right| \leq H^2\epsilon_0,$$

$$\left|Q_{h,M_1}^*(s,a) - Q_{h,M_2}^*(s,a)\right| \leq H^2\epsilon_0,$$

*where $H$ is the horizon length.*

*Specifically, for the value function at the initial state $s_0$, it holds that*

$$\left|V_{M_1}^* - V_{M_2}^*\right| \leq H^2\epsilon_0.$$

*Proof.* We denote $\pi_1^*$ as the optimal policy of $M_1$ and $\pi_2^*$ as the optimal policy of $M_2$. For $0 \leq i \leq H-1$, we have

$$
\left| V_{i,M_1}^{\pi_1^*}(s) - V_{i,M_2}^{\pi_2^*}(s) \right| \overset{(1)}{\leq} \max_a \left| Q_{i,M_1}^{\pi_1^*}(s,a) - Q_{i,M_2}^{\pi_2^*}(s,a) \right|
$$

$$
\leq \max_a \left( \left| \sum_{s'} P_i'(s' \mid s,a) \cdot V_{i+1,M_1}^{\pi_1^*}(s') - \sum_{s'} P_i''(s' \mid s,a) \cdot V_{i+1,M_2}^{\pi_2^*}(s') \right| \right)
$$

$$
\leq \max_a \left( \left| \sum_{s'} P_i'(s' \mid s,a) \cdot \left( V_{i+1,M_1}^{\pi_1^*}(s') - V_{i+1,M_2}^{\pi_2^*}(s') \right) \right| \right.
$$

$$
\left. + \left| \sum_{s'} \left( P_i'(s' \mid s,a) - P_i''(s' \mid s,a) \right) \cdot V_{i+1,M_2}^{\pi_2^*}(s') \right| \right)
$$

$$
\overset{(2)}{\leq} H\epsilon_0 + \max_s \left| V_{i+1,M_1}^{\pi_1^*}(s) - V_{i+1,M_2}^{\pi_2^*}(s) \right|.
$$

**Inequality (1):** This follows from selecting $a^*$ as the optimal action and $\hat{a}$ as the action selected by the policy, which ensures $Q_{i,M_2}^{\pi_1^*}(s,a) \leq Q_{i,M_2}^{\pi_2^*}(s,a)$.

**Inequality (2):** This holds because $V_{i+1}^{\pi^*}(s') \leq H$, the total variation bound $\sum_{s' \in S} |P_i'(s' \mid s,a) - P_i''(s' \mid s,a)| \leq \epsilon_0$, and the fact that $\sum_{s'} P_i'(s' \mid s,a) = 1$.

At layer $H$, it is given that $V_{H,M_1}^{\pi_1^*} = V_{H,M_2}^{\pi_2^*} = 0$. Applying the above inequality recursively, we obtain

$$
\left| V_{i,M_1}^{\pi_1^*}(s) - V_{i,M_2}^{\pi_2^*}(s) \right| \leq H(H-i)\epsilon_0 \leq H^2\epsilon_0,
$$

$$
\left| Q_{i,M_1}^{\pi_1^*}(s,a) - Q_{i,M_2}^{\pi_2^*}(s,a) \right| \leq H\epsilon_0 + \max_s \left| V_{i+1,M_1}^{\pi_1^*}(s) - V_{i+1,M_2}^{\pi_2^*}(s) \right| \leq H\epsilon_0 + H(H-1)\epsilon_0 \leq H^2\epsilon_0.
$$

In particular, for the initial layer,

$$
\left| V_{M_1}^* - V_{M_2}^* \right| = \left| V_{0,M_1}^{\pi_1^*}(s_0) - V_{0,M_2}^{\pi_2^*}(s_0) \right| \leq H^2\epsilon_0.
$$

$\square$

**Lemma G.2.** *Consider two MDP $M_1$ and $M_2$ that are $\epsilon_0$-related . Let $P'$ and $P''$ denote the transition models of $M_1$ and $M_2$, respectively. For any policy $\pi$, it holds that*

$$
\left| V_{M_1}^\pi - V_{M_2}^\pi \right| \leq H^2\epsilon_0,
$$

*where $H$ is the horizon length.*

*Proof.* For $0 \le i \le H - 1$, we have

$$\left|V_{i,M_1}^\pi(s) - V_{i,M_2}^\pi(s)\right| = \left|Q_{i,M_1}^\pi(s, \pi_i(s)) - Q_{i,M_2}^\pi(s, \pi_i(s))\right|$$

$$\le \max_a \left( \left| \sum_{s'} P_i'(s' \mid s, a) \cdot V_{i+1,M_1}^\pi(s') - \sum_{s'} P_i''(s' \mid s, a) \cdot V_{i+1,M_2}^\pi(s') \right| \right)$$

$$\le \max_a \left( \left| \sum_{s'} P_i'(s' \mid s, a) \cdot \left(V_{i+1,M_1}^\pi(s') - V_{i+1,M_2}^\pi(s')\right) \right| \right.$$

$$\left. + \left| \sum_{s'} \left(P_i'(s' \mid s, a) - P_i''(s' \mid s, a)\right) \cdot V_{i+1,M_2}^\pi(s') \right| \right)$$

$$\overset{(1)}{\le} H\epsilon_0 + \max_s \left|V_{i+1,M_1}^\pi(s) - V_{i+1,M_2}^\pi(s)\right|.$$

**Inequality (1):** This holds because $V_{i+1}^{\pi^*}(s') \le H$, the total variation bound $\sum_{s' \in S} |P_i'(s' \mid s, a) - P_i''(s' \mid s, a)| \le \epsilon_0$, and the fact that $\sum_{s'} \hat{P}_i(s' \mid s, a) = 1$.

At layer $H$, it is given that $V_{H,M_1}^\pi = V_{H,M_2}^\pi = 0$.

Applying the above inequality recursively, we obtain

$$\left|V_{i,M_1}^\pi(s) - V_{i,M_2}^\pi(s)\right| \le H^2 \epsilon_0,$$

In particular, for the initial layer,

$$\left|V_{0,M_1}^\pi(s_0) - V_{0,M_2}^\pi(s_0)\right| \le H^2 \epsilon_0,$$

$\square$

**Lemma G.3.** *For any policy $\pi$, we have*

$$0 \le V_M^\pi - V_{M^r}^\pi \le H^2 |S| r,$$

*where $M^r$ is defined as in Definition D.6 and $|S|$ is the size of the state space.*

*Proof.* Clearly, $V_M^\pi - V_{M^r}^\pi \ge 0$.

We observe that for any $h$ and $s_h \in S$, the following holds:

$$\sum_{s_h \in S} d_{M^r}^\pi(s_h, h) \left( V_{h,M}^\pi(s_h) - V_{h,M^r}^\pi(s_h) \right)$$

$$\overset{(1)}{=} \sum_{s_h \in U_h(r)} d_{M^r}^\pi(s_h, h) V_{h,M}^\pi(s_h) + \sum_{s_h \notin U_h(r)} d_{M^r}^\pi(s_h, h) \left( V_{h,M}^\pi(s_h) - V_{h,M^r}^\pi(s_h) \right)$$

$$\overset{(2)}{\leq} |S| \cdot r \cdot H + \sum_{s_h \notin U_h(r)} d_{M^r}^\pi(s_h, h) \left( V_{h,M}^\pi(s_h) - V_{h,M^r}^\pi(s_h) \right)$$

$$\overset{(3)}{=} |S| \cdot r \cdot H + \sum_{s_h \notin U_h(r)} d_{M^r}^\pi(s_0, h) \left( r_h(s_h, \pi(s_h)) + \sum_{s_{h+1} \in S} P_h(s_{h+1}|s_h, \pi(s_h)) V_{h+1,M}^\pi(s_{h+1}) \right.$$

$$\left. - r_h(s_h, \pi(s_h)) - \sum_{s_{h+1} \in S} P_h(s_{h+1}|s_h, \pi(s_h)) V_{h+1,M^r}^\pi(s_{h+1}) \right)$$

$$= |S| \cdot r \cdot H + \sum_{s_h \notin U_h(r)} d_{M^r}^\pi(s_{h+1}, h+1) \left( V_{h+1,M}^\pi(s_{h+1}) - V_{h+1,M^r}^\pi(s_{h+1}) \right)$$

$$\overset{(4)}{=} |S| \cdot r \cdot H + \sum_{s_{h+1} \in S} d_{M^r}^\pi(s_{h+1}, h+1) \left( V_{h+1,M}^\pi(s_{h+1}) - V_{h+1,M^r}^\pi(s_{h+1}) \right)$$

- **Step (1):** The first equality arises because for all $s_h \in U_h(r)$, the value function $V_{h,M^r}^\pi(s_h) = 0$.

- **Step (2):** The inequality follows from the definition of $d_{M^r}^\pi(s_h, h) \leq r$ and the fact that $V_{h,M}^\pi(s_h) \leq H$. This ensures that the first term in the sum is bounded by $|S| \cdot r \cdot H$.

- **Step (3):** The equality holds because for all $s_h \notin U_h(r)$, the transition probability $P_h(s_{h+1}|s_h, \pi(s_h))$ under the original model $M$ is identical to that under the modified model $M^r$, i.e., $P_h(s_{h+1}|s_h, \pi(s_h)) = P_h^r(s_{h+1}|s_h, \pi(s_h))$. Thus, the only difference in the value functions is the difference in the values at the next time step.

- **Step (4):** The final equality follows from interchanging the order of summation, allowing us to express the sum over $s_h$ as a sum over $s_{h+1}$.

Next, we observe that

$$V_{0,M}^\pi(s_0) - V_{0,M^r}^\pi(s_0) \overset{(5)}{=} \sum_{s_1 \in S} d_{M^r}^\pi(s_1, 1) \left( V_{1,M}^\pi(s_1) - V_{1,M^r}^\pi(s_1) \right),$$

where **Step (5):** holds because $s_0$ is the fixed initial state, and by definition, $d_{M^r}^\pi(s_1, 1) = d_M^\pi(s_1, 1) = P_0(s_1|s_0, \pi(s_0))$.

By recursively applying the same reasoning for each time step $h$, we obtain the following upper bound:

$$V_{0,M}^\pi(s_0) - V_{0,M^r}^\pi(s_0) \leq |S| \cdot r \cdot H^2.$$

Thus, we conclude that

$$0 \leq V_M^\pi - V_{M^r}^\pi \leq H^2 |S| r.$$

$\square$

**Lemma G.4.** *For any policy $\pi$, we have*

$$0 \le V_M^\pi - V_{\overline{M}^r}^\pi \le H^2 |S| r,$$

*where $\overline{M}^r$ is defined as in Definition F.4 and $|S|$ is the size of the state space.*

*Proof.* Clearly, $V_M^\pi - V_{\overline{M}^r}^\pi \ge 0$.

By the similar analysis as above, we observe that for any $h$ and $s_h \in S$, the following holds:

$$\sum_{s_h \in S} d_{M^r}^\pi(s_h, h) \left( V_{h,M}^\pi(s_h) - V_{h,\overline{M}^r}^\pi(s_h) \right)$$

$$= \sum_{s_h \in T_h(r)} d_{\overline{M}^r}^\pi(s_h, h) V_{h,M}^\pi(s_h) + \sum_{s_h \notin T_h(r)} d_{\overline{M}^r}^\pi(s_h, h) \left( V_{h,M}^\pi(s_h) - V_{h,\overline{M}^r}^\pi(s_h) \right)$$

$$\overset{(1)}{\le} |S| \cdot r \cdot H + \sum_{s_h \notin T_h(r)} d_{\overline{M}^r}^\pi(s_h, h) \left( V_{h,M}^\pi(s_h) - V_{h,\overline{M}^r}^\pi(s_h) \right)$$

$$= |S| \cdot r \cdot H + \sum_{s_h \notin T_h(r)} d_{\overline{M}^r}^\pi(s_0, h) \left( r_h(s_h, \pi(s_h)) + \sum_{s_{h+1} \in S} P_h(s_{h+1}|s_h, \pi(s_h)) V_{h+1,M}^\pi(s_{h+1}) \right.$$

$$\left. - r_h(s_h, \pi(s_h)) - \sum_{s_{h+1} \in S} P_h(s_{h+1}|s_h, \pi(s_h)) V_{h+1,\overline{M}^r}^\pi(s_{h+1}) \right)$$

$$= |S| \cdot r \cdot H + \sum_{s_{h+1} \in S} d_{\overline{M}^r}^\pi(s_{h+1}, h+1) \left( V_{h+1,M}^\pi(s_{h+1}) - V_{h+1,\overline{M}^r}^\pi(s_{h+1}) \right)$$

- **Step (1):** The inequality follows from the definition of $d_{\overline{M}^r}^\pi(s_h, h) \le \max_\pi \Pr[s_h = s \mid M, \pi] \le r$ and the fact that $V_{h,M}^\pi(s_h) \le H$. This ensures that the first term in the sum is bounded by $|S| \cdot r \cdot H$.

Next, we observe that

$$V_{0,M}^\pi(s_0) - V_{0,\overline{M}^r}^\pi(s_0) = \sum_{s_1 \in S} d_{\overline{M}^r}^\pi(s_1, 1) \left( V_{1,M}^\pi(s_1) - V_{1,\overline{M}^r}^\pi(s_1) \right),$$

By recursively applying the same reasoning for each time step $h$, we obtain the following upper bound:

$$V_{0,M}^\pi(s_0) - V_{0,\overline{M}^r}^\pi(s_0) \le |S| \cdot r \cdot H^2.$$

Thus, we conclude that

$$0 \le V_M^\pi - V_{\overline{M}^r}^\pi \le H^2 |S| r.$$

$\square$

# H  HARDNESS RESULT

**Definition H.1** (BESTARM Problem). *Consider a $k$-armed bandit problem. Let $k$ be the number of arms, and fix parameters $\epsilon > 0$ and $\delta \in (0, 1)$. The $(k, \epsilon, \delta)$-BESTARM problem is defined as follows: given access to $k$ arms,*

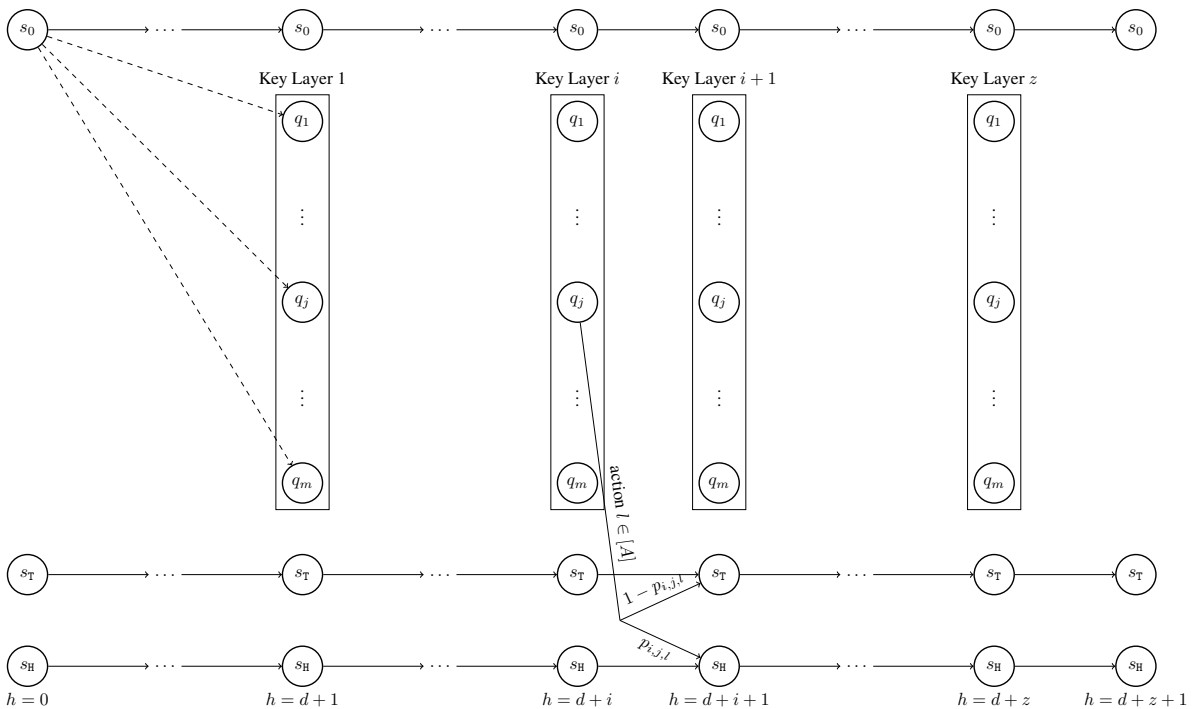

Figure 10: MDP to solve BESTARM.

*each associated with an unknown distribution (e.g., Bernoulli), the goal for an algorithm is to identify an arm whose mean reward is within $\epsilon$ of the best arms mean, with probability at least $1 - \delta$.*

**Lemma H.2** ((Chen et al., 2025)). *Consider a $k$-armed bandit problem. Let $\epsilon \leq \frac{1}{2k}$ and $\delta \leq \frac{1}{k+1}$. Then, there exists no $(k-1)$-list replicable algorithm for the $(k, \epsilon, \delta)$-BESTARM problem, even when each arm follows a Bernoulli distribution and an unbounded number of samples is allowed.*

**Theorem H.3.** *Suppose there exists a weakly $\ell$-list replicable RL algorithm that interacts with an MDP $M$ with state space $S$, action space $A$, and horizon length $H$, such that there is a list of policies $\Pi(M)$ with cardinality at most $\ell$ that depend only on $M$, so that with probability at least $1 - \delta$, $\pi$ is $\epsilon$-optimal and $\pi \in \Pi(M)$, where $\pi$ is the near-optimal policy returned by the algorithm when interacting with $M$. Suppose $\epsilon \leq \frac{1}{2|S||A|H}$ and $\delta \leq \frac{1}{|S||A|H+1}$. Then it must hold that*

$$\ell \geq \frac{|S||A|\left(H - \lceil \log_{|A|} |S| \rceil - 3\right)}{3}.$$

*Proof.* Assume for contradiction that there exists an RL algorithm that satisfies the conditions of the theorem, with

$$\ell < \frac{|S||A|\left(H - \lceil \log_{|A|} |S| \rceil - 3\right)}{3}.$$

We will show that this assumption leads to a contradiction with Lemma H.2.

Without loss of generality, assume $|S|$ is divisible by 3. Let $m = |S|/3$, $n = |A|$, $z = H - \lceil \log_n m \rceil - 3$, and define $k = mnz$. We now construct a reduction from the $k$-armed bandit problem (with Bernoulli rewards) to an MDP instance.

We index the $k$ arms by triplets $(i, j, \ell)$, where $i \in [z]$, $j \in [m]$, and $\ell \in [n]$. Each arm is associated with a Bernoulli distribution $D_{i,j,\ell}$ with mean $p_{i,j,\ell}$. We will design an MDP $M$ such that interacting with it corresponds to querying these $k$ arms.

**Key Layer Construction.** Let $\{q_1, \ldots, q_m\} \subset S$ denote a set of $m$ designated *key-layer* states (illustrated in Figure 10). We will construct the MDP such that for each $i \in [z]$ and $j \in [m]$, there exists a unique deterministic policy that reaches state $q_j$ precisely at time step $h_i = d + i$, where $d = \lceil \log_n m \rceil$.

Once in state $q_j$ at time $h_i$, the agent can choose action $a_\ell \in A$ to simulate pulling arm $(i, j, \ell)$. Let $s_H, s_T \in S$ denote two absorbing states. We define

$$\forall (i, j, \ell), \quad P_{h_i}(s_H \mid q_j, a_\ell) = p_{i,j,\ell}, \quad P_{h_i}(s_T \mid q_j, a_\ell) = 1 - p_{i,j,\ell}.$$

and for all $h, a$: $r_h(s_H, a) = \mathbb{1}[h = H - 1]$ and $r_h(s_T, a) = 0$. Both $s_H$ and $s_T$ are absorbing: $P(s' \mid s_H, a) = \mathbb{1}[s' = s_H]$ and similarly for $s_T$.

**Auxiliary Structure.** We now describe the deterministic routing structure that reaches each $q_j$ in exactly $d$ steps. We construct a complete $n$-ary tree rooted at a state $w_1 \in S$. Every non-leaf state in the tree has $n$ children, one for each action in $A$, and transitions deterministically based on the action played.

The final layer connects to key-layer states $q_1, \ldots, q_m$. There may be more than $m$ leaf actions; any excess actions simply self-loop. The tree has depth $d$, requires at most $2m$ states, and all transitions have reward zero. Transitions are time-homogeneous.

**Initial State and Entry Mechanism.** Let $s_0 \in S$ be the initial state. Define its transitions as follows:

1. Playing a designated action $a_0 \in A$ transitions to the root $w_1$ of the $n$-ary tree;

2. Playing a designated action $a_1 \in A$ causes the agent to remain in $s_0$;

3. All other actions lead to $s_T$.

To reach a key-layer state $q_j$ at time $h_i = d + i$, a policy selects $a_1$ for $i$ time steps in $s_0$, followed by action $a_0$ to enter the tree, and then a sequence of $d$ actions that leads to $q_j$. From there, it plays $a_\ell$ to simulate arm $(i, j, \ell)$.

**Correctness of the Reduction.** This construction yields a one-to-one correspondence between bandit arms and deterministic policies in the MDP that reach $q_j$ at $h_i$ and play $a_\ell$. Thus, any $\epsilon$-optimal policy in the MDP induces an $\epsilon$-optimal arm in the bandit problem. Note also that all non-rewarding policies cannot match the optimal value due to the delayed structure and reward placement.

**Contradiction.** Now suppose we run the assumed RL algorithm on this MDP. By hypothesis, the algorithm returns a $\epsilon$-optimal policy that lies in a list of $\ell$ policies with $\ell < k = mnz$, with probability at least $1 - \delta$, where $\epsilon \leq \frac{1}{2k}$ and $\delta \leq \frac{1}{k+1}$. Since each policy corresponds to a unique arm, this implies the existence of a $(k-1)$-list replicable algorithm for the $(k, \epsilon, \delta)$-BESTARM problem. This contradicts Lemma H.2, completing the proof. $\qquad\square$

# I EXPERIMENTS OF MORE COMPLEX ENVIRONMENT

All our experiments are performed based on environments in the Gymnasium (Towers et al., 2024) package, and we use the PyTorch 2.1.2 for training neural networks. We use fixed random seeds in our experiments for better reproducibility.

## I.1 CARTPOLE-V1 WITH DQN

We evaluate the performance of the DQN algorithm (Mnih et al., 2015) on `CartPole-v1`, where we replace the planning algorithm with our robust planner (Algorithm 1) in Section 5.

**Network Architecture:**

We use a feedforward neural network to approximate the Q-function.

- Input layer: 4-dimensional state vector

- Hidden layer 1: Fully connected, 64 units, ReLU

- Hidden layer 2: Fully connected, 64 units, ReLU

- Output layer: Fully connected, 2 units (Q-values)

**Experience Replay:**

- Buffer capacity: $10^5$ transitions stored in a FIFO deque

- Batch size: $B = 256$

- Learning begins once buffer size $\geq B$

**Target Network Updates:**

- Two networks: local ($\theta$) and target ($\theta^-$)

- We use soft target updates to stabilize learning. After every Q-network update (which occurs every step once the buffer contains $\geq 256$ transitions), the target network parameters are softly updated using $\theta_{\text{target}} \leftarrow \tau \theta_{\text{online}} + (1 - \tau)\theta_{\text{target}}$ with $\tau = 0.001$.

**Hyperparameters:**

| Parameter | Symbol | Value(s) | Description |
|---|---|---|---|
| Learning rate | $\alpha$ | $2.5 \times 10^{-3}$ | Adam optimizer step size |
| Discount factor | $\gamma$ | 0.99 | Future reward discount |
| Replay batch size | $B$ | 256 | Transitions per learning update |
| Replay buffer capacity | $N$ | $10^5$ | Max number of stored transitions |
| Soft update factor | $\tau$ | $10^{-3}$ | Target network mixing coefficient |
| Exploration start | $\epsilon_0$ | 1.0 | Initial exploration probability |
| Exploration end | $\epsilon_{\min}$ | 0.01 | Minimum exploration probability |
| Exploration decay | $\epsilon_{\text{decay}}$ | 0.997 | Multiplicative decay per episode |
| Training episodes | – | 400 | Total training episodes |
| Max steps per episode | – | 500 | Episode length limit |
| Evaluation episodes | – | 100 | Used to compute mean returns |
| Independent runs | – | 50 | Used to report mean/std |

**Training Procedure:**

1. Initialize local and target networks; create empty replay buffer.

2. For each episode:
   - Reset environment; compute $\epsilon_t = \max(\epsilon_{\min}, \epsilon_0 \cdot \epsilon_{\text{decay}}^t)$
   - For each step $t$:
     - Select action using $\epsilon$-greedy or Algorithm 1
     - Store transition $(s, a, r, s')$ in the replay buffer
     - If buffer size $\geq B$, sample mini-batch and update Q-network
     - Update target network using soft update rule

When invoking Algorithm 1, we use the Q-network as our estimate of $Q^*_{h,\hat{M}}$, and select actions using Algorithm 1 with $r_{\text{action}} \in \{0.0, 0.05, 0.1, 0.5\}$. Note that when $r_{\text{action}} = 0$, Algorithm 1 is equivalent to picking actions that maximize the estimated $Q$-value as in the original DQN algorithm.

**Evaluation Protocol:**

Every 10 training episodes, we evaluate the policy over 100 test episodes, where each episode is initialized using a fixed random seed for reproducibility. During the evaluation, we disable $\epsilon$-greedy but still use Algorithm 1 to choose actions. In Figure 1(a), we report the average award of the trained policy, $\pm$ standard deviation, across different runs.

## I.2 ACROBOT-V1 WITH DOUBLE DQN

We evaluate the performance of the Double DQN algorithm (Van Hasselt et al., 2016) on `Acrobot-v1`, where we replace the planning algorithm with our robust planner (Algorithm 1) in Section 5.

**Network Architecture:** We use a feedforward neural network to approximate the Q-function.

- Input layer: state vector ($\dim = 6$)
- Hidden layers: 256  512  512 units, ReLU activations
- Output layer: Q-values for each action ($\dim = 3$)

**Hyperparameters:**

| Parameter | Symbol | Value(s) | Description |
|---|---|---|---|
| Learning rate | $\alpha$ | $1 \times 10^{-5}$ | Adam step size |
| Discount factor | $\gamma$ | 0.99 | Future reward discount |
| Batch size | $B$ | 8192 | Samples per update |
| Replay capacity | $N$ | $5 \times 10^4$ | Max transitions stored |
| Target update freq. | – | 100 steps | Hard copy interval |
| Initial $\varepsilon$ | $\varepsilon_0$ | 1.0 | Exploration start |
| Min $\varepsilon$ | $\varepsilon_{\min}$ | 0.01 | Exploration floor |
| $\varepsilon$-decay | $\delta$ | $5 \times 10^{-4}$ | Exploration decay per episode |
| Training epochs | – | 90 | Total learning epochs |
| Eval interval | – | 10 episodes | Test frequency |
| Eval episodes | – | 100 runs | Used to compute mean returns |
| Independent runs | – | 25 | Used to report mean/std |

**Replay Buffer:**

- Capacity: 50,000 transitions

- Batch size: $B = 8192$

**Training Procedure:**

1. Initialize networks, replay buffer, and seeds.

2. For each episode $t$:

    - Reset environment; compute $\varepsilon_t = \max(\varepsilon_{\min}, \varepsilon_0 - t\delta)$
    - For each step:
        - Select action using $\epsilon$-greedy or Algorithm 1
        - Store transition $(s, a, r, s')$ in the replay buffer.
        - If buffer size $\geq B$, sample mini-batch and update Q-network using double Q-learning
        - Every 100 learning steps, replace target weights

When invoking Algorithm 1, we use the Q-network as our estimate of $Q^*_{h,\hat{M}}$, and select actions using Algorithm 1 with $r_{\text{action}} \in \{0, 0.05, 0.1, 0.2\}$. Note that when $r_{\text{action}} = 0$, Algorithm 1 is equivalent to picking actions that maximize the estimated $Q$-value as in the original Double DQN algorithm.

**Evaluation Protocol:**

Same as Section I.1.

I.3   MOUNTAINCAR-V0 WITH TABULAR Q-LEARNING

We evaluate the performance of the Q-Learning on `MountainCar-v0`, where we replace the planning algorithm with our robust planner (Algorithm 1) in Section 5.

**State Discretization:**

- Discretized into a $20 \times 20$ grid

- Bin size computed from environment bounds

- Discrete state: `tuple`$((s - s_{\min})/\Delta s)$

**Q-table:**

- Shape: $(20, 20, 3)$

- Initialized uniformly in $[-2, 0]$

**Hyperparameters:**

| Parameter | Symbol | Value(s) | Description |
|---|---|---|---|
| Learning rate | $\alpha$ | 0.1 | Q-learning update step |
| Discount factor | $\gamma$ | 0.95 | Discount for future rewards |
| Exploration schedule | $\epsilon$ | $\max(0.01, 1 - t/500)$ | Episode-based decay |
| State bins | – | $20 \times 20$ | For discretization |
| Training episodes | – | 10,000 | Total learning episodes |
| Evaluation interval | – | 200 | Test policy every 200 episodes |
| Test episodes | – | 100 | Used to compute mean returns |
| Independent runs | – | 25 | Used to report mean/std |

**Training Procedure:**

For each episode $t$:

- Reset environment; discretize initial state; compute $\epsilon_t = \max(0.01, 1 - t/500)$

- Select actions using $\epsilon$-greedy or Algorithm 1

- Update Q-table with learning rate $\alpha = 0.1$ and discount factor $\gamma = 0.95$:

$$Q(s, a) \leftarrow (1 - \alpha)Q(s, a) + \alpha \left[ r + \gamma \max_{a'} Q(s', a') \right]$$

- If terminal state is reached and the goal is achieved, set $Q(s, a) \leftarrow 0$

When invoking Algorithm 1, we use the Q-table as our estimate of $Q^*_{h, \hat{M}}$, and select actions using Algorithm 1 with $r_{\text{action}} \in \{0, 0.001, 0.005, 0.02\}$. Note that when $r_{\text{action}} = 0$, Algorithm 1 is equivalent to picking actions that maximize the estimated $Q$-value as in the original Q-learning algorithm.

**Evaluation Protocol:** Same as Section I.1.

I.4 NAMETHISGAME WITH BEYOND THE RAINBOW

We evaluate the performance of the Beyond The Rainbow on `Namethisgame`, where we replace the planning algorithm with our robust planner (Algorithm 1) in Section 5.

**Environment:**

- Domain: Atari 2600, evaluated on `NameThisGame`

- Simulator: ALE with frame skip $= 4$

- Observations: grayscale $84 \times 84$ stacked frames

- Actions: discrete Atari action set

**Baseline:**

- Algorithm: BTR (Bootstrapped Transformer Reinforcement learning)

- Training budget: 100M Atari frames

**Threshold Strategy:**

- Planner augmented with a decaying action-threshold rule

- At each decision point, we select
$$a = \arg\max_{a'} Q(s, a') \quad \text{subject to} \quad Q(s, a) \geq \max_{a'} Q(s, a') - r_{\text{action}}(t),$$
where $r_{\text{action}}(t)$ is a step-dependent threshold

- Decay schedule:
$$r_{\text{action}}(t) = 0.4 \times (0.98)^{\lfloor t/5000 \rfloor},$$
with $t$ denoting the training step index

- When $r_{\text{action}}(t) \rightarrow 0$, the method reduces to the vanilla BTR algorithm

| Parameter | Symbol | Value(s) | Description |
|---|---|---|---|
| Learning rate | $lr$ | $1 \times 10^{-4}$ | Optimizer step size (Adam/AdamW) |
| Discount factor | $\gamma$ | 0.997 | Discount for future rewards |
| Batch size | $B$ | 256 | Mini-batch size for updates |
| Replay buffer size | – | $10^6$ | PER capacity |
| PER coefficient | $\alpha$ | 0.2 | Priority exponent |
| PER annealing | $\beta$ | $0.45 \rightarrow 1.0$ | Importance weight schedule |
| Gradient clipping | – | 10.0 | Norm clipping for stability |
| Target update | – | 500 steps | Replace target network |
| Slow net update | – | 5000 steps | Replace slow network |
| Optimizer | – | Adam/AdamW | With $\epsilon = 0.005/B$ |
| Loss function | – | Huber | Temporal difference loss |
| Replay ratio | – | 1.0 | Grad updates per env step |
| Exploration schedule | $\epsilon$ | $1.0 \rightarrow 0.01$ (2M steps) | $\epsilon$-greedy decay |
| Noisy layers | – | Enabled | Factorized Gaussian noise |
| Network arch. | – | Impala-IQN / C51 | Conv backbone + distributional head |
| Model size | – | 2 | Scale factor for Impala CNN |
| Linear hidden size | – | 512 | Fully-connected layer width |
| Cosine embeddings | $n_{\cos}$ | 64 | IQN quantile embedding size |
| Number of quantiles | $\tau$ | 8 | Quantile samples for IQN |
| Frame stack | – | 4 | History frames per state |
| Image size | – | $84 \times 84$ | Input resolution |
| Trust-region | – | Disabled | Optional stabilizer |
| EMA stabilizer | $\tau$ | 0.001 | Soft target update (if enabled) |
| Munchausen | $\alpha$ | 0.9 | Entropy regularization (if enabled) |
| Distributional | – | C51/IQN | Distributional RL variants |
| Threshold start | $D_{\text{start}}$ | 0.4 | Initial threshold ratio |
| Threshold decay | $D_{\text{decay}}$ | 0.98 | Multiplicative decay factor |
| Threshold interval | – | 5000 steps | Decay period |
| D-strategy | – | none / minnumber / lastact / slownet | Action selection rule |
| Training frames | – | 200M | Total Atari interaction budget |
| Evaluation freq. | – | 250k frames | Eval episodes per checkpoint |
| Independent runs | – | 5 seeds | Reported mean/std |

**Training Procedure:**

- Interact with the environment for 100M frames using $\epsilon$-greedy exploration

- Store transitions into a replay buffer and update the Q-network with Adam optimizer

- Report mean and standard deviation over 5 independent seeds

We observe that augmenting BTR with the threshold strategy improves performance in `NameThisGame` by over 10% compared to the baseline.

## J    LLM USAGE

We used large language models (LLMs) only for minor language polishing and for assistance in generating plotting scripts. No LLMs were involved in the research ideation, theoretical derivations, experiment design, or analysis. All scientific contributions of this work are entirely our own.

