# OpenReview forum: "List Replicable Reinforcement Learning"
_ICLR.cc/2026/Conference — Submitted to ICLR 2026_

### Official Review · Reviewer_neL8 · 2025-10-20

**Soundness:** 2
**Presentation:** 2
**Contribution:** 2
**Rating:** 2
**Confidence:** 5

**Summary:**

The paper focuses on replicability in the context of RL, with the ultimate goal of designing provably efficient PAC RL algorithms that not only converge to an $\epsilon$-optimal policy with high probability after a polynomial number of samples, but also that are guaranteed to always output a policy belonging to a given list $\Pi(M)$ with polynomial size (weak replicability), or additionally that any policy played by the algorithm during the learning process also belongs to a given list (strong replicability).

After having described the high-level idea and intuition in Section 4, the paper presents a robust planning algorithm, that takes in input an estimated MDP, and computes a near-optimal policy based on some input parameter $r_{\text{action}}$ that trades-off accuracy and replicability. Next, the authors present a strongly replicable RL algorithm. Finally, some numerical simulations are conducted.

**Strengths:**

- The paper provides both theoretical guarantees and numerical simulations to support their claims.

**Weaknesses:**

In summary: it is not clear why this notion of replicability is desirable. Moreover, the theoretical analysis is quite poor (too large complexities) and the numerical simulations are too limited (mostly continuous problems) and provide very weird results (the robust planner improves performance?). Lastly, the presentation of the paper is quite poor.

- main contributions: it is not clear why this notion of k-list replicability in RL is desirable. Even though it was desirable, it is not clear why the algorithms presented by the authors are meaningful to achieve it, as they have prohibitive sample complexities (e.g., Algorithm 2 requires $H^{24} S^{11}$ samples !!!) and also quite prohibitive list complexities (e.g., Algorithm 2 requires $S^3$, which limits to very small problems). I recognize the value of these results as "binary" to tell us that polynomial instead of exponential complexity can be achieved, but the rates are too large.
- other limitations: the approach can work only in problems with a finite number of actions (because the main idea is simply to take smallest lexicographically)
- numerical simulations: all simulations (except for mountain car) concern continuous state spaces and not tabular MDPs which is the focus of the paper and the setting where the algorithms work. Moreover, except for cartpole, the claim of the authors at lines 460-461 that their approach makes the RL algorithm more stable is not clear by the graphs. In addition, it is quite strange and surprising that using the robust planner, which by definition is robust and suboptimal (!), actually improves performance for some unclear reason (see Fig. 2).
- presentation: section 4 is too dense and confusing; there is no formal presentation of the results (e.g., the notion of sample complexity in corollary 5.3 is not formalized); some results are reported only in the appendix (e.g., Theorem F.3); the paper redefines some standard symbols (e.g., advantage function in Eq. 1) and re-prove some standard results (e.g., Lemmas E.1,E.2) of RL theory (e.g., see "Reinforcement Learning: Theory and Algorithms"); there are many typos (e.g., bad references like "National Academies of Sciences... 2019" line 033); the presentation of related works through a single giant list of papers is quite useless.

**Questions:**

Please, address all the weaknesses above and questions below:
- Why should we care about the notion of replicability that you consider in the context of RL?

---

> ### Author Response · Authors · 2025-11-23
>
> We are extremely thankful for reviewer's insightful feedback.
>
> **1** . We agree that $(k,\delta)$-list replicability is a relaxation compared to requiring a single replicable output. However, we believe the distinction between an exponential and a polynomial bound on $k$ is substantial: in our view, an exponential versus polynomial dependence in $k$ is of the same importance as the usual exponential versus polynomial separations in sample or time complexity. By contrast, for sample complexity itself it is standard in the literature to focus on the order of dependence (for example, on $|S|,|A|,H,1/\varepsilon$) rather than the exact numerical magnitude.
>
> **2.** We acknowledge this point. However, current theoretical work in this area almost always focuses on discrete models and state spaces. Given that this is, to the best of our knowledge, the first paper on list-replicable reinforcement learning, we view the present discrete-state analysis as already covering a broad and nontrivial setting. Extending our results to continuous state and action spaces would require additional technical development (for example, function approximation and measure-theoretic complications), and we therefore leave this extension as an interesting direction for future work.
>
> **3.** We acknowledge the disconnect between the current theoretical framework and the experimental validation.
>
> (1) To narrow this gap, we have added more directly relevant experiments in Appendix~B of the revised manuscript. Instead of using indirect variance metrics to characterize stability, we now directly count the number of distinct policies produced under different values of $r$. This provides direct empirical evidence for the ``list replicability'' phenomenon suggested by our theory.
>
> (2) We would also like to clarify that the experiments in the main text are not designed to be a direct empirical test of replicability. Their main purpose is to illustrate the practical effect of our robust planner: by inserting this simple planning step, the algorithm can substantially reduce variance and in some cases even improve performance. If the reviewers feel that this connection to the theory is not sufficiently tight, we are happy to move or remove these experiments and keep only the new experiments that directly measure the number of distinct policies, as presented in the revised version.
>
> (3) We believe that the performance gains we observe on Atari may be related to the higher complexity and volatility of these environments. In such settings, small approximation errors and frequent action switching can have a large impact on performance.
>
> In particular, Reference~[1] makes the following observation:
>
> {Without IQN and Munchausen (they are employed to reduce action switching) the agent experiences very low action gaps (absolute Q-value difference between the highest two valued actions), causing the agent to swap its argmax action almost every other step. This is likely to result in approximation errors altering the policy and causing a high degree of off-policyness in the replay buffer. This is particularly detrimental in games requiring fine-grained control, such as Phoenix where the agent needs to narrowly dodge many projectiles, reflected in BTR’s performance without these components.}
>
> We view our robust planner as a more direct mechanism to address exactly this issue (by enforcing stability with respect to small Q-value differences), and the experimental results are consistent with this interpretation.
>
> (4) In Figure~X, because results for many different values of $r$ are plotted together, the effect is not very striking at first glance. However, a closer look at the early part of the Acrobot plot and the final part of the Mountain Car plot shows a clear separation in variance. Moreover, the contrast between larger values of $r$ and the baseline $r=0$ is quite pronounced, which is precisely why we felt it was important to include these experiments.
>
> **4.**
>
> (1) Due to space constraints, we cannot present all technical details and intermediate results in the main text, so some of them are deferred to the appendices. To make the structure easier to follow, we have added a short guide to the appendices and included a proof outline that highlights the main ideas and how the pieces fit together.
>
> (2) We retain some standard notations and well-known intermediate results in order to keep the logical flow clear and to make it easier for readers familiar with the literature to follow the arguments.
>
> We hope that these clarifications and revisions address your concerns. We would be very happy to further adjust the exposition or add additional results or discussion if there are other points that, in your view, would help to make the scope and implications of our results clearer.
>
>
> [1] Clark, T., Towers, M., Evers, C.,  Hare, J. (2025). Beyond the rainbow: High performance deep reinforcement learning

---

> > ### Comment · Reviewer_neL8 · 2025-11-25
> >
> > Thank you for the rebuttal.
> >
> > - I still do not see an answer to my main question. As an RL practitioner, why should I desire a list replicable algorithm? It is not clear to me.
> > - In 1., are you saying that in the literature it is standard to not care exponents of $S,A...$? If so, I disagree, because in RL theory exponents care a lot (there are a lot of works about trying to match upper and lower bounds to identify the optimal minimax rates)
> > - I appreciate the effort of the new experiments in Appendix B.
> >
> > However, I am still convinced that this paper is not ready for publication because (1) not clear why list replicability is important; (2) theoretical analysis is poor (in the sense that the bounds are too large); (3) the simulations in the main paper are not meaningful (not convinced by your answer on robust method, and too variance to extrapolate results from the plots).
> >
> > For these reasons, I will keep my score.

---

> > > ### Author Response · Authors · 2025-12-03
> > >
> > > We thank the reviewer for carefully reading our rebuttal and for the follow‑up comments. We address the remaining concerns below.
> > >
> > > **(1) Why should an RL practitioner care about list replicability?**
> > >
> > > We apologize for not explaining the practical motivation clearly enough in our original submission and rebuttal.
> > >
> > > Informally, list replicability means that when we rerun the learning procedure on fresh data (or with different random seeds), the learned policy will, with high probability, belong to **a small and fixed list of policies**. We note that ρ‑replicability only guarantees that the algorithm returns the same policy under a fixed random seed; it does not provide any control over the set of policies that may be returned under different random seeds. Therefore, list replicability can be viewed as a strengthened form of ρ‑replicability.
> > >
> > > From a practitioner’s perspective, this has several concrete benefits:
> > >
> > > **Predictable behavior under retraining.**
> > >
> > > In many RL applications (e.g., robotics, recommendation, offline RL in high‑stakes domains), practitioners routinely retrain on new batches of data. Standard algorithms can produce qualitatively different policies across runs, even when their performance on training data is similar. List replicability guarantees that the space of possible deployed policies is confined to a small, pre‑specified set.
> > >
> > > **Auditing, safety, and compliance.**
> > > If the algorithm is list‑replicable, one can audit and stress‑test each policy in the list (for safety, fairness, constraint satisfaction, etc.), and then be confident that any future retraining will produce one of these audited policies. This substantially reduces the overhead of repeated verification compared to dealing with an unconstrained, potentially huge space of policies.
> > >
> > > **Robust offline evaluation and model selection.**
> > > In offline RL, practitioners typically evaluate a small number of candidate policies using logged data. List replicability provides a formal guarantee that the policies that one may need to evaluate and compare are restricted to a small list, rather than an arbitrary policy that appears only in one particular run.
> > >
> > > **(2) On the theoretical analysis and the exponents in the bounds**
> > >
> > > We fully agree with the reviewer that exponents in RL bounds are important, and that much of modern RL theory focuses on matching upper and lower bounds and identifying optimal minimax rates. We did not intend to suggest that the community “does not care” about exponents.
> > >
> > > What we aimed to convey (and clearly did not articulate well) is:
> > >
> > > 1. This paper is a **first step** toward understanding list replicability in RL.
> > >
> > > 2. Our main theoretical message is that list replicability is compatible with polynomial sample complexity under standard RL assumptions, i.e., that one can obtain list‑replicable algorithms without blowing up sample complexity to something super‑polynomial.
> > >
> > > 3. We do not claim that our bounds are tight; improving the dependence on ε, δ, the horizon, and other parameters is an important direction for future work.
> > >
> > > Because we are enforcing a stronger requirement (list replicability) than standard PAC‑style guarantees, some degradation in the rates is expected. Our theorems provide the first nontrivial guarantees that such a property is achievable in RL with polynomial overhead. As is common when a new desideratum is introduced (e.g., safe RL, fairness constraints, differential privacy), the first theoretical results are rarely rate‑optimal; subsequent work typically sharpens exponents and closes gaps to known lower bounds. We expect a similar trajectory here.
> > >
> > > **(3) Empirical evaluation**
> > >
> > > We appreciate that the reviewer took the time to look at the additional experiments in Appendix B.
> > >
> > > To address the concern that the current plots are too noisy and not sufficiently informative, we plan to move the key experiments from Appendix B into the main text and reorganize them so that they more directly mirror our theoretical results . This will make the empirical section better connected to the theory, and will give readers a clearer picture of how the list‑replicable algorithm behaves in practice.

---

### Official Review · Reviewer_cvGb · 2025-10-30

**Soundness:** 3
**Presentation:** 1
**Contribution:** 3
**Rating:** 4
**Confidence:** 3

**Summary:**

This work studies a PAC version of tabular RL under the notion of list-replicability. It provides results in generative as well as episodic exploration settings using weak-as well as strong list-replicability. The manuscript ends with deep RL experiments using updates updated from the theoretical algorithms.

**Strengths:**

**Strengths**
**Motivation**
* The problem of studying architectural components that improve deep reinforcement learning is important and well motivated.

**Related Work**
* The related work section on tabular reinforcement learning in both the generative and episodic exploration settings seem largely complete.

**Novelty**
* The paper provides 3 novel results in the area of replicable RL. I am not aware of any results on list-replicable RL.

**Theoretical results**
* While I did not check all the proofs in detail (see in weaknesses) I believe on a high level that the algorithms are correct.

**Weaknesses:**

**Clarity**
* One key issue that I have with the manuscript is that the structure is very difficult to follow. The text often jumps from algorithm to algorithm rather than finishing one of them completely before moving on. The manuscript follows a traditionally more theoretical standard in which an intuitive explanation of the results is given in the introduction which is then usually later followed by formal statements. In this manuscript however, there is first an intuitive section then a slightly less intuitive section and then (only for some results) a formal part. This leads to repetition and makes it hard to follow which part is following from which.
    * The formal section should establish all definitions and theorem statements that are the main contributions formally. I believe section 3 would benefit from the usage of definitions rather than paragraphs only. Section 4 only points to the informal statements of the main results. Section 5 then proves things required for the results in section 4. This structure is confusing to me.
    * Section 4 describes the three main results but then only 1 of them gets a formal section in section 6. I think it would be significantly easier to follow the manuscript if each result had its own subsection. One way to do that would be to describe the result intuitively, then give the algorithm and formalize the result. One could also simply talk about the formal and omit the intuitive explanation from section 1. There are a lot of possibilities that would make the text easier to read.
* The notation and description in the example at the beginning of section 4 is confusing. After reading it multiple times I believe I understand it now. I think a simple Figure would help alleviate this confusion since the example in hindsight is not all too complicated.
* I tried to look at the proofs for correctness but the Appendix is largely a conglomerate of Lemmata and Theorem statements without guidance for the reader. This is incredibly difficult to read because I’m not even sure where to start.

**Related Work**
* While the treatment of the RL related work is quite nice, replicability falls a little short in the related work section. There is a large chunk of work in the area and to demonstrate that the presented work is in fact novel, citing relevant replicability work may be useful.

**Empirical Claims and Evidence**
* The experiments are conducted with old algorithms that are known to have issues when it comes to function fitting, raising concerns about spurious effects that are unrelated to the stability from statistical noise.
* The experiment section claims that with the proposed approach “the performance of the algorithm becomes more stable at the cost of worse accuracy.” It is not defined what that means. The work sets out to study list-replicability but at no point is there any evidence that the policies that are being learned are similar. If the metric of stability that is meant is variance, then across the first three experiments the results are effectively statistically insignificant. I cannot make out any difference at all in Acrobot or MountainCar between the different r values.
* The final experiment directly contradicts the theory. The theory argues that list-replicability is more expensive in terms of sample complexity. It does not tell us anything about improved return. It is unclear what this experiment is showing with respect to the stability that the paper is studying.

In summary, I believe that the clarity changes can easily be made and should not hinder acceptance. The big issue is the empirical section. As of right now, the empirical section is disconnected from the theory and the claims made in it are not supported by sufficient evidence. There are two solutions in my mind. First, the empirical section can simply be removed which would get rid of claims that are not substantiated. The theoretical results are likely of interest to the PAC/replicability community regardless. The second option is to strengthen the experiments to validate the claims that are made via additional measurements and by relating the results to the theory.

**Questions:**

None

---

> ### Author Response · Authors · 2025-11-23
>
> We are extremely thankful for reviewer's insightful feedback.
>
> **First**, we would like to highlight that we have made improvements to the current version. We briefly summarize them here, and the details can be found in the global response: **(1)** Added numerical experiments. **(2)** Expanded related work **(3)** Give a guide to the appendix and include proof outlines **(4)** Correct the typos.
>
> **Clarity**: We acknowledge that the writing in the main text is a little disjointed, and the appendices lack a guide for readers. To address this issue, we have added an appendix guide and roadmaps for robust planner, weak replicability and strong replicability in Appendix A , supplemented with several figures to facilitate understanding.
>
> Due to space constraints, we are unable to formally present all three results in the main text. Instead, we only provide a brief overview of the techniques employed and the most critical "strong list replicability" proposition. However, we are very happy to further adjust the structure of the main text if the reviewers request it !
>
> **Related Work**
>
> We agree that the current version lacks certain citations. We have supplemented the related work about replicability and conducted a corresponding discussion in response to the comments.
>
> **Empirical Claims and Evidence**:
>
>
> We acknowledge the disconnect between the current theoretical framework and experimental validation.
>
> First, To address this gap, we have supplemented more directly relevant experiments in Appendix B of the revised manuscript. Instead of using indirect variance metrics to characterize stability, we directly count the number of distinct policies under different values of $r_{action}$ , which provides direct empirical evidence for the "list replicability" proposition.
>
> We also wish to clarify that the experiments we conducted in the previous version are not intended to directly demonstrate replicability. Instead, they aim to illustrate the practical potential of our robust planner: by incorporating this simple operation, the algorithm can significantly reduce variance and even yield performance improvements. Of course, if the reviewers consider the connection between this part and the theoretical section insufficient, we are willing to remove these experiments and retain only the newly supplemented results as presented in the revised manuscript.
>
> **We now address the reviewer' concerns**.
>
> **(1)** We agree that DQN and Double DQN are not new algorithms, but this is precisely why we use them: they are simple, standard baselines that are known to be unstable, so they are a natural testbed for a stability-oriented planner. Our robust planner does not change the function class or optimizer, only the final action-selection rule, so any function-approximation issues affect both variants equally; the difference we observe (smaller run-to-run variance and smaller empirical list size) comes from the planner. In addition, we already plug the same planner into a modern Rainbow-style agent (BTR) on Atari and see similar stability improvements
>
> **(2)** Thanks for the reviewers' reminder. As noted earlier, we have supplemented more relevant experiments. In these three experiments, we indeed used variance to measure stability. Due to the dense aggregation of multiple $r_{action}$ values, the effect is not particularly pronounced at first glance. However, a careful observation of the initial segment of Acrobot and the final segment of Mountain Car reveals a distinct gap in variance. In fact, the difference becomes highly significant when comparing large rvalues with $r_{action} = 0$, which is precisely the rationale for including these experiments.

---

> > ### Author Response · Authors · 2025-11-25
> >
> > **(3)** As noted earlier, this set of experiments is not designed to demonstrate replicability; instead, it serves to illustrate the potential of the robust planner. We note that the following description is provided in Reference [1]:
> >
> > Without IQN and Munchausen ( They are employed to reduce action switching) the agent
> > experiences very low action gaps (absolute Q-value difference between the highest two valued actions), causing the
> > agent to swap its argmax action almost every other step.
> > This is likely to result in approximation errors altering the
> > policy and causing a high degree of off-policyness in the
> > replay buffer. This is particularly detrimental in games
> > requiring fine-grained control, such as Phoenix where the
> > agent needs to narrowly dodge many projectiles, reflected
> > in BTR’s performance without these components.
> >
> > We argue that our robust planner offers a more direct approach to addressing this problem. In fact, empirical results validate its effectiveness in practice, demonstrating substantial practical potential of the robust planner.
> >
> >
> > We hope that these clarifications and revisions address your concerns. We are extremely happy to further adjust the exposition or add additional results or discussions if there are other points you see that would help make the scope and implications of our results clearer and get a better score.
> >
> >
> > [1] Clark, T., Towers, M., Evers, C.,  Hare, J. (2025). Beyond the rainbow: High performance deep reinforcement learning

---

### Official Review · Reviewer_i586 · 2025-10-30

**Soundness:** 2
**Presentation:** 3
**Contribution:** 2
**Rating:** 6
**Confidence:** 4

**Summary:**

This paper addresses replicable RL in the MDP setup, where the output policy (and/or the trace) from the algorithm lies within a small list of policies across different runs. This paper proposes two algorithms with weak and strong $(k,\delta)$ replicability, which output a near-optimal policy with polynomial sample complexity. Numerical results on various game environments show that the proposed planning algorithm helps stabilize the learning process.

**Strengths:**

(+) This paper extends replicable RL to the MDP setup. The analysis and results are technically solid.

**Weaknesses:**

(-) The numerical results cannot reflect the replicability of the algorithm.

**Questions:**

Typo: In Line 158, should it be "at least 1-$\rho$" instead?

Q1. Have you thought about providing numerical results to demonstrate the replicability of the proposed algorithm?

Q2. Does Theorem 1.3 imply that there is a gap of $O(HS)$ between Algorithm 3 and the lower bound?

Q3. Are there any trade-offs between the list complexity and the sample complexity? In the work of [1], I can see that a smaller $k$ tends to associate with a higher regret.

[1] Chen et al. Regret-Optimal List Replicable Bandit Learning: Matching Upper and Lower Bounds. ICLR 2025.

---

> ### Author Response · Authors · 2025-11-23
>
> We sincerely thank the reviewer for the careful reading and helpful comments.
>
> **First**, we would like to highlight that we have made improvements to the current version. We briefly summarize them here, and the details can be found in the global response: **(1)** Added numerical experiments. **(2)** Expanded related work **(3)** Give a guide to the appendix and include proof outlines **(4)** Correct the typos.
>
> **Typo**: Thank you for the meticulous examination, which has helped us identify potential issues. We have corrected it accordingly.
>
> **Q1**: We agree the current figures mostly show stability (variance bars) but not the list size. In the revision we will add direct experiment to demonstrate list‑replicability. The details are presented in Apppendix B.
>
> **Q2**: Yes. Algorithm 3  achieves list size $(H|S||A| + 1)(H|S| + 1) = O(|S|^2|A|H^2)$, while Theorem 1.3 gives a weak-list lower bound $\Omega(|S||A|H)$. Hence there is an $O(|S|H)$ multiplicative gap.
>
> We explain in Appendix F that our upper bound factors arise from (i) **action-gap discretization** $ \leq |S||A|H + 1 $ policies due to Algorithm 1 , and (ii) the **monotone reachability-truncation patterns** $\leq |S|H + 1$ ; multiplying yields the stated bound. A more intuitive explanation can be found in Appendix A , which include the proof roadmap.
>
>
> However, we note that the list constant for the generative model is $O(|S||A|H)$ , which to some extent suggests that our lower bound might be tight. We tried to make the gap between the upper and lower bounds vanish, but this is not easy, because we need a truncated MDP in both weak and strong replicability.
>
> **Q3**: No. We believe this is also one of the elegant aspects of our results: Theoretically, the upper bound of the list complexity only depends on $S$, $H$, and $A$, while the sample complexity is negatively correlated with the tolerance threshold and the failure probability. This is mainly because in our setting, when $\epsilon$ is small: the sample complexity increases, and the scale of our constructed $r_{action}$ shrinks in sync with the confidence interval. More intuitively, regardless of the scale, the list complexity depends only on the number of gaps in the confidence interval. This can be seen from the figure in Appendix A. But in practice, as shown in Appendix B, when $r_{action}$ increases, the regret will increase and the list length will decrease.
>
>
> We hope these changes and clarifications address your concerns. We are happy to further revise the paper to make the scope and implications of our results clearer and better. We believe this work is quite solid, and we want to make it as good as possible. Please let us know any concern and new suggestion!

---

### Official Review · Reviewer_PCZR · 2025-11-01

**Soundness:** 3
**Presentation:** 3
**Contribution:** 2
**Rating:** 6
**Confidence:** 3

**Summary:**

This paper introduces the notion of list replicability in the PAC reinforcement learning (RL) framework to formally address instability across training runs. The authors define weak and strong forms of list replicability, requiring that the learned policy—or the entire policy execution trace—lies within a small list of possible outcomes. The main contributions include new tabular RL algorithms, one which is weakly list replicable and one which is strongly list replicable, that achieve polynomial list and sample complexities, in contrast to the exponential instability of existing methods. A complementary lower bound for list complexity is also developed. Empirical results show that incorporating the proposed planner into standard RL frameworks improves stability while maintaining good performance.

**Strengths:**

- **Quality:** The paper’s main claims are supported by rigorous and well-structured proofs, demonstrating a solid theoretical foundation. Additionally, the empirical results, while limited in scope, align well with the theoretical claims and demonstrate practical relevance.
- **Clarity:** The paper's central contribution—introducing list replicability as a performance criterion in RL alongside efficient algorithms—is both well-motivated and clearly presented. The exposition is supported by intuitive explanations, and informal versions of the main theorems are introduced early in the text to guide reader understanding.
- **Significance:** This work addresses an important issue in RL—replicability—by proposing a formal framework and provably efficient solutions. Notably, the framework extends beyond the theoretical tabular setting, demonstrating practical utility when integrated into empirical RL algorithms.
- **Originality:** The paper introduces list replicability in the context of PAC reinforcement learning, extending ideas from replicable learning and multi-armed bandits to the more complex RL setting. The authors are the first to demonstrate that even strong list replicability—constraining the entire policy execution trace—can be achieved with polynomial complexity in tabular RL.

**Weaknesses:**

Rather than separating into broad quality, clarity, significance and originality categories, I will outline my main concerns in a more detailed manner below.
- Theorem 1.3 establishes a lower bound on list complexity of $\Omega(SAH)$ for weakly list-replicable RL algorithms, which is notably lower than the upper bounds achieved by Algorithms 2 and 3. This discrepancy raises the question of whether the proposed algorithms admit non-tight bounds that could be further improved. A discussion addressing the gap between upper and lower bounds would strengthen the paper by clarifying whether the current techniques are potentially suboptimal or if the bounds reflect inherent limitations.
- Algorithm 3 exhibits a higher list complexity of $O(S^3 A H^3)$ compared to Algorithm 2's $O(S^2 A H^2)$, despite offering stronger guarantees. A brief discussion clarifying this trade-off—whether it reflects inherent algorithmic requirements or potential room for tightening the analysis—would enhance the paper’s clarity.
- In Figure 1, the authors claim that increased stability comes at the cost of reduced performance. However, this trade-off is not clearly visible in subfigures 1(b) and 1(c), where the degradation in performance is minimal or absent. Conversely, Figure 2 appears to show both improved stability and increased performance when using the robust planner. These observations create ambiguity: it is unclear whether the proposed method generally entails a trade-off between performance and stability, or if it can yield gains on both fronts. A discussion clarifying this point would significantly strengthen the empirical section.
- Although Algorithms 2 and 3 achieve list replicability with polynomial list complexity, their sample complexity scales with $1/\delta$, in contrast to the more favorable $1/\log \delta$ dependence of standard PAC RL algorithms [1]. This increased sample requirement may help explain the reduced performance observed in Figure 1 when the robust planner is applied, suggesting a potential trade-off between replicability guarantees and sample efficiency.
- As the authors note, the theoretical analysis is restricted to the tabular PAC RL setting, which limits direct applicability to large-scale or real-world problems. However, this is not a major concern, given that list replicability is a newly proposed performance criterion. Moreover, the paper demonstrates that key components of the framework can be effectively adapted to practical RL algorithms, supporting its relevance beyond the theoretical setting.
- The authors assume known and deterministic rewards, and briefly mention that their methods can be extended to handle more general cases. However, the paper does not provide further details on how such extensions would be implemented or how they might affect the theoretical guarantees.
- While the paper presents its results within the PAC RL framework and frequently refers to sample complexity, it does not formally define either term. Including precise definitions for PAC RL and sample complexity early in the paper would enhance clarity and make the work more accessible to readers who are less familiar with these concepts.
- Additionally, the paper would benefit from more thorough proofreading. For instance, in line 69 ‘work’ should be replaced with ‘works’. In line 158, $\delta$ should be replaced with $\rho$.  In line 249, ‘the’ should be removed.

[1] Dann, Christoph, Tor Lattimore, and Emma Brunskill. "Unifying PAC and regret: Uniform PAC bounds for episodic reinforcement learning." Advances in Neural Information Processing Systems 30 (2017).

**Questions:**

1. As noted earlier, Figure 1 suggests a trade-off between accuracy and stability, whereas Figure 2 appears to show simultaneous improvements in both when using a robust planner. Can the authors elaborate on why this occurs, and in what scenarios we should expect each phenomenon?
2. Given the gap between the list complexity upper bounds of Algorithms 2 and 3 and the lower bound established by the hardness result, do the authors believe there is room to tighten their analyses or improve the algorithms further?
3. Additionally, is it possible to establish a hardness result for the strong list replicability setting? If not, what are the main technical obstacles preventing such a result?
4. The proof of Theorem 1.3 is said to rely on a reduction to the multi-armed bandit setting, leveraging known lower bounds. Could the authors clarify what technical challenges arise in this reduction, and what their main contributions are in making it work for the RL setting?

**Details Of Ethics Concerns:**

I do not have any concerns.

---

> ### Author Response · Authors · 2025-11-23
>
> We are extremely grateful for your detailed and constructive feedback.
>
> **Weakness (2)** We argue that further improvements in the analysis are challenging, as the analysis must be conducted layer by layer with inherent dependencies between consecutive layers.
>
>
>
> **Weakness (4)** Thanks for the reviewers' valuable views; we find this perspective highly insightful.
>
>
>
> **Weakness (6)** We note that stochastic rewards do not increase the difficulty of the proof. This is because we observe that for the same sample size, if $\|P - \hat{P}\| < \epsilon_0,$ then it necessarily holds that $|r - \hat{r}| < \epsilon_0.$ The order of magnitude in the subsequent analysis remains unchanged, with only potential adjustments to the constant terms.
>
>
> **Weakness (7)**  Thanks for the reviewer' suggestion. Due to space constraints in the main text, we have incorporated this content into Appendix A.
>
> **Weakness (8)**  Thanks for the reviewers' careful checking; we have corrected these typos.
>
>
> **W3\&Q1**:We argue that the performance improvement of our method on Atari environments may stem from the greater complexity and volatility of these environments.
>
> We note that the following description is provided in Reference [1]:
>
> Without IQN and Munchausen ( They are employed to reduce action switching) the agent
> experiences very low action gaps (absolute Q-value difference between the highest two valued actions), causing the
> agent to swap its argmax action almost every other step.
> This is likely to result in approximation errors altering the
> policy and causing a high degree of off-policyness in the
> replay buffer. This is particularly detrimental in games
> requiring fine-grained control, such as Phoenix where the
> agent needs to narrowly dodge many projectiles, reflected
> in BTR’s performance without these components.
>
> We think that our robust planner offers a more direct approach to addressing this problem and as we hypothesized, the experimental results are highly promising.
>
> **W1 \& Q2**:We believe there is limited room for further analysis. Firstly, for the generative model, its upper bound aligns with the hardness results. However, the setup of our Algorithm 2 differs significantly from that of the generative model—primarily, we must remove certain states. We have attempted to use fixed values or a constant ratio for \( r_{\text{action}} \) and \( r_{\text{trunc}} \), but this approach did not yield effective results.
>
> **Q3**:We currently do not have a matching hardness result for the strong notion of list-replicability, and we view this as an interesting open problem.
>
> Our weak lower bound (Thm. 1.3) comes from a reduction to BESTARM bandits and only needs to track which final policy is output. In the strong setting, however, the object that must lie in a small list is the entire exploration trace (sequence of policies/actions). To get a strong lower bound one would need an instance where restricting the set of possible traces is itself information-theoretically costly. Technically, this requires (i) encoding hardness into the exploration trajectory rather than a single “critical” decision layer, and (ii) bandit-style lower bounds that depend on the number of possible traces, not just the number of final actions. We do not know how to do this at present, so we leave strong list-replicability hardness as an explicit open question.
>
>
> **Q4**: Theorem 1.3 reduces a hard list-replicable bandit instance to a family of episodic MDPs. The main technical challenge is that RL policies are much richer objects than bandit arms: a policy chooses actions at many states and time steps, whereas a bandit algorithm outputs a single arm.
>
> We address this by designing a layered “key-layer” MDP (Fig. 3) where each bandit arm corresponds to a unique choice at one key layer, so that optimal policies are in one-to-one correspondence with bandit arms and each episode simulates exactly one bandit pull. A second challenge is to transfer the list-replicability guarantee: we show that any small list of RL policies for this MDP family induces a comparably small list of bandit arms with essentially the same success probability. With these two ingredients, the known lower bound for list-replicable bandits directly yields the Ω(∣S∣∣A∣H) lower bound in Theorem 1.3.

---

### Official Review · Reviewer_BYFm · 2025-11-09

**Soundness:** 4
**Presentation:** 3
**Contribution:** 2
**Rating:** 2
**Confidence:** 4

**Summary:**

This paper gives a black box reduction from tabular RL algorithms to a list replicable algorithm for the same setting. They give algorithms satisfying two different notions of list replicability: one requiring that only the final policy output by the algorithm lies in some instance-dependent list (weak), and one requiring that all policies used for exploration lie in some instance-dependent list of policy traces (strong). Their weakly list replicable algorithm has list size $k = O(|S|^2|A|H^2)$, and requires $|S|H $ calls to the non-replicable algorithm, while the strongly list replicable algorithm has a modest increase in list size to $k = O(|S|^3|A|H^3)$ and has polynomial sample complexity in all relevant parameters (though unfortunately depends polynomially on $1/\delta$ rather than $\log(1/\delta)$. As an additional contribution, they empirically show that their stable approach to planning can be incorporated into existing RL algorithms to improve stability, giving an approach to variance reduction that may be generally useful.

**Strengths:**

The paper is very well written and the novel techniques are clearly explained. This is to the best of my knowledge the first work to consider the question of list-replicable RL, and so contributes to our theoretical understanding of the feasibility of stable RL.

The approach to stable planning is a nice contribution in that it is simple enough to be adapted to a variety of algorithms and empirically improves stability, a common problem for empirical RL.

**Weaknesses:**

My concerns regarding the results are mostly about comparison to prior results in replicable RL.

First, this paper omits reference to related work [1] that seems algorithmically similar. [1] improves upon prior work on replicable RL by giving more sample efficient replicable algorithms in the tabular setting. Their results also rely on this idea of stably learning a collection of ignorable states, then doing backward induction with data collected from unignorable states to learn a good policy.

Second, as this paper acknowledges, replicable algorithms imply list replicable algorithms, though with potentially exponential list size, depending on the amount of randomness required to run the replicable algorithm. To understand the contribution of this work, it seems important to discuss whether existing replicable algorithms have randomness usage that directly implies list-replicable algorithms.

[1] “From Generative to Episodic: Sample-Efficient Replicable Reinforcement Learning” Hopkins, Liu, Ye, Yoshida

**Questions:**

Please see weaknesses. How do the list sizes in this paper compare to those implicit in prior work on replicable RL for tabular MDPs? How do the algorithmic techniques in this paper compare to those of [HLYY25]?

---

> ### Author Response · Authors · 2025-11-23
>
> We sincerely thank the reviewer for the insightful and helpful comments !
>
> **Weakness (1)**: We appreciate the reviewers' reminder. This work is indeed highly relevant to ours, and we have incorporated a discussion on this study in the revised manuscript. Conceptually, HLYY and our work share the high-level idea of identifying a set of states that can be treated as “ignorable” and then performing some form of backward induction on the remaining states using data collected during exploration. However, the formal goals and guarantees are different. HLYY design sample-efficient ρ‑replicable algorithms in tabular MDPs: with a fixed random seed, two runs output the same policy with high probability, and they optimize the sample complexity for this notion. In contrast, we study (weak and strong) list replicability, where the focus is on bounding the number of distinct policies (and even full policy traces) that can appear across all runs. Our main results show that one can achieve near-optimal PAC guarantees while keeping the list size polynomial in  $∣S∣,∣A∣,H$. To the best of our knowledge, HLYY do not analyze the list complexity of their algorithms, and their analysis does not give a nontrivial upper bound on the number of distinct policies that may arise as the random seed varies.
>
> **Weakness (2)**： We have also cited relevant literature and incorporated corresponding discussions into the revised manuscript. we agree that any ρ‑replicable algorithm can be seen as a weakly list‑replicable algorithm by taking the list to be the set of all policies that can be output as we vary the internal randomness. However, this transformation only gives the trivial bound that the list size is at most the number of possible random seeds, which can be exponential in $∣S∣,∣A∣,H$.
>
> In addition, we would like to point out that our list-replicability guarantee also implies 𝜌-replicability. When the random seed is fixed, equivalently, when  $r_{action}$ and $r_{trunc}$ are fixed, our algorithm will return the same policy, so list-replicability can be viewed as a stronger notion than 𝜌-replicability. And also, in this line of work, our algorithm is the first one that ensures replicability not only for the returned policy but also for the entire execution trajectory.
>
> We are very grateful to you for pointing out these missing and important references; your comments have substantially improved the accuracy and balance of our paper writing. Overall, we believe the revised version now presents a solid, important and impressive result, with a more careful and accurate framing of its scope and relation to prior work. We are extremely happy to further adjust the paper if you think additional clarifications would be helpful, and we are fully willing to revise again in line with any further suggestions!

---

### Author Response · Authors · 2025-11-23
**Global Response**

We sincerely thank the reviewers for their detailed and constructive feedback.

In this round of response, we have made the following revisions.

1.**Added numerical experiments**.

Regarding the phenomenon of disconnection between our theory and experiments mentioned by the reviewer, we added  experiments to directly demonstrate list‑replicability.

Over independent runs with fresh samples in chainMDP and GridWorld , we count the number of distinct final deterministic policies  produced by both the robust planner and the greedy planner. The results are presented in Appendix B:

We observe that under the greedy strategy, the number of returned policies is almost the same as the number of runs; and under robust planner, as $r_{\text{action}}$ increases, the number of distinct returned policies not only decreases but drops to a very small level. This further confirms that once robustness is added, the algorithm compresses the exponentially many policies down to  polynomial number.


2.**Expanded related work**.

We have included some comparisons with different literatures. In particular, we have conducted a detailed comparison of recent literature on reproducibility.

3.**Give a guide to the appendix and include proof outlines**.

Specifically, we added a table of contents for the appendix and briefly introduced the content of each appendix section. Then, we provided the proof outlines for the robust planner, weak replicability, and strong replicability. Theses are presented in Appendix A. We also include several figures to facilitate understanding. We hope this will address the reviewer’s concerns about unclear structure. Of course, if the reviewer finds it necessary, we are also open to further adjusting the structure of the main text in future revisions.

4.**Correct the typos**.

We sincerely thank the reviewer for the careful reading and for pointing out the typos. We have corrected them accordingly.


 We also addressed other minor issues. We will detail them in the individual response.

We believe our revision should address the concerns. However, please let us know any further suggestions and concerns, and we are willing to revise accordingly!

---

### Meta-Review · Area_Chair_8PrB · 2025-12-29

**Summary:**

In general, the reviewers had concerns regarding the relationship with related work, the paper's presentation and structure, and the overall quality of the empirical evaluation. On the positive side, the paper has a clear contribution.
There were mixed impressions regarding the significance of the problem.

The rebuttal partially addressed some of the concerns but ultimately seems insufficient to convince most reviewers to increase their ratings.

In summary, the paper needs a significant revision to do justice to its contribution.

**Reviewer Concerns:**

- `BYFm` found the comparison with related work on replicable RL insufficient and missed a comparison with other replicable algorithms that also use randomization implicitly, like the proposed approach. The author partially incorporated the feedback into the manuscript. Nevertheless, it is unclear if the rebuttal was sufficient to address the second question.
- `PCZR` raised a large number of detailed points (mostly suggestions for further discussion and improvements). The rebuttal shows that the points are relevant and mentions that the paper has been improved accordingly.
- `i586` was positive about the paper but pointed out that the empirical results could assess the replicability of the experiments directly. Some results were added to the appendix, but that does not seem sufficient to address the concerns of the reviewer as they should be presented in the main paper.
- `cvGb`
	- also raised concerns about the metrics presented in the empirical evaluation. New results were added to the appendices during the rebuttal period; nonetheless, it is unclear why the main document still considers older results that assess the algorithm's replicability only indirectly.
	- mentioned that the paper has considerable issues regarding the structure and presentation. The authors added a roadmap to the appendix; however, this does not seem to address the problems regarding the presentation of the main document.
	- In summary, I think the reviewer would not be willing to upgrade the score
- `neL8` found unclear the relevance of the problem and approach for RL practitioners,  mentioned that the theoretical analysis could be further refined, and found that the results in the empirical evaluation were not meaningful. The rebuttal was not sufficient to address the concerns of the reviewer.

**Reviewer Scores:**

- `BYFm`: 2 -> 4
- `PCZR`: 6 -> 6
- `i586`: 6 -> 6
- `cvGb`: 4 -> 4
- `neL8`: 2 -> 2

---

### Decision · Program_Chairs · 2026-01-26

Reject